# Cellular and Molecular Mechanisms Underlying Prostate Cancer Development: Therapeutic Implications

**DOI:** 10.3390/medicines6030082

**Published:** 2019-07-30

**Authors:** Ugo Testa, Germana Castelli, Elvira Pelosi

**Affiliations:** Department of Oncology, Istituto Superiore di Sanità, Vaile Regina Elena 299, 00161 Rome, Italy

**Keywords:** prostate cancer, cancer stem cells, tumor xenotrasplantation assay, gene sequencing, gene expression profiling

## Abstract

Prostate cancer is the most frequent nonskin cancer and second most common cause of cancer-related deaths in man. Prostate cancer is a clinically heterogeneous disease with many patients exhibiting an aggressive disease with progression, metastasis, and other patients showing an indolent disease with low tendency to progression. Three stages of development of human prostate tumors have been identified: intraepithelial neoplasia, adenocarcinoma androgen-dependent, and adenocarcinoma androgen-independent or castration-resistant. Advances in molecular technologies have provided a very rapid progress in our understanding of the genomic events responsible for the initial development and progression of prostate cancer. These studies have shown that prostate cancer genome displays a relatively low mutation rate compared with other cancers and few chromosomal loss or gains. The ensemble of these molecular studies has led to suggest the existence of two main molecular groups of prostate cancers: one characterized by the presence of ERG rearrangements (~50% of prostate cancers harbor recurrent gene fusions involving ETS transcription factors, fusing the 5′ untranslated region of the androgen-regulated gene TMPRSS2 to nearly the coding sequence of the ETS family transcription factor ERG) and features of chemoplexy (complex gene rearrangements developing from a coordinated and simultaneous molecular event), and a second one characterized by the absence of ERG rearrangements and by the frequent mutations in the E3 ubiquitin ligase adapter SPOP and/or deletion of CDH1, a chromatin remodeling factor, and interchromosomal rearrangements and SPOP mutations are early events during prostate cancer development. During disease progression, genomic and epigenomic abnormalities accrued and converged on prostate cancer pathways, leading to a highly heterogeneous transcriptomic landscape, characterized by a hyperactive androgen receptor signaling axis.

## 1. Introduction

Prostate cancer is the most frequently diagnosed nonskin cancer and second most common cause of cancer-related deaths in men, with an estimated 1,600,000 cases and 366,000 deaths annually. Despite recent progresses, prostate cancer remains a great medical problem for the men affected, with absolute need to improve the efficacy of current therapies for metastatic disease and to reduce the unnecessary overtreatment of more benign disease. Prostate cancer is a clinically heterogeneous disease with many patients exhibiting an aggressive disease with progression and metastasis and other patients showing an indolent disease with low tendency to progression. The standard treatment of this cancer is based on surgery and radiotherapy; however, patients nonsuitable for radiotherapy or surgery are treated with androgen ablation therapy, which effectively shrinks androgen-dependent tumors. Unfortunately, this treatment is often followed by recurrent androgen-independent prostate cancer, with frequent metastases.

The human prostate contains three cell types: luminal cells (columnar epithelial cells that express secretory proteins, differentiation antigens such as cytokeratin 8, prostate-specific antigen, and high levels of the androgen receptor), basal cells (localized to a lower level express markers such as cytokeratin 5, but express only low levels of androgen receptor), and rare neuroendocrine cells (characterized by the expression of endocrine markers).

Three stages of development of human prostate tumors have been identified: (a) intraepithelial neoplasia that can be considered a precancerous state, characterized by hyperplasia of luminal cells and progressive loss of basal cells; (b) adenocarcinoma androgen-dependent (subdivided into two stages, adenocarcinoma latent and clinical), characterized by the complete loss of basal cells and the strong luminal phenotype: at this stage, the tumor is androgen-dependent and its growth can be controlled by androgen deprivation; and (c) adenocarcinoma androgen-independent (or castration resistant) that represents the evolution of adenocarcinoma and does not depend for its growth by androgens.

During prostate cancer progression, the luminal compartment expands, and basal cells are lost. This corresponds to a luminal phenotype both at immunophenotypic and genotypic levels. Late stages of disease, characterized by castration-resistant prostate cancer and by the development of metastases, enrich for basal cell genes and stem cell genes.

## 2. Tumor Evolution of Prostate Cancer from Precursor Lesions

The histological evaluation of prostate cancers, as well as of other solid tumors is of fundamental importance to assess the biology, the grade of development of the tumor and to have a prognostic projection. For clinical purposes, the histological evaluation of these tumors is expressed in terms of Gleason score: a scoring system that evaluates how much the bioptic prostatic specimen is similar to normal prostate gland (low score, 1 corresponding to normality) or is frankly tumorigenic (high score, 5 corresponding to lack of normal glands and presence of sheets of frankly abnormal tumor cells); between these two extreme grades, there are intermediate grades underlying a progressive transition from a normal tissutal architecture to the progressive loss of tissutal glands and to the acquisition of cellular atypia [1,2,3]. In this evaluation system, the scoring is, rather than assigning the worst grade as the grade of the tumor, the grade was defined as the sum of the two most common grade patterns and reported as the Gleason score [1,2,3]. This evaluation system was and is of fundamental importance in the clinical evaluation of a patient with a prostatic neoplasia. In the time, this scoring system was implemented and in 2014 accepted by the World Health Organization (WHO) [4].

In the actual evaluation system, the Gleason score 1 + 1 = 2 is a grade very rarely diagnosed and corresponding to adenosis (atypical adenomatous hyperplasia, AAH) [4].

Gleason scores 3 and 4 are of very difficult histologic evaluation and the criteria for their assessment are not solid and reproducible [4]. The Gleason pattern 3 (score 3 + 3) corresponds to the presence of highly variably sized and shaped glands, whose glandular architecture is conserved; the Gleason pattern 4 (score 4 + 4) corresponds to the presence of cribriform, poorly-formed, fused glands; finally, the Gleason pattern 5 (score 5 + 5) corresponds to the absence of glandular structures, replaced by sheets, cord, single cells, solid nests and necrotic areas [4]. The actual system of classification of prostate cancers reducing to five prognostic risk categories received support at clinical and genetic level [5].

On the basis of actual evaluation criteria, GS of 3 + 3 = 6 was reclassified into the lowest grade group (GG) of 1, GS of 3 + 4 = 7 to GG2, GS of 4 + 3 = 7 to GG3, GS of 4 + 4 to GG4, and GS of 4 + 5 or 5 + 4 or 5 + 5 to GG5 [5]. According to this classification, the quantity of GP4 or GP5 present in the tumor is a key determinant to assess the risk associated with this tumor. A 5-yr biochemical risk-free survival for prognostic grade groups of 97.5% for PGG1, 93.1% for PGG2, 78.3% for PGG3, 63.6% for PGG4, and 48.9% for PGG5 was reported [5]. The analysis of genetic abnormalities in a large set of prostate cancer specimens subdivided according to the PGG classification system showed that (i) the overall number of somatic mutations slightly increased in risk groups and was particularly significant when comparing lower (PGG1 and PGG2) with higher (PGG3, PGG4 and PGG5) risk groups; (ii) increasing copy number alterations (gain and losses) are observed with increasing PGG; (iii) the large majority of point mutations remained unmodified in the various PGGs, with the exception of *TP53* increasing with PGG; and (iv) the frequent MYC amplification markedly increased with PGG [6].

A putative precursor lesion of prostate cancer is represented by high-grade prostatic intraepithelial neoplasia (HGPIN) that corresponds to a proliferation of prostate glandular epithelial cells displaying clear cytological atypia within the tissue limits of prostatic ducts and acini. HGPIN is considered a precursor lesion of prostate cancer based on two arguments: epidemiological data link HGPINs to the tumor glands and the later occurrence of invasive carcinoma during tumor surveillance; the morphological similarities between epithelial cells of HGPINs and invasive cancer; and colocalization of HGPIN with invasive prostate cancer and their mutually shared genetic rearrangements and other genetic alterations [7].

Thus, several studies have explored the clonal relationship existing between GP3 and GP4 lesions. Sowalski and coworkers have explored a series of adjacent GS3 and

GS4 tumors in radical prostectomy specimens and observed that all were concordant for the *TMPRSS2-ERG* gene fusion: particularly, GS3 and GS4 tumors had identical *TMPRSS2-ERG* fusion breakpoints, thus confirming their clonal origin [8]. These findings were considered compatible with two hypotheses: G3 tumors progress to G4 tumors or G3 and G4 tumors derive from a common precursor lesion [8]. Kovtum and coworkers have analyzed the landscape of large chromosomal alterations in paired GP3 and GP4 lesions by next-generation sequencing and showed that while GP3 and GP4 from the same tumor each possesses unique breakpoints, they also share identical breakpoints, suggesting a common origin [9]. *TMPRSS2-ERG* was the most recurrent rearrangement present in both GP3 and GP4, while PTEN deletion was observed in only a part of *TMPRSS2-ERG* fusion-positive cases [9]. Importantly, hierarchical clustering analysis showed that GP3 exhibits greater breakpoint similarity to its partner GP4, compared with GP3 from other patients [9]. Trock and coworkers performed an analysis of some common genetic alterations of prostate cancer (chromosome 8q gain (*MYC*), 8p loss, and *PTEN* loss) in adjacent GP3 and GP4 tumors in GS6 and GS7 tumors: 8q gain, 8p loss and *PTEN* loss were more common in G3 cores derived from GS7 than GS6 tumors [10].

*TMPRSS2-ERG* is the frequent *ERG* gene rearrangement observed in prostate cancer and SLC45A3 is the second most common *ERG* partner in prostate cancer and in most of patients *SLC45A3-ERG* rearrangements co-occur with *TMPRSS2-ERG* rearrangements [11]. Double rearrangements were relatively rare in GS6 tumors (11.5%) and their frequency increased in GS7 (22.2%) and GS8 (50%) tumors [11]. Double rearrangements together with *PTEN* loss were observed in 0% GS6, 24.7% GS7, and 29.4% GS8 [11].

The analysis of mutational spectrum of GP3 and GP4 tumors allowed defining the time of occurrence of their molecular evolution. Thus, VanderWeele and coworkers examined, by exome sequencing, low-grade (GP3) and high-grade (GP4) foci in four prostate cancers and, in two of these cases, metastatic lesions: 87% of somatic mutations observed in GP3 were private to GP3 foci; GP4 and metastatic lesions displayed a high concordance of the mutational profile; GP4 shared only 9% with GP3, but 82% with metastatic lesions [12]. Mutations in *TP53* pathway were observed only in GP4 and metastatic tumors [12]. These observations are compatible with an early divergence of GP3 from GP4 and metastatic tumors [12].

Similar conclusions were reached in a second study in which sequencing showed that adjacent GP3 and GP4 are clonal based on the presence multiple shared genomic alterations; however, the presence of a large number of unique, nonshared mutations in the GP3 and GP4 tumors suggests that GP4 was not directly derived from GP3 [13]. These findings support a model of branched evolution, based on the existence of an ancestral common precursor from which emerge GP3 and GP4 and then subsequent divergence of these two tumor lesions [13]. These studies open the problem of the identification of the precursor lesion that originates both GP3 and GP4 tumors.

Whole genome DNA sequencing studies carried out on distinct areas of these tumors, including normal prostatic tissue near to the tumor nodules provided evidence that mutations were present at high levels in morphologically normal tissue distant from the cancer, reflecting clonal expansions, thus indicating that the mutational processes operating in the tumor nodules were also at work in normal tissue [13]. The most obvious explanation for this apparently intriguing phenomenon is that in the prostatic tissue an oncogenic field affecting normal prostatic tissue was generated or that normal prostate cells undergo a process of somatic mosaicism involving high mutation rates [14]. The hypothesis of the mutational field may help not only to explain the intrapatient mechanism of cancer development, but also to understand its multifocality [14]. Field effects have been described also for other tumors, such as colorectal cancer, breast cancer, head and neck cancer, and oral cancer [15]. The implication of field effects may have potentially important implications for the understanding of prostate cancer development.

Other studies have explored the genetic alterations observed in HGPIN and adjacent prostate cancer. The mutation profiles of six tumor-associated HGPIN lesions in a single case of *TMPRSS2-ERG* fusion-positive GS7 prostate cancer were evaluated [16]. All six HGPIN foci displayed the same tumor-specific *TMPRSS2-ERG* fusion breakpoint, thus indicating they are clonally related to the adjacent invasive GP3 and GP4 tumor [15]. Among 32 gene targets mutated in the tumor, only mutation of *ORZAP1* gene was found in a single focus of HGPIN [15]. These observations suggest that HGPIN is only a distant precursor of adjacent invasive prostatic adenocarcinoma [16].

Many studies have attempted a comparative molecular genetic characterization of HGPIN and its corresponding prostate cancer to study this tumor progression and transformation. Thus, Jung and coworkers analyzed somatic mutations and copy number alterations (CNA) profiles of paired HGPINs and prostate cancers and reached the conclusion that HGPIN genomes harbor relatively fewer mutations and CNAs and require more genomic alterations to progress to prostate cancer [17]. Furthermore, a study by Haffner and coworkers provided some evidence inducing caution in the interpretation of the data related to the molecular relationship between HGPIN and adjacent prostate cancer [18]. In fact, these authors suggest that invasive prostate adenocarcinoma may morphologically mimic HGPIN through retrograde colonization of benign glands with cancer cells; the same would apply also to intraductal carcinoma adjacent to invasive adenocarcinoma [18].

These observations suggest that the HGPIN lesion adjacent to invasive carcinoma does not necessarily represent its respective precursor lesion and additional studies based on single-cell molecular analysis and lineage tracing studies are required to define such a relationship [7]. Furthermore, recent studies have suggested that some HGPINs are in fact invasive prostate cancers masquerading as a HGPIN-like condition [19].

In spite all the limitations in the definition and identification of precursor lesions of prostate cancer, the search of prostate cancer precursors lesions is of fundamental importance because offers the unique opportunity for disease prevention and treatment [19].

Other two prostate tumor lesions have “large gland” morphology; in contrast with the large majority of prostate cancers exhibiting “small gland” morphology: PIN-like adenocarcinoma and ductal adenocarcinoma [20]. PIN-like carcinoma is a rare and is a variant of acinar carcinoma that is morphologically characterized by large cancer glands lined with pseudostratified epithelium similar to HGPIN [21]. A recent study explored the biologic and clinical features of these tumors showing that they are usually limited in size, not advanced in stage (more than 90% of these tumors correspond to a GG score of 1–3), not associated with high-grade prostate cancer on radical prostatectomies and show frequent TMPs-ERG rearrangement [21].

Ductal adenocarcinoma is a histologic subtype of prostate carcinoma with large glands lined with tall columnar pseudostratified epithelium [20]. It is typically associated with acinar carcinomas and occurs in 3–6% of prostate cancers (with only 0.2% having a pure ductal morphology), and induces a disease more aggressive than acinar carcinomas and is associated with higher stage and risk of recurrence and mortality [21]. Few data are available about the molecular features underlying this histologic subtype: (a) studies using fluorescence in situ hybridization have shown a prevalence of TMPRSS2-ERG fusions in ductal cases than in matched pure acinar adenocarcinoma cases [22,23]; (b) PTEN loss by immunohistochemistry was less common among ductal carcinomas and their synchronous acinar tumors, compared to matched pure acinar carcinomas [23]; (c) 40% of ductal adenocarcinomas display a mismatch repair gene alteration at the level of *MSH2* or *MSH6* genes and 75% of these cases have evidence of hypermutation [24]; (d) ductal/intraductal adenocarcinoma histology is frequently (48%) associated with germline DNA repair gene mutations (*BRCA2*, *ATM*, *CHEK2*, and *BRCA1*) [25]; and (e) exome sequencing studies showed the occurrence in 30% of cases of *CTNNB1* hot spot mutations in the ductal component, but not in the acinar component of these tumors [26]; ductal prostate cancer exhibits a high rate of copy number alterations, comparable to that observed in high-grade prostate acinar adenocarcinomas [27].

Intraductal carcinoma of the prostate (IDC-P) is an intraglandular/ductal neoplastic proliferation of prostatic glandular epithelial cells that is characterized by an expansion of glandular architecture and nuclear atypia [19]. Two typical features of these tumors are represented by the growth of atypical cells forming large dense cribriform and intraductal/acinar location of the atypical cells with preservation of basal cells [19]. It was commonly accepted that IDC-P represents invasive adenocarcinoma invading into benign prostatic duct/acinar tissue, and only in a minority of cases could represent a precursor lesion [27]. In fact, some studies suggest that IDC-Ps, as well as HGPINs, arose from, rather than gave rise to, invasive adenocarcinoma [17,28]. In line with this interpretation, Lindberg and coworkers tracked the origin of metastatic prostate cancer in a patient with prostate cancer comprising an intraductal carcinoma lesion: the analysis of breakpoint of genetic abnormalities leads to the conclusion that the IDC-P component is phylogenetically closer to lymph node metastases than most areas of an adjacent carcinoma [29]. A recent study helped to understand the possible origin of intraductal carcinomas. Taylor and coworkers have investigated prostate cancers occurring in men bearing germline *BRCA2* mutations [30]. Germline mutations in the BRCA2 tumor suppressor gene are associated with an increased lifetime risk of developing prostate cancer and increased risk of aggressive disease [29]. *BRCA2*-mutant prostate cancers display genomic instability and a mutational profile more similar to metastatic than localized disease; importantly, *BRCA2*-mutant prostate cancers show genomic and epigenomic dysregulation of MED12L/MED12 axis, frequently dysregulated in metastatic castration-resistant prostate cancer, and are clearly enriched in BRCA2-mutant prostate cancer harboring IDC [30]. Interestingly, these authors microdissected the IDC and IC components of six sporadic *BRCA2*-mutant prostate cancers bearing IDC: both IDC and IC components arose from the same founding clone, with no evidence of multiple tumors; the parental population was found both in the IDC and IC regions; *MYC* amplifications observed in 75% of these four cases and always occurred before divergence of the IDC and IC components; in contrast, the M*ED12L* gain was clonal in 50% of cases and subclonal in the other 50% of cases [30]. Also, in sporadic prostate cancers with evidence of IDC, there was no evidence of multiple independent tumors: the IDC and IC components arose from a common ancestor and there is no clear evidence as to which compartment this ancestor arose in [30]. Although the origin of IDC remains unclear, it was clearly shown that the presence of IDC, particularly with a cribriform morphology was associated with a poorer disease-specific survival and represents an independent negative prognostic factor [31]. Bottcher and coworkers have explored the genomic features of cribriform/IDC (CR/IDC) prostate cancers of patients analyzed in the context of the Cancer Genome Atlas Project (TCGA) and the Canadian Prostate cancer genome Network (CPC-GENE): CD-IDC frequency was present in 31% of TCGA and 38% of CPC-GENE datasets; CD/IDC presence was associated with deletions of 8p, 16q,10q23, 13q22, 17p13, 21q22, and amplification of 8q24; the most relevant copy number alterations affect some genes associated with aggressive prostate cancer, such as loss of *PTEN*, *CDH1*, and *BCAR1*, and gain of *MYC*; point mutations of *TP53*, *SPOP,* and *FOXA1* are also associated with CR/IDC, but occurred less frequently than copy number alterations [31]. According to these observations it was concluded that CR/IDC growth pattern is associated with genomic instability and is a histological substrate of molecular tumor progression [32].

Another study performed detailed analysis to define the molecular features of cribriform prostate cancer using the TCGA data, compared to that of GS4 non-cribriform tumors and to that of metastatic patients [32]. The results of this interesting study showed distinctive features of cribriform, compared to non-cribriform tumors: (i) increased somatic copy number alterations, such as deletions at 6q, 8q encompassing both *PTEN* and *MAP3K7* losses, and gain 3q; (ii) increased frequency of *SPOP* and *ATM* mutations; (iii) enriched gene expression pattern of mTORC1 and MYC pathways; and (iv) increased methylation of some genes [32]. The comparison with metastatic tumors, showed a higher similarity with metastatic than with non-cribriform GS4 prostate cancers [33]. Although the problem of the definition of prostate cancer precursor lesions and of their potential evolution to high-grade tumors remains an open problem, it is certainly true that some patients display tumor lesions at an initial stage of development and that these lesions may be heterogeneous, with a variable tendency to tumor progression. The use of active surveillance of “low-risk” prostate cancer is increasing and allows evaluating the potential evolution of GS6 (3 + 3) tumors to higher-grade tumors (GS7). Follow-up biopsy is the only available method to directly determine the tumor evolution and whether continued surveillance or active intervention is most appropriate. The introduction of biopsy site tracking via magnetic resonance imaging/ultrasound (NRI/US) fusion allows to sample a specific locus of tumor cells and to follow its evolution in time. The use of this approach allowed to follow in the time a specific cancer clone and to analyze its potential evolution. The initial results of these studies showed that, while many GS6 “low-grade” tumors remained stable in the time, other low-grade tumors harbor deleterious genetic alterations and may progress to higher grade disease during active surveillance [34].

In conclusion, IDCP was categorized by the WHO 2016 a distinct tumor entity and includes two different diseases with a different biological behavior: pure IDCP is a precursor lesion of prostate cancer and IDCP associated with invasive carcinoma (IDCP-iv). It is evident that IDCP-inv must be treated with radical surgery, while there is no consensus whether pure IDCP in needle biopsies should be recommended for surveillance rebiopsy or radical therapy.

## 3. Genetic Abnormalities of Prostate Cancer

### 3.1. Intertumor and Intratumor Heterogeneity

Prostate cancer is a multifocal disease since at diagnosis primary tumors contain multiple and genetically distinct foci of disease. In fact, exome sequencing of prostate cancer foci provided evidence for the presence of somatically independent tumors within the same prostate [35]. This conclusion was confirmed also in more recent studies showing the comparison of genomic landscape in both interrelated and spatially distant regions within prostates has revealed independent tumor origins [14]. It was estimated that prostate cancer is multifocal in up to 80% of men undergoing radical prostectomy for clinically localized disease [14]. Distinct tumor foci from the same tumor were subjected to whole genome sequencing showing no shared copy number alterations and very few shared punctual mutations between tumor foci, thus supporting the existence of a multiclonal disease [36]. These findings have important implications at two different levels: (a) biopsy-based diagnostic assay may miss some genetic alterations, thus leading to a misclassification of the tumor at molecular level, thus precluding optimal treatment, particularly those with new targeted agents, and (b) evaluation of the contribution of the different clones to tumor progression [36].

Using radical prostectomy specimens from patients with localized prostate cancer, several recent studies have performed genomic and transcriptomic studies aiming to evaluate the extent of intratumoral (i.e., different regions within single tumor focus) and intertumoral (i.e., different tumor foci within a single prostate) heterogeneity. A study by Wei and coworkers, based on the study of four prostate cancer patients, showed a considerable intratumoral and intertumoral heterogeneity [37]. These findings have important practical implications in the context of the proposed molecular taxonomy for prostate cancer [38]. According to this classification based on the analysis of molecular abnormalities observed in a large set of localized prostate cancers the tumor foci were classified into one of the seven molecular subgroups based on ETS gene fusion status (*ERG*, *ETV1*, *ETV4*, or *FLI1* fusions) or somatic mutations of either *SPOP1* or *FOXA1* or *IDH1* genes. Interestingly, the majority of foci could not be ascribed to any of the proposed subgroups [37]. The extension of this analysis to other studies assessing intratumoral and intertumoral heterogeneity showed that only a minority of tumor foci can be molecularly classified [37]. These findings suggest that the specific tumor foci and tumor regions sampled differentially impact risk classification. Another study based on the analysis of ten patients confirmed these findings, showing a pattern of consistent intratumor heterogeneity compatible with a branched tumor evolution, with >75% of mutations being subclonal [39]. Finally, consistent intertumoral transcriptomic heterogeneity was observed, largely reflecting a concomitant genomic and grading heterogeneity [40]. These findings were confirmed also using the recently introduced Spatial Transcriptomic method which allows for quantification of the mRNA population in the spatial context of intact tissue [41]. This methodology allowed detection of transcriptomic heterogeneity in the tumoral foci, with gene expression gradients in stroma adjacent to tumor regions [41]. These studies have sequenced bulk tumor samples, comprising at least thousands of individual cells and therefore tend to underestimate the number of subclones. To bypass these limitations, single-cell whole genome profiling of localized prostate tumors is required. Using this methodology, Su and colleagues analyzed two patients, showing consistent intercell variability in mutations: one these patients showed a classical linear evolutionary profile, while the other showed early tumor branching; thus, in the first patient, all the cells shared the same *TP53* mutation, implying a monoclonal origin, while in the second patient, only a subpopulation of cells contained the *TP53* driver mutation, while other cells carried different driver mutations, supporting a polyclonal origin of prostate cancer [42]. Another great limitation of the studies until now performed for the characterization of intratumoral genomic heterogeneity is that these studies were based on the analysis of only few prostate cancer patients. Recently, Lovf and coworkers reported high-coverage whole-exome sequencing of distinct tumor foci in 41 prostate cancer patients, showing a very high degree of interfocal heterogeneity among tumors, corresponding to 76% of pairwise-compared foci from the same prostectomy that had no point mutation in common and rarely display identical copy number changes [43]. In conclusion, studies at genomic, histopathological and molecular levels have identified tumor heterogeneity as a key biological property of prostate cancers, greatly contributing to a considerable complexity in the diagnosis, prognosis and treatment of these tumors. This consistent heterogeneity of primary prostate cancers implies a consistent vulnerabiolity of diagnosis and targeted therapy guided by the results of a single tumor biopsy limited to a single tumor area. The understanding of prostate cancer heterogeneity is essential in developing new improved diagnostic criteria, tools, and biomarkers, and in guiding the choice of ptimalized therapies.

In contrast, metastatic prostate cancer, in spite of its consistent molecular heterogeneity, at the level of the single patient is clonally homogeneous: i.e., in a single patient, different metastases are clonally related reviewed in [44]. In fact, through a high-resolution genome-wide single nucleotide and polymorphism and copy number survey it was shown that the large majority of metastatic prostate cancer have monoclonal origins and maintain a unique signature copy number pattern of the parent cancer cell, while accumulating a variable number of separate subclonally sustained changes [45]. The ensemble of these observations suggest that the prostate gland can be, at the beginning of the neoplastic process, the site of multiple neoplastic transformation events, the majority of which give rise only to latent prostate cancer that does not progress to clinically relevant disease. However, in spite this initial multifocality and heterogeneity, when the disease progresses and becomes metastatic only individual clones with selective survival and growth advantage are selected and drive tumor progression. Through this analysis, in few patients, it was possible to follow the evolution of the lethal clone from the primary tumor to metastases through samples initially collected at diagnosis, then during disease progression, and finally at the time of death. These studies showed that the lethal clone originated from a small, apparently low-grade cancer focus already present in the primary tumor, and not from the bulk high-grade tumor or from metastases [45]. These conclusions were confirmed through the whole genome sequencing of multiple metastatic tumors from 10 prostatic cancers, showing a common clonal origin involving 40–90% of total mutations and, importantly, the large majority of driver mutations [46]. After metastasis, tumor cells undergo clonal evolution and continuously change their properties through a process of metastasis-to-primary and metastasis-to-metastasis reseeding [47,48]. These tumor exchanges promote a process increasing tumor heterogeneity and competition between various clones in function of their microenvironment. Tumor heterogeneity decreases when an emergent clone has developed a high potential for local and at distance metastatic growth and is able to survive to cancer treatments [47,48]. The analysis of metastatic development allows describing phylogenetic trees of tumor development involving three different patterns: linear evolution, branched evolution, and independent evolution [47,48]. The reconstruction of the phylogenetic trees implies the assessment of the clonal relationship between subclones located at the level of different metastatic sites: truncal mutations are present in 100% in the cancer cell fraction present at two different metastatic sites; branch, nontruncal, mutations present in <100% of the cancer cell fraction at two different metastatic sites. Approximately 50% of subjects at autopsy exhibit polyclonal seeding at multiple metastatic sites, corresponding to a process where multiple genetically distinct subclones colonize a single metastatic site [47]. A recent study explored the relation between lymph node metastases and primary tumor lesions [49]. Particularly, Knoppers and coworkers compared copy number alterations of primary prostate cancer lesions with matching pelvic lymph node metastases of 30 prostate cancer patients: in 23% of these patients, the regional metastasis was not clonally linked to the index primary lesion [49]. These findings have important implications for the focal ablation therapy, which, when based on the ablation of the sole index lesion, may represent an undertreatment of a significant proportion of prostate cancer patients. In conclusion, the studies on metastatic disease suggest that the metastatic process does not uniformily originate from the index lesion, but may also originate from small, secondary non-iundex primary lesions.

Under the selection exerted by treatment with androgen receptor targeted androgen deprivation therapy, rare subpopulations of cells present in origin tumor foci that reactivate androgen receptors through a variety of molecular processes from the acquisition of mutations; copy number alterations to synthesis of constitutively active androgen receptor splice variants acquire the capacity to evade androgen deprivation therapy, while other cells acquire alterations in MYC and CTNNB1 and develop the capacity to seed and reseed multiple sites through a metastatic process [47,48,49,50]. These studies imply the potential clinical utility of performing a detailed genomic analysis at the level of multiple metastatic sites. In this context, Bova and coworkers have performed a combined analysis of whole genome sequencing and transcriptome sequence analysis of multiple prostate cancer metastases in a single patient: liver metastases displayed the presence of *AR* pL702H mutation, associated with increased expression of AR-regulated genes; the metastases displayed truncal mutation in *PIK3CG*, homozygous deletion of *TP53*, hemizygous deletion of *RB1* and *CHD1*, and amplification of *FGFR1* [51].

From a histopathological point of view prostate cancers are highly homogenous, in that the large majority corresponds to acinar adenocarcinomas, while other histotypes (such as ductal adenocarcinoma and mucinous carcinoma) are very rare. Although prostate cancer is relatively homogeneous at histological level, recent genomic profiling studies have shown a consistent degree of heterogeneity and have supported the existence of molecularly distinct subtypes.

### 3.2. Main Genetic Abnormalities in Prostate Cancer

Nonmetastatic prostate cancers have, on average, 0.7 mutations per megabase (Mb), a relatively low value if compared to that observed in other tumors, such as breast (1.2 Mb per Mb), colorectal (3.1 per Mb), or melanoma (12.1 per Mb) [52]. However, despite having relatively few mutational events, prostate cancer is characterized by a high level of genomic instability and chromosomal rearrangements.

In prostate cancer, gene abnormalities have been detected as single nucleotide variants (SNVs), small insertions or deletions, rearrangements, aberrant methylation, and changes in gene copy number. Single base pair changes occurring in prostate cancer have been explored in various studies of large-scale genomic analysis. In an initial study based on the analysis of seven high-risk primary prostate cancers, Berger and coworkers reported an average of 20 SNVs of nonsynonymous SNVs [53]. Two more recent studies have explored the presence of SNVs in 112 primary tumors and 50 metastases. Thus, Grasso and coworkers have reported an exome sequencing study in 50 metastatic, highly pretreated patients with castration-resistant disease and have shown that nine genes were frequently mutated: *TP53*, *AR*, *ZFHX3*, *RB1*, *PTEN*, *MLL2*, *CDK12*, *APC*, and *OR5L1* [54]. The last three genes were not previously reported to be mutated in prostate cancer. The ensemble of these data suggests that aberrations in AR and interacting proteins, including protein remodelers, ETS genes, and known AR coregulators are commonly mutated in prostate cancer [9]. Barbieri and coworkers have analyzed 112 primary tumors reporting a median of 30 nonsynonymous SNVs [55]. In their analysis, these authors have identified 12 genes that were recurrently mutated in primary prostate cancer: *TP53*, *PTEN*, *PIK3CA*, *SPOP*, *FOXA1*, *MED12*, *CDKN1B*, *ZNF595*, *THSD7B*, *NIPA2*, *C14orf49*, and *SCN11A* [55]. Some of these genes are involved in the androgen signaling pathway [55].

*CDH1* gene (encoding an ATP-dependent chromatin remodeling enzyme) was found to be focally deleted/mutated in 8% prostate cancers, all negative for *ETS* rearrangements [56]. A subsequent study analyzed *CDH1* abnormalities in a large number of prostate cancers, showing that 9% harbor *CDH1* deletion and 2% harbor *CDH1* mutations [56]. The frequency of *CDH1* deletions increases with tumor grade and is markedly higher among ERG fusion-negative cancers than among fusion-positive cancers [56]. Functional experiments have shown that *CDH1* expression is required for efficient recruitment of AR at the level of responsive gene promoters: this finding explains why *CDH1* deletion prevents formation of *ERG* rearrangements [56].

Kumar and coworkers have performed exome sequencing on 23 prostate cancers, 16 from lethal, metastatic tumors, and three with high-grade primary carcinomas [57]. In these patients, nonsynonymous alterations of *TP53*, *DLK2*, *GPC6*, and *SDF4* genes were detected [57]. These authors reported also a “hypermutated” phenotype in three patients with aggressive disease. Interestingly, the comparison of castration-resistant and castration-sensitive matched tumor pairs derived from the same site of origin shows that mutations in the Wnt pathway are more frequent in castration-resistant tumors and, therefore, could contribute to the development of AR resistance in prostate cancer [57]. A recent study reported a frequency of ~11.6% of hypermutated phenotype among patients with advanced prostate cancer [58]. Complex structural rearrangements in mismatch DNA repair genes *MSH2* and *MSH6* represent a major mechanism underlying hypermutation in these patients [58]. This observation is in line with the findings observed in other tumors and showing that hypermutated tumors are associated with phenotypic instability and loss of function DNA mismatch repair genes via mutation or epigenetic silencing [58].

A recent study performed whole sequencing on 11 patients with early onset prostate cancer [59]. In this group of patients an average of only 16 nonsynonymous SNVs was detected, a finding probably explained by the early disease stage of the samples analyzed [59]. In these tumor samples derived from early-onset patients they observed an overall lower number of structural rearrangements compared to those observed in patients with advanced disease; however, in these patients it was reported an increase in balanced rearrangements affecting androgen-driven genes [59]. In contrast, in patients with advanced disease, the accumulation of nonandrogen-associated structural rearrangements was observed. According to these observations it was proposed that prostate cancers at the early onset involve, in most instances, an androgen-related pathogenic mechanism which implies a pronounced abundance of balanced DNA structural abnormalities involving androgen-regulated genes [59]. More recently, the same authors have reexplored this issue in a larger study [60]. This study confirmed that EOPCs (defined as prostate cancer occurring in patients <55 years old) displayed a lower number of genetic alterations than late-onset prostate cancers (LOPC) [60]. *ETS* fusions are more frequent in EOPCs (70%) than in LOPCs (50%) [60]. After *ETS* fusions, the most frequent alterations involved *NKX3.1* and *FOXP1*, occurring in 37% and 30% of cases, respectively [60]. Recurrent genetic alterations were observed also at the level of the *KLF5* (Kruppel-like factor 5 gene) in 27% of cases and of the *ESRP1* (Epithelial Splicing Regulatory Protein 1 gene) [60]. Biallelic *PTEN* and *PT53* losses were observed in 6 and 4% of cases, respectively [60]. Interestingly, the authors of this study developed a conditional probability-based model (PRESCENT) to determine the sequence of occurrence of somatic genomic events; this approach suggested that *TMPRSS2-ERG* fusion is the initiating event, followed by *FOXP1* loss [60]. Interestingly, the PRESCENT model was able to predict disease course based on the data of a single biopsy [60].

In 2015, the prostate cancer branch of the Cancer Genome Atlas (TCGA) published a landmark study of extensive characterization (genomic, epigenomic, and proteomic) of 333 primary prostate cancers, mostly T2 and T3 cancers, ~80% of Caucasian patients [37]. This analysis showed that 74% of all patients pertain to one of seven molecular classes, based on distinct genomic drivers: *ERG* fusions (46%), *ETV1* fusions (8%), *ETV4* fusions (4%), *FLI1* fusions (1%), *SPOP* mutations (11%), FOXA1 mutations (3%), and *IDH1* mutations (1%). The four different fusions involving an *ETS* gene involve *TMPRS22* as the most frequent fusion partner and less frequently with other androgen-regulated 5’ partner genes, such as *SLC45A3* and *NDRG1* [38]. Fusions in the four genes were usually mutually exclusive, with exception of rare cases showing evidence for fusions involving more than one of these genes [38]. Tumors characterized by *SPOP* mutations are always mutually exclusive with *ETS* fusions; however, some of the *SPOP* mutated cases also possessed *FOXA1* mutations [38]. The co-occurrence of alterations in other key prostate cancer genes defined tumor subtypes: (i) *PTEN* deletions were predominant in *ERG*-fusion positive cases and (ii) SPOP-mutant prostate cancers were characterized by distinctive somatic copy number alterations (SCNAs) (such as deletion of *CHD1*, 6q and 2q): particularly, the *SPOP*-mutated/*CHD1*-deleted prostate cancer subsets have peculiar molecular features, such as elevated DNA methylation, homogeneous gene expression patterns, and frequent overexpression of SPINK1 mRNA; *FOXA1* and *SPOP*-mutated tumors display similar molecular features [37]. Approximately 26% of primary prostate cancers appear to be driven by occult molecular abnormalities or by one or more frequent alterations that co-occur with the genomically defined classes; a part of these tumors is characterized by a high burden of SCNAs or DNA hypermethylation; furthermore, these tumors were enriched for mutations in *TP53*, *KDM6A*, *KMT2D*, deletions of chromosomes 6 and 1b, and amplifications of chromosomes 8 and 11 [38]. This study showed also that 13 genes were recurrently mutated in prostate cancer, in addition to previously reported recurrent mutations: deletions of *SPOP*, *TP53*, *FOXA1*, *PTEN*, *MED12*, and *CDKN1B*; additional clinically relevant genes were identified with lower frequencies, including *BRAF*, *HRAS*, *AKT1*, *CTNNB1*, and *ATM* (Figure 1) [38]. Metastatic prostate cancer samples have more copy number alterations and mutations than primary prostate cancers; the relative distribution of the main subtypes is similar in primary and metastatic tumors; some genetic alterations, such as those involving *AR*, *ZBTB16*, *NCOR2*, *PTEN*, *PIK3CB*, *PIK3R1*, *TP53*, *RB1*, *KMT2C*, and *KMT2D* are more frequent in metastatic than primary samples (Figure 1) [38]. *AR* showed a broad spectrum of activity between genomic subtypes: *ETS* fusion-positive prostate cancers display a variable AR transcriptional activity; tumors with *SPOP* or *FOXA1* mutations had the highest AR transcriptional activity. Interesting observations of this study included that (i) ~19% of patients display clinically actionable DNA repair defects, such as *BRCA1* or *BRCA2* or *ATM* or *CDK12*, potentially indicating a sensitivity to PARP inhibitors, and (ii) ~17% of patients have clinically actionable lesions in PI3K and Ras signaling [38]. A final interesting finding of this study was related to the comparative analysis of genetic alterations reported in primary tumors and those observed in metastatic tumors, showing that (a) the spectrum of genetic alterations was similar in primary and metastatic tumors; (b) overall burden of copy number alterations and mutations was higher in the metastatic samples; (c) androgen receptor alterations in terms of amplifications or mutations are much more frequent in metastatic than in primary tumors; and (d) deletion or mutation of *PTEN*, *TP53*, *KMT2C*, *KMT2D*, *PIK3CB*, *PIK3R1*, *NCOR2*, and 2*BTB16* is significantly more frequent in metastatic than in primary tumors [38].

Although the studies of characterization of genetic alterations of prostate cancer are numerous, these studies lack of uniform pipeline analysis; to bypass this important limitation, Armenia and coworkers have reanalyzed 1013 available Wide Exome Sequencing data using a common analysis pipeline [61]. The study provided evidence that the incidence of significantly mutated genes follows a long-tail distribution, with many genes mutated in less than 3% of cases. It is important to point out that this analysis encompasses 680 primary and 333 metastatic prostate cancers. Through this approach, 20% of prostate cancers were found to display mutations in genes that encode epigenetic modifiers or chromatin remodeling genes, more frequently observed in tumors that lack an ETS fusion; recurrently mutated genes were observed in the ubiquitin protease and ligase gene family, of which SPOP is a member, with mutations found in *USP28* (1.4%), *UPS7* (1.2%), and *CUL3* (1.3%) genes; *AR*-genes are mutated in ~12% of these tumors (*AR*, 5%; *SPEN*, 2.4%; *NCOR1*, 2.5%; and *NOCR2*, 1.9%); WNT pathway was altered in 25% of samples, with predominant alterations of *PTEN* (16%) [61]. The comparative analysis of the mutational alterations in primary and metastatic tumors compared to the primary tumors is as follows; PI3K (40% vs. 17%), DNA repair (27% vs 10%), Epigenetic regulators (27% vs. 17%), Cell cycle (24% vs. 9%), WNT/CTNNB1 (19% vs. 6%) RAS/RAF/MAPK (8% vs. 4%), and Splicing (7% vs. 2%) (Figure 1) [61].

Prostate cancers are highly variable from a clinical point of view and are highly variable in their response to therapies. A group of these tumors correspond to intermediate risk prostate cancers, nonindolent and clinically heterogeneous. It is therefore very important to define the genetic factors that may contribute to the initial aggressiveness of prostate cancers. Thus, some studies have characterized the genetic abnormalities of localized, nonindolent prostate cancers: thus, Fraser and coworkers showed that these tumors were usually characterized by the paucity of clinically actionable single nucleotide variants, unlike metastatic prostate cancers; local hypermutation events are frequent in these tumors and correlated with specific genomic profiles; some molecular events were prognostic for disease recurrence, such as some DNA methylation events [62]. These patients displayed the typical CNAs observed in prostate cancer, including recurrent allelic gains of MYC and deletions of *PTEN*, *TP53*, and *NKX3.1*; the percentage of genome affected by CNAs was highly variable in these tumors [62]. Only six genes were mutated by SNVs in more than 2% of samples, including *SPOP* (8%), *TIN* (4.4%), *TP53* (3.4%), *MUC16* (2.5%), *MED12* (2.3%), and *FOXA1* (2.3%) [62]. A subsequent study analyzed the subclonal architecture of localized nonindolent prostate cancers showing that multiple subclones were observed in 55% of patients, with specific subclonal architectures associating with adverse clinicopathological features [63]. Early tumor development is characterized by point mutations and deletions, followed by later events consisting in amplifications and changes in trinucleotide mutational signatures [63]. Some genes are typically mutated before or after subclonal diversification, such as *MTOR*, *NKX3-1*, and *RB1* [63]. Specific mutational processes changed during tumor evolution, with an increasing fraction of mutations attributable to deficiency in homologous recombination repair; this finding is supported by the marked increase in *BRCA*-mutant tumors observed in metastatic lesions [63]. Reconstructing the evolutionary tumor progression trees, these tumors were classified by being monoclonal (the tumors had only clonal mutations) or polyclonal (the tumors showed evidence of multiple tumor populations originating from a single ancestral clone: biclonal in the majority of these cases, triclonal in 20% of these cases). Importantly, patients with monoclonal tumor rarely relapse (7% of cases), while those with polyclonal tumors frequently relapse (61% of cases) [63]. Aggressive polyclonal tumors are characterized by elevated genomic instability and specific mutational profiles, and these findings strongly support the assessment of tumor evolution as a biomarker to guide the delivery of precision medicine [63]. The patients with aggressive tumors may benefit from adjuvant systemic treatments, such as androgen deprivation, to reduce the risk derived from occult metastatic disease.

Another study suggested a possible link between mitochondrial mutations and prostate cancer aggressiveness [64]. Particularly, frequent agent-dependent mitochondrial mutations are observed in prostate cancer; furthermore, strong links between mitochondrial and nuclear mutational profiles were associated with clinical aggressiveness of prostate cancers [64].

A recent study reported the data of whole genome sequencing of 112 primary and metastatic prostate cancer samples [65]. From comparative analysis of previous sequencing data on more than 900 prostate cancer patients, emerged evidence for the identification of 22 new putative genes harboring coding mutations, such as truncating mutations of the *TBL1XR1* and *ZMYM3* genes that could act as prostate cancer tumor suppressors; furthermore, this study evidenced non-coding *NEAT1* and *FOXA1* mutations, acting as driver mutational events [65]. Through the temporal analysis of occurrence of aberrations, some driver mutations specifically associated with steps in the progression of prostate cancer are identified: thus, mutations in *SPOP* and *ETS* fusions occur early in cancer development and are exclusively clonal; loss of *CHD1* and *BRCA2* appear to be early events in development of ETS fusion-negative prostate cancers [65]. Tumors initiated by an *ETS* fusion event display gain of 8q (*MYC*) and loss of part of chromosome 10 harboring PTEN as very early events, while tumors that were not intiated by ETS rearrangements, show loss of chromosome 13 regions (*RB1* and *BRCA2*) as very early events [65]. Interestingly, this study through the comparative analysis of primary and metastatic cancers confirmed a higher mutational burden in metastatic than primary tumors, and provided also evidence that among metastatic subset, mutation burden was higher in men treated with androgen deprivation therapy than treatment-naïve patients; furthermore, more rearrangements in metastatic than in primary tumors were observed, whereas the proportion of breakpoints attributed to a chromoplexy-like event was similar in the two groups of patients [65].

The studies carried out in prostate cancers have shown the existence of ETS-rearranged and ETS-negative tumors. Some studies have compared the properties of ETS-positive and ETS-negative tumors. ETS-positive tumors were characterized by the presence of various ETS fusions, and display more alterations of *PTEN* and *TP53* genes; ETS-negative tumors display FOXA1 and SPOP mutations, absent in the ETS-positive group, and display a higher frequency of *CDK12*, *KDM6A*, *ROBO1*, and *ROBO2* mutations than the ETS-positive tumors (Figure 1) [65,66]. Several identical chromosome regions were amplified or deleted both in ETS-positive and ETS-negative tumors; a notable exception is related to the 3p13 region deleted in ETS-positive, but not in ETS-negative tumors (Figure 1) [65,66]. A large number of copy number gene alterations were similarly present in ETS-positive and ETS-negative tumors, but a number of these CNAs is significantly different between the two groups of tumors [65,66].

The genomic differences between primary prostate cancer and metastatic castration-resistant prostate cancer (mCRPC) were investigated in detail. The most frequent genetic alterations occurring in mCRPCs occur at the level of *AR*, *ETS* (*ETS* fusions), *TP53*, and *PTEN*; both *AR* and *GNAS* are mutated exclusively in mCRPC; *TP53* alterations are much more frequent in mCRPC than in primary cancers; there are no gene alterations exclusively observed in primary prostate cancers (Figure 2) [67]. In this study, new genomic alterations in *PIK3CA/B*, *R-Spondin*, *BRAF/RAF1*, *APC*, *β-catenin*, and *ZBTB16/PLZF* were oberserved [67]. Moreover, aberrations of *BRCA2*, *BRCA1*, and *ATM* were observed in mCRPC at clearly higher frequencies than in primary prostate cancers [67]. Importantly, this study showed that 89% of individuals with mCRPC harbor a clinically actionable aberration, including 63% with aberrations in AR, 65% in other cancer-related genes, and 8% with actionable pathogenic germline alterations (Figure 2) [67]. These observations are important because suggest differential therapeutic approaches for these patients: second-generation AR-directed therapies for mCRPC with AR pathway alterations; PI3K inhibitors for a part of patients with cancer-related gene alterations (i.e., PIK3CB-specific inhibitors for patients with alterations of this gene, MEK inhibitors for patients with RAF kinase fusions, and PARP inhibitors for patients with biallelic inactivation of BRCA2, BRCA1, or ATM). A recent study provided an accurate analysis of the structural, mutational and expression abnormalities observed in metastatic prostate cancer at genome-wide level. This extensive analysis included 101 metastatic prostate cancer patients and allowed to identify structural variants altering critical regulators of tumorigenesis and progression not detectable by exome approaches. Copy number alterations were frequent in these tumors, with a percent of the genome altered in these tumors ranging from 7% to 47% (median 23%); the median mutation frequency was 4.1 mutations/Mb, which is much higher than in primary prostate cancers [68]. Approximately 40% of these tumors were triploid. The gene loci most frequently affected by structural variation contained key driver genes of prostate cancer, including androgen receptor, the transmembrane serine protease 2 (*TMPRSS2*) and ETS transcription factor genes that produce *TMPRSS2/ERG* fusion protein, the oncogene *MYC*, *FOXA1*, and *PTEN* and clusters of deletions affecting genes located at the level of fragile sites [62]. An integrated analysis of structural variations and mRNA expression levels allowed to define cases where structural variations inactivated tumor suppressor genes: *PTEN* was affected by biallelic alterations in 36% of tumors and by monoallelic alterations in 26% of tumors; the *PTEN* sequence or promoter was affected by translocation (7% of cases) or by inversions (5% of cases); *TP53* was affected by biallelic somatic alterations in 46% of tumors and monoallelic alterations in 30% of tumors; structural variation contributed also to inactivation of *RB1*, *CDKN1B*, and *CHD1* [68]. A majority of metastatic prostatic cancers harbor fusions from the juxtaposition of the 5’ regulatory region of the androgen-responsive gene *TMPRSS2* upstream of *ERG*: mutually exclusive fusions activating the ETS family member *ERGm*, *ETV1*, *ETRV4*, and *ETRV5* in 59 of cases was observed [68]. In addition to these more classical, also rarer fusions were observed in these metastatic patients involving an ETS gene fused to various genes. Tandem duplications events were also frequent in metastatic cancers, involving: an enhancer, amplified in 87% of castration-resistant metastatic patients, that can act independently of androgen receptor locus amplification to increase expression of androgen receptor in response to androgen deprivation therapy; intergenic regions near *MYC* at 8q24 and *FOXA1* at 14q13.3 are targets of structural variation and determine tandem duplication events, contributing to increase *MYC* and *FOXA1* expression [68]. Interestingly, this study explored also the possible molecular mechanisms responsible for induction of structural variation. This analysis identified biallelic BRCA2 inactivation as strongly linked to the level of deletions, while biallelic CDK12 inactivation was associated with a significant increase in tandem duplications; furthermore, *TP53* inactivation was the event most significantly associated with inversion rearrangements and with the presence of chromothripsis [68]. The integrated analysis of somatic alterations and structural variants allowed to define a landscape of the genetic alterations observed in metastatic prostate cancers (Figure 2): (i) 85% of the tumors displayed either pathogenic activating androgen receptor mutations, amplifications of androgen receptor, or putative androgen receptor enhancer region amplifications; (ii) ETS family genes were activated by fusions in 59% of cases; (iii) *RAS/MAPK* mutations were present in 3% of cases and were mutually exclusive with ETS gene family activations; (iv) *SPOP* (5% of cases) and *CHD1* (9% of cases) were mutually exclusive with *ETS* gene family activations; (v) mutually exclusive alterations that affect genes that modulate androgen receptor pathway (*FOXA1*, *NCOR1*, *NCOR2*, and *ASXL2*) were present in 29% of cases; (vi) biallelic *BRCA2*, *CDK12*, and *ATM* inactivating mutations (all together observed in 15% of cases) were mutually exclusive; and (vii) two hypermutated cases displayed mismatch repair genes defects; alterations in WNT pathway members *CTNNB1*, *APC*, and *ZNRF3* were mutually exclusive in all but one of the 17% of cases where they were present [68].

The current view suggests that chromosomal rearrangements occur gradually over time, but recent studies suggest that in some tumors many genomic rearrangements, involving only one or few chromosomes, can occur in a one-off cellular crisis, resulting in the cancer causing multiple molecular abnormalities. Recent studies suggest that this phenomenon, known as chromotripsis, may occur in prostate cancer. Thus, in an initial study of the DNA of six patients with prostate cancers showed that in one of these patients two chromosome arms (2p and 9q) were found to harbor much more deletions than other chromosome arms [69]. In a more recent study the same authors have sequenced the genomes and transcriptomes of two prostate tumors exhibiting evidence of chromotripsis [69]. Chromotripsis is a pattern of complex chromosomal rearrangement that is affected by a number of structural variant breakpoints, usually >100, which are densely clustered in mostly one or few chromosomal arms. Through this analysis they provided evidence about the existence of multiple complex fusion transcripts, each containing sequences from three different genes, originating from different parts of the genome [69]. Evidence about the existence of poly-gene fusion transcripts was obtained also in some PC cell lines. In one tumor with chromotripsis, multiple mutations in p53 signaling pathways were observed, suggesting a link between aberrant DNA response mechanisms and chromotripsis [70]. Chromotripsis was also described in the context of progression to CRPC in patients undergoing androgen deprivation therapy: in these patients, chromotripsis may be considered as a punctuated progression to androgen independency [71]. In a more recent study Baca and coworkers sequenced the genomes of 57 prostate tumors and matched normal tissues to characterize the somatic alterations occurring during tumors progression: by modeling the genesis of the more frequent genomic rearrangements, these authors identified major DNA translocations and deletions, occurring by highly independent mechanisms [72]. Statistical analysis indicated that these complex rearrangements are unlikely to originate independently, and instead may develop from a coordinated and simultaneous molecular event. The ensemble of these complex series of genetic events was called “chemoplexy”, and seems to be responsible for the coordinated dysregulation of many prostate cancer genes [72]. Therefore, chromoplexy seems to be responsible for a considerable genomic derangement, in consequence of few genetic events [72]. Chromoplexy is another pattern of complex rearrangements that has many interdependent structural variant breakpoints (interchromosomal translocations), but usually fewer than chromotripsis. Basically, chromoplexy is an extended version of balanced translocations that reshuffles multiple chromosomes, rather than two chromosomes, as in balanced translocations. Chromoplexy mechanisms frequently disrupt tumor suppressor genes and activate oncogenes by the formation of fusion genes (i.e., *TMPRSS2-ERG*). The prevalence of chromoplexy in prostate cancer is ~90% [72]. This study proposes also the existence of at least two different molecular subtypes of prostate cancer: one characterized by the presence of ERG rearrangements and features of chromoplexy and the other one characterized by the absence of *ERG* rearrangements and *CHD1* deletions, exhibiting intrachromosomal rearrangements and features of chromotripsis [72]. Importantly, this study had led also to propose, through analysis of the clonality of genomic events a tumor’s natural history with ERG rearrangements, *NKX3-1* deletion, *SPOP*, and *FOXA1* mutations as clonal events, occurring early during the natural history of prostate cancer; these events are followed by genetic alterations at the level of *TP53* and *CKN1B* and, finally, by inactivation of *PTEN* [72].

### 3.3. Genetic Abnormalities of Metastatic Disease

Three diffent mechanisms may undeline the metastatic process and may differentially originate metastasis heterogeneity: (a) the original clone seeds all metastases, and therefore all metastases share some founding driver mutations; (b) a single highly metastatic subclone evolves and gives rise to all metastases; and (c) a new subclone with an additional driver mutation evolves and seed metastases. It is important to note that prostate cancer exhibits a substantial level of intratumor heterogeneity in unifocal tumors on multiregional biopsies, as evidenced by the abundance of private or region-specific mutations at the level of tumor foci within each tumor [74]. This finding emphasizes the necessity to have a view of different tumor areas to obtain complete information about the whole complexity of the genomic alterations of a single prostate cancer [74]. However, a different conclusion was reached through the analysis of multiple tumors from men with metastatic prostatic cancer through various molecular genomic techniques and on the comparison of the genomic diversity within and between individuals [73]. The number of somatic mutations, the burden of genomic copy number alterations, and various types of aberrations in known oncogenic drivers are concordant, as well as cell cycling activity [67]. According to these findings, the conclusion was reached that the majority of patients, the evaluation of a single metastasis allows an acceptable assessment of the major driver oncogenic alterations [73]. Interestingly, this study provided also evidence that prostate cancer patients with aberrations in Fanconi anemia-complex genes or in ATM serine/threonine kinase displayed markedly longer treatment responses to carboplatin than did patients without defects in genes encoding DNA repair proteins [73]. The analysis of various solid tumors showed minimal gene heterogeneity among untreated metastases [75]. Different mechanisms contribute to limit the intermetastatic heterogeneity: (i) driver mutations may not confer the same advantage in the microenvironment of the primary tumor and of a distant metastatic site, thus reducing the chances of heterogeneity; (ii) the primary tumor may reduce its growth rate because of nutrient constraints or surgical resection, thus reducing the intermetastatic heterogeneity; and (iii) advanced cancer cells have already acquired multiple driver mutations, thus reducing the number of additional driver mutations that may confer a substantial selective advantage [75]. In spite of limited functional driver heterogeneity among the metastases of prostate cancer patients, some recent studies suggest the existence of some metastasis heterogeneity. Thus, Nava Rodrigues and coworkers reported a high fraction of genes with concordant copy number status across metastases from the same patient (on average 0.93); however, heterogeneity was observed in some patients, related to the presence of private events [76]. Thus, aberrations of the WNT signaling pathway were seem as private mutational events in two patients; in additional two patients, heterogeneous *RB1* alterations were identified between metastases [76]. In another study, Iglesias-Gato and coworkers have reported that, compared with primary tumors, bone metastases were more heterogeneous and showed increased levels of proteins involved in cell cycle response, DNA damage response, RNA processing and fatty acid beta-oxidation, but reduced levels of cell adhesion-related proteins, and carbohydrate metabolism [77]. Two phenotypic subgroups of bone metastasis were identified: BM1, expressing higher levels of androgen receptor targets, mitochondrial, and Golgi apparatus-resistant proteins, and BM2, expressing increased levels of proliferation and DNA repair-related proteins [77]. BM1-expressing prostate cancers might be sensitive to drugs targeting metabolic function, in combination with AR targeting drugs [77].

As mentioned above, the mutational index of prostate cancer is relatively low. In contrast, the frequency of large-scale copy number alterations (CNAs) and genomic rearrangements is significantly higher, thus suggesting that the development and progression of prostate cancer is more seemingly related to the accumulation of genomic aberrations, such as gains, deletions and fusion gene events than more localized mutational events. The analysis of CNAs in prostate cancer showed a total of 14 regions of recurrent deletion and five regions of recurrent gain (Table 1) [78,79]. Among the deletions, deletion of chromosome 8p was the most recurrent CNA observed in the prostate cancer being observed in ~62% of these tumors (55% in localized tumors and 90% in advanced tumors); this chromosome region contains the gene encoding the prostate-specific tumor suppressor *NKX3-1* [78]. The second most frequent deletion consisted in the deletion at the level of the 13q chromosome region, containing the tumor suppressor *RB1*; this deletion is observed in ~53% of prostate cancers (45% of primary tumors and 90% of advanced tumors) [78]. The third most common deletion is at the level of the 16q region. Concerning the chromosome gains, chromosome 8q gain was identified in ~21% of localized tumors and in ~84% of advanced cases; in some of these cases, particularly in advanced tumors, there is a small focal region of high gain at 8q24.21, which corresponds to the MYC oncogene [78]. Frequent chromosome gains are observed also at the level of chromosome 7 and 16p arm; it is important to note that chromosome 7 gain was much more frequent in advanced tumors (64%) than in primary tumors (14%) [78]. Importantly, the AR locus, present on chromosome X, is frequently interested in chromosome gains in advanced tumors (~66%), but only rarely in primary tumors (~3%) [78]. According to the presence of specific CNAs, the primary prostate cancers were subdivided into three groups: A, lacking any CNA; B, lacking 8p deletions; and C, the most frequent (~80%), with Bp (base pair) deletions and a wide range of CNAs [78]. In the study of CNAs of prostate cancer, particular emphasis is given to *PTEN* deletions, often occurring in concomitance with ETS gene fusions; *PTEN* deletions were observed in ~23% of localized tumors and in ~70% of advanced tumors [78]. *PTEN* deletions were present as both hemizygous and homozygous deletions; the homozygous deletions being much more frequent in advanced than localized prostate tumors [78]. The frequency of concomitant *ETS* fusions and *PTEN* loss was much higher in advanced (42%) than in localized (9%) tumors [78]. Recent studies have shown that the total level of copy number alterations present in the genome of a prostatic cancer is prognostic for cancer recurrence and metastasis. In an initial study it was observed that the pattern of CNAs in prostate tumors at prostatectomy was associated with biochemical recurrence [80]. These findings were corroborated by a second study carried out by the same group of authors: this study was based on the analysis of 104 primary prostate cancers and included also the updating of the initial cohort of 168 patients [81]. The results of this study clearly showed that the total CNA burden, defined as the percentage of the tumor genome affected by CNAs was associated with biochemical recurrence and metastasis after surgery, independent of PSA levels or Gleason grade [81]. Interestingly, copy number alteration is a prognostic factor also for many other solid tumors, associated with recurrence and death, as recently shown [82].

Other recent data confirmed that the burden of somatic copy number alterations was predictive of biochemical recurrence and defined nine individual regions that are associated with relapse and highlighted the possible importance of ion channel and G-protein-coupled receptor pathways in cancer development [83]. Importantly, this study explored the possible oncogenetic mechanisms on CNAs in prostate cancer using whole genome sequencing approach [83]. This study explored at what extent the CNAs occurring in prostate cancer follow a classical two hit genetic model of cancer development based on the assumption that mutations or CNAs are required in each of the copies of a single gene [84] or alternative models supporting a role also for hemizygous focal copy number alterations collectively contributing to cancer development [85]. This study provided evidence that 64 recurrent regions of loss or gain were detected, including some regions of loss with more than 15% of frequency at Chr 4p15.2–p15.1 (15.5%), Chr 6q27 (16.5%), and Chr 18q12.3 (17.5%) [83]. Importantly, a two-hit genetic model accounts for approximately one-third of CNAs, indicating that other mechanisms, such as haploinsufficiency and epigenetic inactivation, account for the remaining cases of CNAs [83]. Recurrent breakpoints and regions of inversions frequently occur within the Knudson two-hit model of CNAs, leading to the identification of ZNF292 as a target gene for deletion at 6q14.3–q15 and NKX3.1 as a two-hit target at 8q21.3–p21.2 [83]. According to these observations it was concluded that a two-hit genetic model accounts for about one third of copy number alterations, suggesting that mechanisms such as haploinsufficiency and epigenetic inactivation account for the remaining copy number alteration losses [83]. Copy number variations in regions encompassing important prostate cancer genes, such as *PTEN* and *CHD1* or *ASAP1*, *MYC*, and *HDAC9*, are predictive of cancer significance and represent useful biomarkers to distinguish low-risk prostate cancer from intermediate- and high-risk prostate cancer [86].

Another frequent focal gene deletion occurring in prostate cancer is represented by a consensus deletion of a 800 Kb locus present on chromosome 6q15.1: the *MAPK3K7* gene, encoding TGF-beta kinase 1 (TAK1), maps in this chromosome region [77]. In various experimental models, including murine prostate stem cells, *TAK1* loss promotes prostate tumorigenesis [87]. *TAK1* deletions were more frequently (~27%) observed among *ERG* rearrangement-negative tumors than among *ERG*-rearranged (~11%) tumors [88]. TAK1 deletion was associated in both ERG-rearranged and not ERG-rearranged groups with early tumor recurrence [88]. Another quantitative abnormality frequently observed among non-rearranged ERG prostate cancers (11%) is represented by SPINK1 protein overexpression [89]. SPINK1 protein overexpression does not seem to be a predictor of recurrence or lethal prostate cancer amongst patients treated with radical prostatectomy [89]. A recent study explored the intratumor heterogeneity of CNAs and of DNA methylation abnormalities in advanced prostate cancer. Aberrant DNA methylation patterns are concomitantly found in prostate tumors and frequently affect genes involved in cell cycle control, hormonal response, and DNA damage repair [90]. This study was based on the analysis of multiple topographically distinct tumor sites, premalignant lesions, and lymph node metastases within five cases of prostate cancer, and showed (i) the presence of shared methylation patterns and chromosomal breakpoint profiles between all tumor regions of a given patient supported a monoclonal origin in all these five patients; (ii) the analysis of the various tumor regions showed the presence of multiple subclonal cell populations, characterized by different copy number as well as DNA methylation profiles; and (iii) copy number losses and DNA hypermethylation events are more clonal than copy number gains and DNA hypomethylation (clonal deletions or hypermethylation events included known tumor suppressor genes, such as *PTEN*, *TP53*, or *GSTP1*) [90]. These data supported the existence of an extensive spatial DNA methylation and copy number heterogeneity in prostate cancers of monoclonal origin [80]. Importantly, this study showed also a high epigenetic heterogeneity at androgen receptor-bound enhancer domains [90].

### 3.4. Genetic Abnormalities in Neuroendocrine Prostate Cancer

In a minority of patients, therapeutic resistance to androgen receptor deprivation therapy is associated with the emergence of a peculiar histologic subtype termed small cell neuroendocrine (t-SCNC) prostate cancer: a highly aggressive prostate cancer subtype observed in <1% of de novo prostate cancers [91]. Neuroendocrine prostate cancer is a lethal form of the disease, characterized by loss of AR signaling during transdifferentiation, which results in resistance to AR-targeted therapy. In a recent study, Aggarwal and coworkers evaluated 148 prostate cancer patients in progression under abiraterone and/or enzalutamide, showing in 17% of these patients the t-SCNC variant [92]. These highly aggressive and lethal tumors display reactivation of developmental programmes associated with epithelial–mesenchymal plasticity and acquisition of stem cell-like properties. AR amplification and protein expression were observed in 67% and 75%, respectively, of t-SCNC biopsy specimens; t-SCNC was observed at the level of various metastatic sites [92]. *TP53* and *RB1* alterations were more frequent among t-SCNC tumors (85%) than in those without this histology (34%) [92]. The detection of alterations of genes involved in DNA repair was rare in t-SNC tumors (8%), compared to that observed in tumors without this histology (40%). Detection of t-SCNC *SMAD4* and *BCL2* was associated with shortened overall survival [92]. Beltran and coworkers have investigated a group of t-SCNC metastatic prostatic cancers and showed that *RB1* loss was more frequent in t-SCNC (70%) than in Adeno (32%) CRPCs; *TP53* was more frequently mutated in t-SCNC (66%) than in Adeno (31%) CRPC samples; *AR* point mutations were absent in t-SNCN samples and AR signaling is usually attenuated in these tumors [93]. Analysis of biopsy samples from the same individuals over time has led to propose a model of t-SNCN genesis based on divergent evolution rather than by linear or independent clonal evolution (therefore, t-SNCNs do not pre-exist in the parental tumors as a very minoritary subpopulation), with selective pressure of subclonal populations with wild type *AR* and the acquisition new genomic and epigenomic drivers associated with decreased AR signaling and epithelial plasticity [93]. In addition to the loss of tumor suppressor genes *TP53* and *RB1*, the gain of *MYCN* and *AURKA* oncogenes represent other key genetic alterations associated with the development of neuroendocrine prostate cancers [94]. These genetic changes converge on biochemical pathways upregulating *SOX2* and *EZH2* expression, thus facilitating lineage plasticity and neuroendocrine differentiation [94]. In line with these findings, *N-MYC* overexpression in multiple preclinical models drives prostate cancer that, at molecular level, resembles clinical neuroendocrine prostate cancers and sensitizes to the Aurora kinase and EZH2 inhibitors [95].

Given these findings, a recent phase II clinical trial evaluated an Aurora kinase inhibitor, Alisertib, in neuroendocrine prostate cancer patients; this drug inhibits the interaction between N-MYC and its stabilizing factor Aurora-A [96]. However, only a minority of these patients responded to this treatment, including some exceptional responders [96].

Other recent studies have identified additional driver genetic events inducing neuroendocrine trans-differentiation. The tumors are characterized by neuroendocrine differentiation and enhanced angiogenesis: both these events are induced by androgen deprivation therapy activated CREB (c-AMP response-element binding protein) that in turn enhances EZH2 activity [97]. CREB inhibition reduces the growth of neuroendocrine tumors [97]. Through analysis of differentiated neuroendocrine tumors, ONECUT2 was identified as a master transcriptional regulator of poorly differentiated neuroendocrine prostate cancers; ONECUT2 ectopic expression synergizes with hypoxia to suppress androgen signaling and to induce neuroendocrine trans-differentiation [98]. Particularly, ONECUT2 is overexpressed in poorly differentiated neuroendocrine prostate tumors, and its expression increases with tumor progression; in tumor cells, ONECUT2 acts as a regulator of hypoxia signaling and regulates HIF-1α binding to chromatin through SMAD3 activation [98].

Another recent study provided evidence that protein kinase C (PKC)λ/ι is downregulated in both de novo and during therapy induced neuroendocrine prostate cancers, which results in the upregulation of serine biosynthesis via an mTORC1/ATF4-driven pathway [99]. This metabolic reprogramming is required to sustain the proliferation of neuroendocrine prostate cancer cells and determines an increase of S-adenosyl methionine levels, an event involved in epigenetic changes favoring neuroendocrine cell differentiation [99]. This finding shows the existence of a metabolic vulnerability of neuroiendocrine pprostate cancer cells [99].

A recent study provided clear evidence that the combination of five oncogenic drivers (dominant negative TP53, myrostoylated AKT1 mimicking PTEN loss, c-Myc or N-Myc overexpression, RB1 short hairpin RNA, and BCL2 overexpression) induced the reprogramming of normal human prostatic epithelial tissues to a common, lethal neuroendocrine prostate cancer [100].

## 4. Most Recurrent Genetic Abnormalities Observed in Prostate Cancer

Some frequent molecular events occurring in the majority of prostate cancers have been characterized in the last years and are here briefly analyzed.

### 4.1. TMRSS2-ERG

In a high proportion of prostate cancers chromosomal rearrangements activate members of the ETS family of transcription factors, such as ERG. The most frequent of these rearrangements create a *TMPRSS2-ERG* fusion gene, observed in ~15% of prostate intraepithelial neoplasia (PIN) and in ~50% of localized prostate cancer: this observation suggests that this genetic anomaly may represent an early event predisposing to tumor progression [101]. The frequency of *TMPRSS-ERG* fusion significantly varies in different ethnic groups: Caucasian (50%), African American (30%), and Asian (20%). In line with this conclusion, a recent study showed that the *TMPRSS2-ERG* fusion was present in 11% of high-grade prostatic intraepithelial neoplasia (HGPIN): patients positive for *TMPRSS2-ERG* expression in HGPIN have a higher probability of progression to prostate cancer than TMPRSS2-ERG-negative patients [102]. The fusion TMPRSS2-ERG was occasionally detected in advanced cancers, not initially carrying a diagnosis of prostate carcinoma; Lara and coworkers reported that *TMPRSS2-ERG* fusions were identified for 0.86% (250/29,030) of male cancer patients, including 30% of prostate cancer patients and six tumors classified as squamous carcinoma, without evidence of prostate cancer [103]. Interestingly, *TMPRSS2-ERG*-positive tumors exhibit some peculiar properties related to androgen metabolism; in fact, patients bearing *TMPRS2-ERG* tumors have different androgen profiles compared to *TMPRSS2-ERG*-negative patients, consisting of enhanced androgen-regulated gene expression and altered intratumoral androgen metabolism, demonstrated by reduced testosterone concentrations and increased dihydrotestosterone (DHT)/testosterone ratios [104]. Therefore, patients with *TMPRSS2-ERG*-positive prostate cancer could benefit from novel inhibitors targeting the alternative DHT biosynthesis.

The *TMPRSS2* gene encodes an androgen-regulated, type II transmembrane-bound serine protease that is highly expressed in normal and neoplastic prostatic tissue. The formation of this fusion gene determines the expression of the N-terminally truncated ERG protein under the control of the androgen responsive promoter of *TMPRSS2* (transmembrane serine protease isoform 2). Through the generation of mouse models, it was possible to demonstrate that this protease regulates cancer cell invasion and metastasis to distant organs through activation of the Hepatocyte Growth Factor/c-met axis [105]. Interestingly, a TMPRSS2 inhibitor suppressed prostate cancer metastasis. A recent study showed that androgen signaling promotes corecruitment of androgen receptor and topoisomerase II beta (TOP2B) to sites of *TMPRSS2* breakpoints, triggering TOP2B-mediated double-strand breaks [106]. This important observation indicates that androgen receptor activation triggers TMPRSS23-ERG rearrangement. In line with these findings, *TMPRSS2-ERG* fusions are frequent among young patients with prostate cancer, suggesting that this condition could be caused by increased androgen signaling in younger men [105]. With age, the frequency of TMPRSS2-ERG fusions decreases, however only in low-grade cancers [107]. Another study confirmed the very high prevalence of *ETS* fusions in the early-onset prostate cancers, defined as prostate cancers diagnosed in patients under 50 years of age [59,60]. It is important to note that early-onset patients are characterized by higher expression of *AR* and about 90% of these patients had *ERG* fusions and deletions of *AR* corepressor *NCOR*, which is significantly higher than the estimated 50% for all prostate cancers [59,60]. According to these findings it was suggested that AR signaling consistently increases the probability of certain DNA rearrangements, such as those involving ERG or other ETS transcription factors and androgen responsive elements in *TMPRSS2* [59,60]. It is of interest to note that the early-onset prostate cancer tumors exhibited structural rearrangement breakpoints situated nearer to AR binding sites than those in elderly onset prostate cancers [59,60].

Transgenic *TMPRSS2-ERG* mice develop prostate intraepithelial neoplasia, but only in the context of PI3K pathway activation [108]. Three chromosomal regions of recurrent copy number loss associated with the *TMPRSS2-ERG* fusion: two regions spanning the tumor suppressors *PTEN* and *TP53*, respectively, and a third spanning the multigenic region of at 3p14. In line with this observation, prostate cancer specimens containing the *TMPRSS2-ERG* rearrangement are significantly enriched for loss of the tumor suppressor *PTEN*. In line with this finding, transgenic overexpression of ERG into mouse prostate tissue promoted marked acceleration and progression of high-grade prostatic intraepithelial neoplasia to prostatic adenocarcinoma in *PTEN* heterozygous background [109]. According to the ensemble of these observations, it was concluded that ERG activation, induced by the *TMPRSS2-ERG* rearrangement, has an important role in prostate cancer progression and cooperates with PTEN haploinsufficiency to promote prostate cancer progression [109]. It is important to note that the *TMPRSS2-ERG* fusion is correlated with aggressive prostate cancer and poor prognosis. *TMPRSS2-ERG* fusion protein activates a transcriptional program that contributes to prostate oncogenesis through upregulation of the expression of some key genes including *MYC*, *EZH2* and *SOX9* and repression of *NKX3* expression. A recent study has provided evidence that the *TMPRSS2-ERG* fusion protein blocks neuroendocrine and luminal cell differentiation to maintain prostate cancer proliferation [110].

In prostate cancer cells, ERG recruits the AR at the level of novel genetic loci and interacts with other transcription factors at the level of AR binding sites, and, through these effects, modifies the transcriptional activity induced by androgen signaling. Interestingly, at the reprogrammed AR binding sites in human prostate cancer cells it was reported the colocalized binding of FOXA1 and HOXB13—two prostate master transcription factors [111]. A recent study showed the existence of a link between *TMPRSS2-ERG* fusion and *HOXB3* and *FOXA1*: particularly, it was shown that TMPRSS2-ERG co-opts *HOXB3* and *FOXA1*, thus modifying the AR cistrome (the AR binding sites at the level of genome) [112]. These authors discovered also the existence of a T*MPRSS2-ERG*-specific CORE (Cluster of Regulatory Element, super enhancer elements controlling gene transcription) on the structurally rearranged ERG locus [112]. Finally, it was provided evidence that TMPRSS2-ERG activates NOTCH signaling, thus inducing a druggable dependency on NOTCH signaling in TMPRSS2-ERG-positive prostate cancers [112]. The role of TMPRSS2-ERG as an oncogenetic fusion protein modifying gene expression at transcriptional level is supported also by a recent study showing that TMPRSS2-ERG drives genome-wide retargeting of BAF, a SWI/SNF ATP-dependent chromatin remodeling complex regulating gene transcription [95]. In prostate organoid models, BAF complexes are required for ERG-mediated basal-to-luminal transition, a typical feature of ERG activity in prostate cancer [113].

Studies in mice *PTEN/TP53*-mutated/ERG-overexpressing allowed defining a blocking effect of ERG overexpression on *TP53/PTEN* alteration-induced decrease of AR expression and downstream luminal epithelial genes [114]. Particularly, ERG suppressed the expression of cell cycle-regulated genes, with consequent RB hypophosphorylation and repression of E2F1-mediated expression of mesenchymal regulators, thus maintaining antiandrogen sensitivity and restricting adenocarcinoma plasticity [114]. These findings suggest that ERG fusion represents a potential biomarker to guide treatment of *PTEN/TP53*-altered, *RB1*-intact prostate cancers [114].

Oncogenic activation of ERG represents an early prostate cancer driver event and is, therefore, an appropriate therapeutic target to attempt an early eradication of this neoplasia. Therefore, blocking ERG expression/activity may represent a useful therapeutic strategy in tumors harboring *TMPRSS2-ERG* fusion. A recent study reported that 1-[2-Thiazolylazo]-2-naphtol (called ERGi-USU) acts as an ERG inhibitor, inhibiting the growth of ERG-positive cancer cell lines [115]. This compound has potential for further development of ERG-targeted therapy of prostate cancer [115].

Other frequent chromosomal rearrangements involve other genes of the ETS transcription family, such as ETV1. In the case of *ETV1* gene the 5′ fusion partners are more heterogeneous, being *TMPRSS2*, *SCL45A3*, and *ACSL3*. These fusion partners are androgen-responsive genes. A recent study in part clarified the mechanism through which *ETV1* could promote prostate cancerogenesis. In fact, it was shown that *ETV1*, but not *ERG*, upregulates the expression of AR target genes and of the genes involved in steroid biosynthesis and metabolism. These molecular events activate an oncogenic program, predisposing prostate cells for cooperation with other oncogenic events, such as *PTEN* loss, leading to more aggressive tumor development in animal models, as well as in human patients [116]. In fact, patients with high *ETV1* frequently had a metastatic disease, and are associated with a poor prognosis; furthermore, patients with high *ETV1* expression and loss of *PTEN* displayed a much poorer disease-free survival [116].

The cooperation between *EGR* and *ETV1* with *PTEN* loss in promoting prostate tumorigenesis was supported also by another recent study [117]. In the *PTEN* loss setting, *ERG* overexpression promoted the restoration of *AR* transcriptional output and upregulation of genes involved in the control of cell death, migration, inflammation, and angiogenesis, while *ETV1* overexpression upregulated AR cistrome and transcriptional output [117]. According to these findings it was proposed that ETS transcription factors cause prostate-specific transformation by altering the AR cistrome, inducing the prostate epithelium to respond to aberrant signaling signals, as those deriving from the *PTEN* loss [117]. A recent study, through the analysis of individual tumoral foci, analyzed the chronological relationship existing between *ERG* fusions and *PTEN* deletion. This analysis showed that PTEN deletions, when present were usually heterogeneously expressed in the large majority of tumor foci (92%), while only 8% of tumor foci were homogenously aberrant [118]. Importantly, the observation that the large majority of foci with homogenous *ERG* rearrangements had focal *PTEN* alterations, but none of the foci with homogeneous *PTEN* alterations had focal ERG positivity, strongly suggests that *PTEN* alterations usually develop after ERG fusions [119]. It is important to note that prostate cancer exhibiting *PTEN* loss concomitantly displays *TMPRSS2-ERG* fusion; in contrast, not all of the *TMPSS2-ERG* fusion-positive tumors show *PTEN* deletion [119].

Interestingly, 2–4% of prostate cancers display mutational alteration of the *ERF* gene and ETS transcriptional repressor [120]. ERF mutations cause decreased protein stability and occur in prostate cancers without ERG upregulation [120]. ERF loss in prostate cells recapitulates the biologic effects induced by *ERG* gain: activation of androgen-dependent gene expression and induction of tumor formation in cooperation with *PTEN* loss [120]. Thus, loss of *ERF* activity by rare genomic loss-of-function mutations or by competition with the *TMPRSS2-ERG* oncogenic product leads to activation of the androgen receptor pathway and prostate cancer [79]. However, the fact that ERG translocations are much more frequent than ERF mutations suggests that ERG may have additional gain-of-function activities that promote its oncogenic capacity [79].

The initial therapy for advanced prostate cancer involves androgen ablation that leads to a reduction in *TMPRSS2-ERG* expression in tumors expressing this fusion protein; however, patients nearly invariably progress with development of mechanisms of androgen resistance and restore of *TMPRSS2-ERG* expression [121]. The TMPRSS2-EERG fusion protein represents an attractive therapeutic target since it is a key oncogenic driver for some prostate cancers sensitive and resistant to androgen deprivation. Furthermore, growing evidences suggest that *TMPRSS2-ERG*-positive tumors display some peculiar molecular and biological properties that could offer the way to new therapeutic approaches. In the *TMPRSS2-ERG* fusion the transcription factor ERG is rearranged, and this suggests that targeting *ERG* activity may have a great therapeutic relevance. Direct targeting of transcription factors has proven to be a tremendous challenge and these molecules have been considered “undruggable”. A recent study by Wang et al. reported the identification of ERG inhibitory peptides (EIPs), specifically interacting with the DNA-binding domain of ERG; as a consequence, EIPs block *ERG*-mediated transcription and recruitment to target genes and determine proteolytic degradation of ERG protein [122]. Importantly, EIPs reduce proliferation, invasion, and tumor growth of prostate tumors bearing the *TMPRSS2-ERG* gene fusion [122].

Other studies have attempted to define peculiar biologic properties of *TMPRSS2-ERG*-positive prostate cancers. A recent study showed that NOTCH factors are direct transcriptional targets of ERG; inhibition of ERG in *TMPRSS2-ERG*-positive prostate cancer cells and decreased NOTCH1 and NOTCH2 levels [123]. Importantly, treatment of tumor cells with a NOTCH γ-secretase inhibitor conferred an increased sensitivity to androgen receptor inhibitors [123]. According to these observations it was suggested that combinatorial targeting of NOTCH and AR signaling may have a therapeutic potential in advanced prostate cancers bearing ERG rearrangements [123]. As discussed above, several recent studies have provided evidence about some notable molecular differences between prostate cancers exhibiting or not *ERG* rearrangements. Recently Bratsalaksky et al. investigated the genomic signatures observed in a large group of castration-resistant prostate cancers (2424 CRPCs and 143 CRNEPCs) subdivided into *TMPRSS2-ERG*-positive and *TMPRSS2-ERG*-negative: TMPRSS2^+^ tumors displayed greater *TP53* and *PTEN* genomic alterations, while TMPRSS2-tumors showed higher *MYC* and *ATM* genomic alterations; differences in BRCA2 and RB1 were not significant between these two different groups [124].

### 4.2. SPOP Mutations

Somatic heterozygous missense mutations occurring at the level of the substrate-binding cleft of speckle-type PO2 protein (SPOP) gene were identified in up to 15% of human prostate cancers, thus making SPOP the gene most commonly affected by nonsynonymous point mutation in this cancer [125]. Furthermore, SPOP protein expression is often downregulated in prostate tumors. SPOP is a tumor suppressor protein and substrate adaptor of the cullin 3-RING-ubiquitin ligase (CUL3); tumor-associated *SPOP* mutations disrupt substrate binding and ubiquitination, leading to increased expression of oncogenic substrates. The SPOP protein was able to interact with the p160 steroid receptor coactivators (SRC-3), playing a key role in the control of AR activity. The normal SPOP protein, but not the mutants observed in prostate cancer, is able to interact with SRC-3 and promote its ubiquitination and degradation [125]. According to these observations it was concluded that the SPOP protein plays a key tumor suppressor role in prostate cancer cells, but this effect is lost by the prostate cancer-associated SPOP mutants [125]. It is of interest to note that the presence of a *SPOP* mutation was mutually exclusive with mutations in *TP53*, *PTEN*, or the *TMPRSS2-ERG* rearrangement. As above reported, the SPOP gene is frequently mutated in prostate cancer (6–15% of cases). Interestingly, *SPOP* mutant prostate cancers lacked *EGR* rearrangements and exhibited a peculiar pattern of genomic alterations, thus suggesting that they form a molecular subtype of prostate cancer [8]. Many recent studies have shown a link between SPOP and AR pathway. SPOP WT Protein promotes ubiquitination and degradation of several protein substrates, including the AR coactivator SRC-3; *SPOP* mutations are usually missense mutations in the substrate-binding pocket and then inactivate the SPOP protein with the consequent loss of degradation of the AR coactivator SRC-3 [126] and of its inhibitory effect on AR signaling [125]. In line with this finding, tumor xenograft expressing mutated SPOP display elevated AR levels [125]. AR is a direct target of SPOP: in fact, this protein recognizes a Set/Thr-rich domain with degron activity, present in the hinge domain of the AR, inducing degradation of AR and inhibition of AR-mediated transcription [127]. *SPOP* mutants are unable to recognize the AR and to promote its degradation [127]. Androgens antagonize the SPOP-mediated degradation of AR, while antiandrogens promote this degradation, thus favoring the activity of SPOP [127]. Recent studies suggested an important role of other targets of SPOP as mediators of the oncogenic effects of the mutated *SPOP*. In fact, Geng and coworkers showed that SPOP-WT can physically interact with c-MYC protein and can promote c-MYC ubiquitination and degradation; this function is attenuated in SPOP-mutants, thus promoting c-MYC accumulation and promotion of prostate cancer cell proliferation [128]. Theurillat and coworkers showed that DEK and TRIM24 are effector SPOP substrates, upregulated in *SPOP*-mutant cells [129]. DEK stabilization was shown to be able to induce prostate epithelial cell invasion and then seems to be an important effector of *SPOP* mutants [130]. Recent studies have clarified the role of TRIM24 as an important effector substrate of SPOP. TRIM24 becomes stabilized in prostate cancer with SPOP mutations, through mechanisms involving TRIM28, upregulated in aggressive prostate cancer and associated with elevated TRIM24 levels: TRIM28 interacts with TRIM24 to prevent its ubiquitination and degradation by SPOP [131]. TRIM24 protein augments AR signaling and promotes prostate cancer proliferation under low androgen conditions [132]. TRIM24 levels increase during prostate cancer progression, and the AR/TRIM24 gene signature as well as TRIM24 levels predict disease recurrence [132].

Analysis of a large number (>8000) prostate cancer patients allowed defining the clinicopathologic features of SPOP-mutant tumors: lower frequency of positive margins, extra prostatic extension, and seminal vesicle invasion at prostatectomy; higher pretreatment serum PSA levels [133]. Despite high PSA pretreatment values, the *SPOP*-mutant prostate cancers have a favorable prognosis with improved metastasis-free survival, especially in patients exhibiting high PSA pretreatment levels [133]. Interestingly, this observation supports the view that a common risk stratification parameter, such as PSA, is influenced by underlying molecular abnormality [133].

Recent studies have better clarified the potential role of *SPOP* mutations to prostate cancer pathogenesis. Thus, the analysis of primary prostate cancer samples, as well as studies on tumor organoids, provided no evidence that *ERG* is an effector of *SPOP* mutation in human prostate cancer and indicates that *SPOP* mutations activate an oncogenic program leading to prostate cancer independently on ERG stabilization/activation [134]. Using animal models of prostate cancer, as well as primary cancer samples, it was shown that SPOP mutation activates both PI3K/mTOR and androgen receptor signaling, efficiently uncoupling the normal feedback occurring between these two pathways and through this mechanism promoting prostate cancer development [135]. Finally, other recent studies clarified a spectrum of therapeutic sensitivity of these tumors. In fact, SPOP-mutated metastatic prostate cancers are strongly enriched for *CDH1* loss; these tumors appear to be highly sensitive to Abiraterone treatment [136]. However, *SPOP* mutations act as a negative regulator of BET protein stability and, through this mechanism, confer resistance to BET inhibitors [137]. Two recent studies have explored the mechanisms through which SPOP mutations promote stemness features in prostate cancer cells [138,139]. In fact, both these studies showed that WT *SPOP* suppresses stem cell tracts promoting Nanog poly-ubiquitination; *SPOP* mutations determine a loss of this important activity, increasing Nanog levels and determining increased cancer stem cell traits of prostate cancers [138,139].

As above discussed, Gleason pattern 4, cribriform morphology is a prostate cancer subtype associated with unfavorable clinicopathological factors [33]. At molecular level, two main genomic abnormalities define two molecular subgroups of cribriform prostate cancers: *SPOP^mut^*, which is associated with *CHD1* and *MAP3K* codeletions, and (ii) *PTEN*^loss^, which is mutually exclusive with SPOP^mut^ [33].

Interestingly, a recent study provided evidence that SPOP-mutant prostate cancers failed to promote PD-L1 degradation due to their deficiency in binding to PD-L1 and promoting poly-ubiquitination [140]. These observations support the use of immunotherapy with immune checkpoint blockade inhibitors for the treatment of these prostate tumors [140].

### 4.3. CDH1 Abnormalities

The *CHD1* gene encodes the chromo-domain helicase DNA-binding protein 1 and, as above discussed, is one of the genes most frequently deleted or mutated in prostate cancer. This deletion is estimated to underlie 10–26% of all prostate cancers. Interestingly, the *CHD1* loss/deletion was more frequent among Chinese primary prostate cancer patients than in corresponding Caucasian patients (31% vs. 16%); conversely, Chinese prostate cancers exhibited only 6% of *ERG* gene rearrangements [141]. *CHD1* plays a key role as a tumor suppressor gene playing many biologic functions involving chromatin remodeling, AR-dependent transcriptional regulation, recruiting homologous recombination repair proteins to double-strand DNA breaks, and promoting cell invasiveness.

Targeted disruption of the *CHD1* gene in human cells leads to a defect in early double-strand break (DSB) repair in homologous recombination, resulting in an increased sensitivity to various DNA stresses, such as ionizing radiation, and to PARP as well as to PTEN inhibition [142]. Particularly, C*HD1* knockout in cells induces reduced H2AX phosphorylation and foci formation, as well as impairments in CtlP recruitment to the damaged DNA sites [142].

*CHD1* deletion is very frequently associated (in 65–85% of cases) with *MAP3K7* deletion [143]. *CHD1* and *MAP3K7* genes are codeleted in 10–20% of prostate cancers and correlated with poor disease-free survival [143]. Loss of *CHD1* has been implicated in the initiation of ETS prostate cancers, preventing *ERG* rearrangement in the prostate [56,144] and thus explaining the exclusivity between ETS positivity and homozygous loss of *CHD1*.

The analysis of large-scale genomic studies of the TCGA and other prostate cancer databases provided evidence that in the large majority of patients CHD1 loss and PTEN deletion (as well as *AKT1*, *AKT2*, *AKT3*, and *PIK3CA* gene alterations) were mutually exclusive [145]. This intriguing finding may be explained assuming that *CHD1* expression may be required for the progression of prostate cancer driven by PTEN loss. In line with this hypothesis, studies in animal models supported the essentiality of *CHD1* in *PTEN*-deficient prostate cancers: in PTEN-deficient prostate cancer cells, *CHD1* suppression inhibited colony formation and induced cell death [145].

A significant proportion of *CHD1*-deleted prostate cancers coexpress *SPOP* mutations. The study of some of these patients with hormone-naïve and castration-resistant tumor samples established 17% *CHD1* loss in tumor biopsies; *CHD1* loss and/or *SPOP* mutations were associated with a higher response rate to Abiraterone and a longer time on Abiraterone [136,146].

Recent studies suggest a role for *CHD1* in DNA double-strand break (DSB) repair in prostate cancer cells: particularly, CHD1 is involved in opening the chromatin around the DSB to facilitate the recruitment of homologous recombinant proteins [147]. These findings were confirmed by a subsequent study providing clear evidence that *CDH1* loss sensitizes to DNA damage and causes a synthetic lethal response to DNA damaging therapy in vitro, in vivo, ex vivo, in patient-derived organoid cultures, and in a patient with metastatic prostate cancer [148]. Particularly, *CHD1* loss leads to a decreased error-free homologous recombination repair [148]. Studies on advanced prostate cancers showed DNA repair pathway mutations in ~23% of patients, with the most frequent alteration being represented by *BRCA2* alterations (13% of alterations, 8% of somatic origin, and 5% of germline origin) and *ATM* alterations (~5%); more rare alterations are observed at the level of *BRCA1*, *RAD51B*, *RSD51C*, *FANCA*, and *CDK12* [67]. Inherited DNA repair mutations were observed in 11.8% of men with metastatic prostate cancer: 53% *BRCA2*, 1.6% *ATM*, 1.9% *CHEK2*, 0.9% *BRCA1*, 0.4% *RAD51D*, and 0.4% *PALB2* [145]. The overall frequency of germline DNA-repair genes was significantly higher in metastatic (11.8%) than in localized (4.6%) prostate cancers [149].

Interestingly, PARP inhibitors have been already tested in prostate cancer. A phase II clinical trial showed that Olaparib, a PARP inhibitor, showed antitumor activity castration-resistant prostate cancer, with 33% of responding patients; interestingly, 88% of the responding patients were positive for alterations of known repair-associated gens (*BRCA2* somatic loss, *BRCA2* germline mutations, *ATM* aberrations) [150]. In a recent phase 2 randomized clinical trial, Olaparib in combination with Abiraterone provided clinical efficacy benefit for patients with metastatic castration-resistant prostate cancer compared with Abiraterone alone [151]. However, more serious adverse events were observed in patients who received Olaparib and Abiraterone than Abiraterone alone [151]. The study of the circulatory cell-free DNA (cfDNA) of the patients allowed to monitor the response to Olaparib treatment: all DNA repair gene alterations observed in tumor biopsies were detectable also in cfDNA; allelic frequency of somatic mutations decreased selectively in responding patients; multiple subclonal aberrations reverting germline and somatic DNA repair mutations (*BRCA2*, *PALB2*) emerged as mechanisms of resistance [152].

A subset of advanced prostate cancers exhibits a hypermutated phenotype with a high number of mutations/Mb of DNA [57]. Pritchard et al. [58] reported that 12% of advanced prostate cancers are hypermutated and have mismatch of prostate repair gene mutations (MMRD) and satellite instability (MSI). The hypermutated subtype of prostate cancer is mainly due to *MSH2* and *MSH6* mutations, frequently corresponding to complex rearrangements [58]. *MSH2* loss was reported in 1.2% of primary prostate cancers: however, it was much more frequent in Gleason pattern 5 (8%) than in tumors with other scores (0.4%) [153]. Twenty-five percent of these patients have germline mutations in *MSH2*; furthermore, tumors with *MSH2* loss have a higher density of infiltrating CD8^+^ lymphocytes and CD8^+^ density correlates with mutation burden among cases with *MSH2* loss [153]. These findings were confirmed in a more recent study showing that defective mismatch repair was associated with mismatch repair gene mutations and increased immune, cell, immune checkpoint, and T cell-associated transcripts [154]. Interestingly, a retrospective analysis on 13 advanced prostate cancer patients with MMR mutations showed that in these patients, responses to standard hormonal therapies were very durable with a median PFS of 67 months to initial androgen deprivation and median of 26 months to Abiraterone/Enzalutamide therapy [155].

Prostate adenocarcinomas with focal pleomorphic giant cell features are rare prostate cancer subtype with dismal clinical prognosis. A recent study reported the genetic analysis of 8 cases of prostatic adenocarcinomas with focal pleomorphic giant cell features, showing that DNA damage repair mutations are common, with two out of eight having biallelic pathogenic mutations in homologous DNA repair genes and two out of eight having biallelic pathogenic mutations in mismatch repair genes (*MSH2* and *MLH1*) [156].

### 4.4. Androgen Receptor Abnormalities

Androgen receptor (AR) makes part of the nuclear receptor superfamily, possesses a structure similar to other steroid receptors and is a transcriptional factor for testosterone and dihydrotestosterone; its structure consists of four main domains, the N-terminal domain, DNA-binding domain, hinge region, and ligand-binding domain (Figure 3).

Ligand binding induces a change in receptor conformation, facilitating both nuclear targeting of AR (through bonding at the level of AR elements present in the promoter and enhancer regions of AR target genes) and the exposition of aregion of LBD, called transcriptional activation function 2, required for receptor homodimerization and stabilization (Figure 3). The DBD of *AR* is a highly conserved structure which contains two zinc finger domains, essential for conferring specificity for DNA binding (Figure 3). The NTD containing a region, called transcriptional activation function-1 (AF-1) which is essential for transcriptional activity (Figure 3) [157]. Approximately 300 AR coregulators have been identified and act as coactivators or corepressors of the AR transactivator effects. Coregulators can alter the transcriptional activity through modulation of a variety of processes, including AR homodimerization, stabilization and nuclear translocation, chromatic remodeling and DNA occupancy, recruitment of general transcription factors, and assembly of initiation transcription factors [157]. The most studied coactivators are members of the p160 coactivator family, comprising SRC1, SRC2, and SRC3. A common property of many coregulators is their ability to enzymatically modify AR and other components of the AR complex, through acetylation, methylation, phosphorylation, SUMOylation and ubiquitination [157]. These events then trigger cellular processes such as proliferation and cell invasion. An example of this link is given by *SPOP* missense mutations, blocking the E3 ubiquitin ligase activity of this protein normally involved in the degradation and turnovers of AR and SRC3 and leading to increased levels of AR.

ARs play a key role in prostate cancer development, particularly in castration-resistant prostate cancer (CRPC). Androgen deprivation therapy can suppress hormone-naïve prostate cancers, but prostate cancer cells undergo various types of adaptive changes of AR and acquire the capacity to survive under castration levels of androgens. These mechanisms of adaptation and resistance involve point mutations of *AR*, *AR* overexpression, production of constitutively active *AR* splice variants without ligand binding, changes of androgen biosynthesis, and changes of androgen cofactors.

Abnormalities in androgen receptor (AR) signaling pathway members are very frequent in prostate cancer. These abnormalities are important given the essential role for AR pathway for growth and differentiation of normal prostate and for treatment failure in castration-resistant disease. Alterations of the AR trough mutations, gene modifications, and/or overexpression are common (58% of cases), but occurred exclusively in metastatic samples [158]. However, the analysis of the AR pathway, including several known activators, coactivators, and corepressors, showed alterations in 56% of primary prostate cancers and 100% of metastases [158].

In untreated primary prostate cancer *AR* gene amplification is low, while 50–85% of CRPCs have an increased AR copy number [159,160,161,162]. These findings were confimed in a more recent study, showing that *AR* is amplified in 70% of metastatic prostate cancers and is associated with elevated AR mRNA expression [68]. *AR* amplification is a mechanism of resistance to androgen deprivation therapy. These findings were confirmed in a more recent study, showing that *AR* is amplified in 70% of metastatic prostate cancers and is associated with elevated AR mRNA expression [68]. AR amplification is a mechanism of resistance to androgen deprivation therapy [159].

More recently, it was reported the very frequent occurrence of amplification of an enhancer region of the AR, detected in virtually all prostate cancers postprogression [68,163,164]. The studies performed until now have mainly focused on protein-coding sequences, basically showing that the development of a metastatic, castration-resistant disease is associated with the development of *AR* mutations or *AR* gene amplifications. However, only very few studies have explored non-coding DNA regions in metastatic, castration-resistant prostatic cancers. Takeda and coworkers have reanalyzed previously published data from castration-resistant prostate cancers and identified repeated DNA sequences acting as enhancers that caused the abnormal amplification of the region upstream AR: duplication of this enhancer region improved the tumorigenic process and increased androgen resistance, while its targeting by gene editing de creased AR expression and tumor cell proliferation [163]. In a second study, Viswanethan et al. have performed a whole genome sequencing on 23 tumor samples, matched with normal tissues, derived from patients with castration-resistant prostate cancer, and discovered in tumor tissues some tumor-specific abnormalities, consisting of tandem duplications occurring in genome sequences located near the *AR* gene and the *MYC* gene [164]. Interestingly, the tandem duplicated region near the AR gene corresponds to the enhancer region identified by Takeda and coworkers [163]. This enhancer region was found to be amplified in 87% of cancer metastatic samples, in association with an amplified copy of AR gene [164]; importantly, this enhancer region was found to be amplified in only 2% of primary tumors [164]. A subset of cases displayed *AR* or *MYC* enhancer duplication in the context of a genome-wide tandem duplicator phenotype associated with CDK12 inactivation.

In a third study, Quigley and coworkers have performed whole genome sequencing of 101 samples of metastatic, castration-resistant prostate cancer tissue showing structural alterations of critical regulators of tumorigenesis not detectable by exome sequencing; the most frequent alteration consisted in the amplification of an intergenic enhancer region 624 kb upstream of the *AR* in 81% of patients, correlating with AR expression [68]. Tandem duplication hot spots also occurred in a region near *MYC* gene, at the level of lncRNAs associated with post-translational *MYC* regulation [68]. A gene duplication maps also near to the *FOXA1* gene [68]. Furthermore, this study showed also that different classes of structural variations are linked to distinct DNA repair deficiencies: *CDK12* mutation with tandem duplications, *TP53* inactivation with inverted rearrangements, and chromothripsis and *BRCA2* with inactivation with deletions [68].

It is of interest to note that Viswanathan et al. analyzed also data from three patients for whom tumor samples were available from before and later second-line antiandrogen therapy with enzalutamide: samples from after treatment display am amplification of *AR* enhancer and of the *AR* [164].

The main drug therapy of prostate cancer consists in reducing the levels of androgens, sutaining the survival and the proliferation of prostate cancer cells. Androgen deprivation therapy (ADT) is based on various types of of hormone therapy. Some of these treatments lower androgen levels or inhibit the biologic effects of androgens and are mainly represented in Table 2: (a) orchiectomy (surgical castration) based on removal of tresticles, the main source of androgens in the body; (b) luteinizing hormone-releasing hormone (LHRH) agonists, including Leuprolide, Goserelin, Triptorelin, and Histrelin, which act by lowering the level of trestosterone synthesized by testicles, providing a treatment that is commonly known as chemical castration or medical castration; (c) LHRH antagonists include only one drug, Degarelix, acting as a LHRH antagonist and lowering testosterone levels more rapidly thatn LHRH anatagonists; (d) CYP17 inhibitors include only two drugs—Abiraterone and Galeterone—blocking the enzyme CYP17, involved in androgen synthesis occurring in all cell types, including small amounts of androgens synthesized by tumor cells; (e) antiandrogens include the first-generation drugs Flutamide, Bicalutamide, and Nilutamide, acting as inhibitors of the androgens with AR; (f) antiandrogens of fisrt-generation include also Cyproterone, a derivative of Progesterone, which binds to AR and blocks the binding of androgens to this receptor; and (g) second-generation antiandrogens including five drugs: Abiraterone (a CYP17 inhibitor), Enzalutamide (a synthetic AR signaling inhibitor), Apalutamide (a small molecule synthetic AR antagonist), Darolutamide (a small molecule synthetic AR antagonist), and EPI-506 (a small molecule synthetic AR antagonist).

Androgen deprivation is the standard treatment strategy for metastatic prostate cancer, but patients undergoing this therapy invariably relapse despite the treatment induces castrate androgen level. This condition is known as castration-resistant prostatic cancer (CRPC). AR is highly expressed and transcriptionally active in CRPCs; this phenomenon seems to be related to synthesis of steroids from adrenal glands, such as dehydroepiandrosterone, which are a source for intratumoral synthesis of dehydrotestosterone (DHT) and are responsible for AR activation in CRPCs. More particularly, as prostate cancer progresses, pronounced changes in androgen metabolism are observed, involving increased expression of steroidogenic enzymes, as well as mutations in some key components of the steroidogenic machinery [165]. In some CRPCs the occurrence of a gain-of-stability mutation leads to a gain-of-function in 3beta-hydroxysteroid dehydrogenase type I, an enzyme catalyzing the initial rate-limiting step in the conversion of dehydroepiandrosterone to DHT: this mutation (N367T) does not affect the catalytic function of the enzyme, but favors rapid enzyme accumulation due to resistance to ubiquitination and degradation [165].

Androgen deprivation therapy is able to control for some time hormone-naïve prostate cancers, but prostate cancer changes AR and adopt a series of adaptation mechanism to survive in the presence on only low levels of androgens as those induced by ADT. Various mechanisms are responsible for the adaptation of prostate cancxer tumors to ADT and include point mutations in androgen receptor, AR amplification, changes of androgen biosynthesis, changes in AR cofactor in prostate cancer cells, AR variants [166,167]. These mechanisms have been explored in patients developing a condition of castration resistance during ADT [166,167].

Point mutations of the AR are observed in 15% to 30% of CRPCs, which are most frequently located at the level of the LBD and more rarely at the level of NTD [54]. These point mutations can activate AR by losing the specificity of the agonist or may lead to resistancxe to the inhibitory effect on an antiandrogenic agent. AR mutations can be detected either through analysis of tumor biopsies [54], but also through analysis of circulating tumor cells (CTCs) or circulating plasmatic DNA (ctDNA) [168,169]. CtDNA is a reliable technique for monitoring of genetic alterations of prostate tumors in mCRPC patients. Thus, a recent study on 514 mCRPC provided evidence that genetic alterations were detected in 94% of these patients at the level of ctDNA and, particularly, 22% of point mutations of *AR* and 30% of *AR* gene amplification [170].

Most of the *AR* mutations relevant at clinical level were L702h, t878A, H875Y, and F877L. T878A mutation is a missense mutation that results in loss of specificity for the agonist and, in consequence, the mutant *AR* results to be activable by progesterone, estrogen, flutamide, bicalutamide, and enzalutamide [168,171]. T878A exterts resitance to both first-generation and second-generation AR-antagonists. T878A confers also resistance to Abiraterone through a peculiar mechanismrelated to the effect of this drug that, via CYP17A1 inhibition, promotes elevated expression of progesterone in tumor tissues and promotes the formation of malignant clones bypassing Abiraterone inhibition [172]. The F876L mutation modifies the LDB and promotes resistance to Enzalutamide [173,174]. This mutant was subsequently shown to be F877L; F877L spontaneous mutations have been detected in patients treated with Enzaluitamide [168]. L702H mutation was found in ctDNA of CRPC patients with Abiraterone resistance; this mutation determines glucocorticoid-mediated activation of ARs [50,175].

For many years the standard therapy for men with a condition of metastatic prostatic cancer remained androgen deprivation therapy, mainly based either on orchiectomy or on the administration of gonadotropin-releasing hormone antagonists. In 2015–2016, two clinical trials, called CHARTED and STAMPEDE, provided evidence that androgen deprivation therapy combined with six-course of a taxane, Docetaxel, improved the overall survival of metastatic prostate cancer patients [176,177]. More recently, the STAMPEDE and the LATITUDE trials evaluated androgen deprivation therapy without or with Abiraterone and a glucocorticoid [178,179]. As above discussed, Abiraterone decreases androgen concentration at the level of the castration level by inhibiting CYP17A1 (and blocking 17 alpha-hydroxylase and 17,20 lyase) and exerts antitumor effects through a common metabolite, acting as an AR antagonist [180]. However, Abiraterone increases steroid precursors upstream of CYP17, if administered without glucocorticoids [181]. Both these studies—STAMPEDE and LATITUDE—provided evidence that Abiraterone treatment, in combination with glucocorticoids and androgen deprivation therapy, improved the risk and the overall survival of metastatic prostate cancer patients compared to androgen deprivation therapy alone [178,179]. The addition of Abiraterone plus prednisone to ADT in patients with newly diagnosed, high-risk castration-naïve prostate cancer improved overall patient-related outcomes, through analysis of progression of pain, prostate cancer symptoms, fatigue, functional decline, and health-related quality of life [182]. The benefits reported in the LATITUDE trial were achieved in selected patients with castration-sensitive prostate cancer classified at high-risk for the presence of two of three negative factors, including Gleason score ≥8, at least three osseous metastases, and visceral metastasis [182]. Very importantly, a recent study reported the final results of the LATITUDE study on 1199 patients randomly assigned to Abiraterone plud Prednisone group or placebo group; after an interim analysis performed at the end of 2016, the patients of the placebo group were allowed to crossover to receive Abiraterone and Prednisone plus ADT in an open-label extension of this study [183]. A final analysis of the study was performed after a median follow-up of 51.8 months with 46% of mortality and median overall survival of 53.3 months in the Abiraterone + Prednisone group, compared with a 57% mortality and 36.5 months of overall survival in the placebo group [183]. Adverse events were comparable in these two groups of patients [183]. These observations support the use of Abiraterone plus Prednisone as a standard of care in patients with high-risk metastatic prostate cancer. The STAMPEDE study compared 566 randomized patients to either Abiraterone + Prednisone or Docetaxel in association with long-term hormone therapy (standard of care therapy) for prostate cancer with metastatic or nonmetastatic high-risk disease, starting ADT for the first time, and showed no evidence of a difference in overall or prostate cancer-specific survival, but not in other relevant outcomes such as symptomatic skeletal metastatic events [184]. Worst toxicity grade was similar, even if different in the specific events, in the two treatment arms [184].

Two ongoing clinical trials with Enzalutamide should provide important informations about the therapeutic role of AR inhibition in combination with standard treatments in the hormone-sensitive prostate cancer setting. The ARCHES trial is investigating Enzalutamide with ADT versus ADT alone in prostate cancer patients with metastatic hormone-sensitive disease. The preliminary results of this trial were recently presented at the Genitourinary Cancers Symposium in San Francisco, CA, USA, 2019, showing a higher objective response rate (83% vs. 64% in the placebo group) and a 61% lower risk of disease progression or death. Data were still immature for analysis of the effect ts on overall survival. The ENZAMET study is evaluating Enzalutamide with ADT in comparison to conventional antiandrogens, such as Bicalutamide, plus ADT.

The first drug shown to be active in CRPC was docetaxel administered either with prednisone [185] or estramustine [186]. New hormonal agents have been shown to exert therapeutic effects on CRPC. The drug combination of Abiraterone and Prednisone was shown to slightly, but significantly improve the overall survival compared to Prednisone + Placebo (median overall urvival 34.7 months, compared to 30.3 months) [187,188]. Patients who progress with Abiraterone treatment may still have some benefit from subsequent Docetaxel therapy [189]. Other studies have been carried out using Enzalutamide—a new antiandrogen receptor antagonist. An initial study provided evidence that Enzalutamide prolonged the survival of CRPC patients previously treated with Docetaxel [190]. Enzalutamide improved the overall survival of chemotherapy-naïve CRCPC patients compared with placebo [191,192]. Importantly, Enzalutamide improved clinical outcomes irrespective of age of patients [193]. However, administration of the drug to older patients (aged 75 years or greater) should be made with caution, in view of the increased falls and cardiac events [193]. The second study, STRIV E was similar to the TERRAIN study in design, with the exception that this study enrolled patients with both nnCRPC and mCRPC [194]. In this study, patients received treatment until confirmed PSA or radiographic progression. Importantly, Enzalutamide treatment improved PFS and reduced the risk of progression or death by 76% compared with Bicalutamide [194]. Since Enzalutamide prolongs overall survival among patients with metastatic CRPC, it was tempting to hypothesize that this drug would delay metastasis in patients with nonmetastatic CRPC and rapidly rising PSA level [195]. The results of a recent clinical trial (PROSPER study) involving the randomization of 1401 patients with nonmetastatic CRPC with rapidly rising PSA level showed that 23% of patients in the Enzalutamide group had metastasis or died, compared with 49% in the placebo group; furthermore, the metastasis-free survival was 36.6 months in the Enzalutamide group versus 14.7 months in the placebo group [195]. These results support the view that in this category of patients Enzalutamide induces a significant 71% lowering of the risk of metastasis or death than placebo. The analysis of patient-reported outcomes in this study showed that patients who received Enzalutamide had low pain levels and prostate cancer symptom burden and high health-related quality of life compared to those in the placebo group [196]. These findings support the view that Enzalutamide induces a clear clinical benefit by delaying pain progression, symptom worsening, and decrease in functional status compared with placebo [196].

A very recent phase II randomized clinical trial compared the quality life of metastatic CRPC patients undergoing treatment with Abiraterone plus Prednisone or Enzalutamidse, showing that elderly patients treated with Abiraterone had better quality of life over time compared with those treated with Enzalutamide, while no significant difference was observed between treatments for the younger subgroup of patients [197].

Given the resultrs observed using Abiraterone in metastatic CRPC patients it seemed logical to explore a possible synergism between this agent and radiotherapy. Thus, the randomized ERA 223 study explored the addition of radium-223 to Abiraterone and Prednisone or Prednisolone in patients with metastatic (bone metastases) CRPC: however, the addition of radium-223 to Abiraterone plus Prednisone or Prednisolone did not improve symptomatic skeletal event-free survival and was associated with an increased frequency of bone fractures compared with placebo [198]. It is important to note that, in the context of the STAMPEDE study, it was shown that radiotherapy to the prostate did not improve overall survival for patients with newly diagnosed metastatic prostate cancer [199].

Another study explored the combination of Abiraterone and Enzalutamide. Particularly, the PLATO study explored whether Enzalutamide resistance could result from raised androgens and can be overcome by combination with Abiraterone [200]. In this trial, patients with chemotherapy-naïve metastatic CRPC initially receive Enzalutamide; patients with PSA progression were randomly assigned to Abiraterone and Prednisone with either Enzalutamide or placebo until disease progression [200]. Median progression-free survival was 5.7 months in the combination group and 5.6 months in the control group [200]. According to these findings, it was concluded that combining Enzalutamide with Abiraterone is not indicated after PSA progression during treatment with Enzalutamide alone [200].

Apalutamide is a new second-generation antiandrogen that has a structure and a mechanism of action similar to Enzalutamide; however, despite similar in vitro profiles, Apulatamide had in vivo increased potency compared to Enzalutamide. The SPARTAN trial evaluated patients with nonmetastatic CRPC and a PSA doubling time of 10 months or less; patients were randomized to receive either Apalutamide or placebo. In the primary planned analysis, the median metastasis-free survival was 40.5 months in the Apulatamide group, as compared with 16.2 months in the placebo group [201]. Importantly, Apalutamide not only improved metastasis-free survival, but also prolonged the time to symptomatic progression [202]. Daralutamide is a second-generation antiandrogen that acts as a high-affinity AR atagonist and, similarly to Enzolutamide, inhibits testosterone-induced nuclear translation of AR; it antagonizes both overexpressed and mutated ARs. The ARAMIS randomized, double-blind, placebo-controlled phase 3 trial evaluated castration-resistant nonmetastatic prostate cancer patients with a PSA-specific doubling time of 10 months or less; the patients were assigned either to receive Darolutamide or placebo [203]. Duralutamide treatment significantly prolonged the metastasis-free survival from 18.4 months observed with placebo to 40.4 months with Darolutamide [203]. The incidence of adverse events was similar for Darolutamide and placebo [203]. An ongoing randomized clinical trial (ARASENS, NCT02799602) is evaluating Dorolutamide versus placebo in addition to standard ADT and Docetaxel in aptients with metastatic hormone-sensitive prostatic cancer.

Since these recent studies have shown the comparable efficacy of Enzalutamide, Apalutamide, and Darolutamide to extend metastasis-free survival of nonmetastatic CRPC, the problem of which of these three drugs to choos for treatment is open and the decision would ideally be based on potential adverse effects, particularly in the context of the presence of patient’s comorbidities [204].

Galaterone is a new androgen-targeting agent that acts inhibiting CYP17: this agent antagonizes the AR, inducing a degradation of both full-length and AR-V7 [205]. Furthermore, Galaterone targets two homologous deubiquitinating enzymes—USP12 and USP46—which control the AR-AKT-MDM2-P53 signaling pathway [206]. Phase I and II studies with Galatereone (ARMOR 1 and ARMOR2) have shown a decline in PSA levels in 49% of treated patients [207]. An ongoing study is evaluating Galaterone in comparison with Enzalutamide in prostatic cancer patients expressing AR-V7 (ARMOR3, NCT02438007T). EPI-506 is an antagonist of the transcriptional regulatory region AF-1 (activation function-1) of the AR, a region present in the NTD of the receptor, and acts as an inhibitor of the growth of prostate cancer cells with aberrant AR activity, including cells with constitutively active AR-7 [208]. A phase I/II clinical trial using EPI-506 in patients with metastatic CRPC resistant to Abiraterone and/or Enzalutamide is ongoing (NCT02606123).

Two recent studies have addressed the important problem of evaluating the impact of androgen deprivation therapy in localized prostate cancer. In a phase II trial, McKay et al. have evaluated neoadjuvant Enzalutamide and Leuprolide (EL) with or without Abiratorone and Prednisolone (ELAP) before radical prostatectomy (RP) in locally advanced prostate cancers. The poathologic complete response or minimal residual disease rate was 30% in ELAP-treated patients and 16% in EL-treated patients; tumor ERG positivity, or PTEN loss, was associated with more extensive residual tumors at radical prostatectomy [209]. Neoadjuvant hormone therapy followed by radical prostatectomy results in favorable pathologic responses in some patients, but longer follow-up is required to assess the impact of the neoadjuvant therapy on recurrence rates. The second study addressed the im portant problem to assess whether the addition of chemotherapy with Docetaxel to ADT and radiotherapy may improve the outcome of lacalized high-risk prostate cancer. The results of the randomized phase III study RTOG 0521 showed that for patients with high-risk nonmetastatic prostate cancer, chemotherapy with docetaxel improved overall survival from 89% to 93% at 4 years, with improved disease-free survival and reduction in the rate of distant metastasis [210]. Studies of neoadjuvant-intensive androgen deprivation represent a tool to investigate the mechanisms of drug resistance. Thus, Swalinsky et al. have investigated the reisdial cancer foci of 18 prostate cancer patients treated with neoadjuvant-intensive androgen deprivation therapy (leuprolide, abiraterone, and predinisone) and analyzed them for resistance mechanisms [211]. Transcriptome profiling provided evidence that reduced, but persistent AR activity in residual tumor foci, with no increase of neuroendocrine differentiation. *RB1* genomic loss was frequently observed and inversely correlated with proliferation [211]. In patients where two or three tumor foci were dissected and analyzed, it was observed a common clonal origin, but multiple oncogenic alterations unique to each focus were also identified [211]. These findings support the view that neoadjuvant intense androgen deprivation select subclones with oncogenic alterations present in primary prostate cancers [212].

Prostate cancer may originate from either glandular luminal or basal cells. A retrospective analysis on a large set of patients undergoing radical prostectomy and postsurgery androgen deprivation therapy showed that (a) patients with luminal tumors exhibit increased androgen signaling and (b) patients with luminal B tumors have poorer outcomes, but potentially improved response to postoperative androgen deprivation therapy [212].

It is important to note that for the patients with localized prostate cancer there is no indication for androgen deprivation therapy and the current therapeutic approachjes are represented by either total prostatectomy or a so-called “watchful-waiting” approach. It is commonly believed that total prostatectomy reduces mortality among men with localized prostate cancer, but evidence derived from randomized clinical trials with long-term follow-up is very limited. In this context, it is particularly important to mention a recent randomized study by the Scandinavian Prostate Cancer Group with a 29-year follow-up, showing that patients with localized prostate cancer and a long-life expectancy benefited from radical prostectomy, with a mean of 2.9 years of life gained [213]. A high Gleason score and the presence of extracapsular extension in the radical prostatectomy specimens were highly predictive of death from prostate cancer [213].

A recent study discovered an additional new mechanism of *AR* deregulation in advanced, metastatic prostate cancer. In fact, Rodriguez-Bravo and coworkers discovered abnormalities of nuclear pores composition during progression to metastatic, lethal prostate cancer, related particularly to the overexpression of the importin Nup POM121; POM121 overexpression promotes nuclear import of key transcription factors driving prostate cancer development, including AR, MYC, and GATA2 [214]. Inhibition of this mechanism reduced tumor growth and restored standard therapy efficacy [214].

The majority of metastatic prostate cancers progress during standard therapy based on androgen deprivation, but in large part remains AR-dependent, as demonstrated by their expression of AR and AR-regulated genes. Interestingly, a recent study showed that metastatic prostate cancers are heterogeneous for that concerns AR/AR signaling expression and neuroendocrine features and, according to these two parameters, can be subdivided into three subgroups: AR^+^/NE^-^, AR^-^/NE^+^, and AR^-^/NE^-^ [215]. Chronic androgen deprivation therapy increased of about four fold the proportion of AR^-^/NE^-^ CRPCs: this subgroup of prostate cancers retained AR amplification, TP53 mutations and PTEN loss like AR^+^/NE^-^ cancers, thus suggesting that differences in recurrent genetic alterations cannot explain the biologic differences between these two prostate cancer subgroups (Figure 4) [215]. Large-scale cellular and molecular screening studies suggest that AR^-^/NE^-^ tumor growth is driven by acquisition of FGFR/MAPK signaling and confers sensitivity to FGFR and MAPK signaling inhibitors [215]. These results provide a clear rationale for clinical trials based on FGFR/MAPK inhibitors in AR^-^/NE^-^ CRPC patients [215]. AR^-^/NE^-^ prostate cancers could be issued from rare AR^-^/NE^-^ prostate stem cell populations present in human prostate [216].

The *AR* has several splicing variants. Wild type full-length *AR* possesses the N-terminal domain (NTD) encoded by exon 1, the DNA-binding domain (DBD) encoded by exons 2 and 3, the Hinge Domain (HD) encoded by exon 4, and the Ligand Binding Domain (LBD) encoded by exons 4–8. More than 20 variants of AR have been described, the majority of which lack the C-terminal domain, including the LBD [217]. *AR-V7* is the most commonly detected variant in CRPC. Importantly, variants without the LBD are functionally active without androgens. *AR-V1* is truncated at the end of exon 3 and contains 19 amino acids from cryptic exon 1; *AR-V7* is truncated at the end of exon 3 and contains 16 amino acids from cryptic exon 3; *AR-V657* has exons 5–7 spliced out and contains only a small portion of the LBD; *AR-V9* is truncated at the end of exon 5 and contains 16 amino acids from cryptic exon 3 [217]. *AR-V1* and *AR-V7* are the mostly abundant AR variants, with a 20-fold higher expression in CRPC than in hormone-naïve prostate cancer [217]. Of these variants, AR-V7 is the most investigated in experimental studies and is associated with an increased risk of biochemical relapse and inferior overall survival outcomes [217]. AR-V7 positive patients treated with Abiraterone or Enzalutamide had a lower PSA response rate and a poor prognosis [218].

A recent study explored 168 prostate cancer patients undergoing androgen deprivation therapy and showed that those positive for AR-V7 expression (19%), compared to AR-V7-negative patients, exhibited lower prostate-specific antigen response rates to androgen deprivation, a much shorter time to castration-resistant prostate cancer, and lower cancer-specific and overall survival [219]. All the studies carried in the last years suggest that AR-V7-positive prostate cancer should be considered as a separate prostate cancer subtype [219]. The negative impact of AR-V7 expression on the response to antiandrogen therapies was confirmed by a recent prospective study. Thus, the PROPHECY multicenter, prospective-blinded study of prostate cancer patients with high-risk mCRPC starting Abiraterone or Enzalutamide treatment, incvestigated the prognostic significance of baseline AR-V7 on clinical progression-free survival [209]. Detection of AR-V7 in CTCs was independently associated with shorter PFS and OS with Abiraterone and Enzalutamide [220].

*AR-V9* is frequently coexpressed with *AR-V7* in prostate cancer metastases and might also lead to a ligand-independent growth of prostate cancer cells; furthermore, *AR-V9* high-levels are predictive of resistance to Abiraterone [221].

Several studies have attempted to define the biology of AR-V7, particularly for that concerns its oncogenic activity. AR-V7 is a truncated isoform of the normal AR-full length protein that lacks the LBD, but retains DBD, involved in AR dimerization and DNA interactions, and the NTD, required for the majority of AR transcriptional activity [222]. These molecular changes maintain AR in a constitutively active state, even in the absence of its ligand [222]. Furthermore, the remarkable structural difference of AR-V7 with respect to AR-WT, confers to the variant receptor a different spectrum of transcriptional activity [222]. *AR-V7* is originated from an alternative splicing of AR mRNA at the level of the cryptic exon 3, compared to the canonical 3′ splice site of AR-WT. *AR* amplification is considered as a condition favoring the genesis of AR-V7 [223]. Few are the genetic determinants that have been identified as promoters of AR-7 generation [224]. In this context, a recent study provided evidence that the histone demethylase JMJD1A promotes alternative splicing of AR-7 in prostate cancer cells: interestingly, knockdown of JMJD1A inhibited splicing of AR-V7, but not AR-WT, in a minigene reporter assay [224].

Sharp and coworkers have performed a recent study based on the screening of AR-V7 protein by immunohistochemistry on a large set of prostate cancers—358 primary and 293 metastatic tumors—and found that AR-V7 protein is rarely (<1%) expressed in primary cancers, but is frequently (75% of cases) detected in metastatic tumors [225]. A further increase in AR-V7 protein expression is observed in metastatic cancers after androgen inhibitor therapy [225]. In CRPCs, AR-V7 protein expression correlates with AR-V7 expression and AR copy number [225]. Interestingly, *AR-V7* expression was heterogeneous between different metastases of the same patient but was similar within the same metastasis [225]. Finally, in gene expression studies *AR-V7* expression correlated with *HOXB13*, a critical regulator of AR-V7 function [225].

The molecular profiling of prostate cancer with liquid bioipsies, such as circulating tumor cells (CTCs) and circulating tumor DNA (ct-DNA) analysis, concerning the presence of AR-V7 in tumor cells provides informative evaluation of tumor prognosis. Thus, Antonorakis and coworkers performed a study on a large set of patients showing the clinical impact of AR-V7 mRNA detection in circulating tumor cells in men with mCRPC treated with firt- and second-line Abiraterone and Enzalutamide [226]. In three cohorts of patients—CTC^-^, CTC^+^/AR7^-^, and CTC^+^/AR-V7^+^—outcomes were best for the CTC^-^ cohort, intermediate for CTC^+^/AR-V7^-^ patients, and worse for CTC^+^/AR-V7 patients [226]. Other recent studies have confirmed the clinical utility of AR-V7 detection in ct-DNA as a biomarker for treatment of CRPC [227,228]. Finally, recent studies have also shown that the testing of AR-V7 mRNA in circulating tumor cells may provide an AR-V7-positive and AR-V7 negative score in a clinically acceptable time range and may provide a guide for the choice of the optimal treatment [229].

The definition of the oncogenic mechanism of AR-V7 largely remains to be elucidated. However, a recent study provided evidence that HOXB13 could represent an important mediator of AR-V7 in prostate cancer cells [230]. This study did take advantage on the development of an AR-V7-specific antibody allowing Chip (chromatin immunoprecipitation)-exonuclease sequencing, a method capable of determining the gene binding locations at the level of which AR-V7 binds within the prostate [230]. The results of this analysis showed that AR-V7 target genes are heterogeneous, even between individual CRPC patient tissues [230]. This study showed also that AR-V7 binding is dependent on HOXB13 and targeting of this transcription factor is sufficient to inhibit CRPC in cell lines and in vivo [230]. Interestingly, AR-V7 and HOXB13 are coexpressed in the same tissues and circulating tumor cells in prostate cancer patients [230]. It is important to note also that in this study it was shown that AR-V7 and AR-FL do not associate at the level of molecular complexes, suggesting that the two Ars have distinct roles in prostate cancer cells [230]. Another recent study provided evidence that AR-V7 is able to bind both canonical and noncanonical androgen receptor elements (ARE): consequently, AR-V7 in part recapitulates AR-WT’s function but induces also an additional program of gene expression via binding to gene promoters rather than ARE enhancers [231]. AR-V7 binding and AR-V7-mediated activation at these noncanonical AREs require ZFX or BRD4, while binding to canonical AREs requires FOXA1 [231]. Inhibition of either ZFX or BRD4 suppresses the growth of prostate cancer cells [231]. AR-V7 contributes to development of resistance to androgens by repressing the transcription of genes with tumor-suppressive activity. Thus, Cato and coworkers have exploited data on the cistrome and trascriptome to reveal that AR-V7 acts as a transcriptional repressor and heterodimerizes with FL-AR at the level of a functionally relevant subset of growth-suppressive genes [232]. Four AR-V7-repressed genes with a negative effect (i.e., genes acting as negative regulators of cell proliferation) on CRPC proliferation are SLC30A7, B4GALT1, HIF1A, and SNX14 [221]. Expression of the AR-V7-repressed genes and AR-V7 protein expression are negatively correlated and predict for outcome in prostate cancer patients [232].

The emergence of AR splice variants is proposed as a mechanism of treatment resistance: thus, AR-Vs have been implicated in the treatment failure of prostate cancer patients undergoing androgen deprivation therapy alone or in combination with radiotherapy [222]. Therefore, there is absolute need for developing therapies capable of targeting AR-Vs and of decreasing their deleterious effects. Since these AR-Vs are originated through a molecular mechanism of alternative splicing, disruption of alternative splicing through modulation of spliceosome is an obvious potential therapeutic approach: however, the understanding of the mechanisms underlying the alternative splicing is still very limited, reflecting the strong limitations of current spliceosome-targeted therapeutic agents [222]. The heterogeneous nuclear ribonucleoprotein A1 (hnRNP A1) is a RNA-binding protein playing a key role in alternative pre-mRNA splicing regulation. A recent study showed that hnRNP A1 plays a pivotal role in the generation of AR splice variants, such as AR-V7. In line with this conclusion, hnRNP A1 is overexpressed in prostate cancers compared to benign prostatic tumors, and its expression is regulated by c-Myc [233]. CRPC cells resistant to Enzalutamide display higher levels of hnRNP A1, AR-V7 and c-Myc: HnRNP A1 and AR-V7 levels are positively correlated with each other in prostate cancer. Importantly, downmodulation of hnRNP A1 and of AR-V7 resensitizes Enzalutamide-resistant cells Enzalutamide, suggesting that enhanced expression of nhRNP A1 may determine resistantance to antiandrogen therapies by promoting the generation of splice variants [233]. Quercetin, a naturally occurring polyphenolic compound, decreases the expression of hnRNP A1 and, concomitantly, of AR-V7 [234]. The inhibition of AR-V7 expression by quercetin sensitizes Enzalutamide-resistant prostate cancer cells to the inhibitory effect of Enzalutamide [223]. The compound VPC-80051, discovered by a computer-aided drug discovery approach, was recently reported as the first small molecule inhibitor of hnRNP A1 splicing activity; this drug interacts directly with hnRNP A1 and reduces AR-V7 mRNA levels in prostate cancer cells [235].

Recent studies have explored the sensitivity of prostate cancers bearing AR-V7 to current treatments. As above mentioned, AR-V7-positive metastatic prostate cancers scarcely respond to androgen deprivation therapy. Recent studies provided evidence that AR-V7-positive prostate cancers respond better to chemotherapy taxanes than to antiandrogens. Thus, in an initial study, Antonorakis et al. provided evidence that detection of AR-V7 in CTCs from men with metastatic CRPC is not associated with resistance to taxane chemotherapy; in fact, in AR-V7-positive patients, taxanes were more efficacious than Abiraterone or Enzalutamide therapy, whereas in AR-V7-negative patients, taxanes and Abiraterone or Enzalutamide display comparable efficacy [236]. Furthermore, this study showed that AR-V7-positive and AR-V7-negative tumors were similar in their response rate to taxane chemotherapy [236]. These findings were confirmed by Onstenk et al., providing evidence that the response of mCRPC to Cobazitaxel was independent of the AR-V7 status of patients [237]. Scher et al., assessing nuclear-localized AR-V7 in CTCs of CRPC patients, showed that these patients respond better to taxanes (overall survival of 14.3 months) than to androgen deprivation (overall survival of 7.3 months) [238]; in contrast, patients who are negative for AR-V7 who are treated with AR inhibitors had superior overall survival relative to those treated with taxanes (19.8 vs. 12.8 months) [238]. The response of AR-V7-positive and AR-V7-negative patients to taxanes was comparable [238]. The TAXYNERGY trial recently explored clinical benefit potentially deriving from early taxane switch (from Docetaxel to Cobazitaxel) and CTC biomarkers to explore mechanisms of resistance and sensitivity to taxanes in patients with chemotherapy-naïve, metastatic, CRPC [239]. The results of this study showed that the early taxane switch strategy was associated with improved PSA response rates, compared to historical controls [239]. However, another recent study, TAXYNERGY, failed to show a better response of AR-V7-mutated prostate cancers to taxanes: in fact, they observed a median PFS of 12 vs. 8.48 months for AR-V7-negative vs. AR-V7-positive tumors [239]. Interestingly, in these patients, the expression of AR-Vs can be investigated in circulating tumor cells [228]. In exploratory analyses, it was assessed the prevalence of AR-V7 aR-V567^ES^ and the association of these biomarkers with PSA reponse rates and PFS [240]. According to the absence or presence of these biomarkers, prostate cancers were subdivided into AR-V7^-^, AR-V7^+^, AR-V567^ES+^, and AR-V567^ES+^/AR-V7^+^ [240]. The analysis of the nuclear expression of AR, a measure of AR activation status, showed a decrease of only 0.4% in AR-V7^+^ tumors, versus a 12.9% and 26% decrease in AR-V7^-^/AR-V567^ES-^ and AR-V7^-^/AR-V567^ES+^, respectively [240]. This last finding suggests a dominant role of AR-V7 over AR-V567^ES^. Median PFS was 12 vs 8.48 months for AR-V7^-^ versus AR-V7^+^ and 12.71 versus 7.3 months for AR-V567^ES-^ versus AR-V567^ES+^; for AR-^V7+^, AR-V7^-^/AR-V567^ES+^ and AR-V7^-^/AR-V567^ES-^ patients, median PFS was 8.5, 11.2 and 16.6 months, respectively [240]. This observation challenges the view that AR-V7-positive prostate cancers respond to taxanes as well as AR-V7-negative tumors. The absence of the hinge domain, required for microtubule binding, in the AR-V7 variant may help to explain the low sensitivity of AR-V7-positive prostate cancers to taxanes [241].

Interestingly, a recent study showed that inhibition of de novo lipogenesis (fatty acid synthase) may represent a new strategy to reduce androgen receptor signaling in castration-resistant prostate cancer [242]. In fact, fatty acid synthase (FASN) inhibitor antagonizes the growth of castration-resistant prostate cancer cells and results in reduced protein expression and transcriptional activity of both full-length AR and AR-V7; importantly, in vivo, FASN inhibitor reduced the growth of AR-V7-driven prostate cancer xenografts [242]. In metastatic prostate cancers, AR-V7 was coexpressed with FASN [242].

As above discussed, small cell neuroendocrine cancer is a very aggressive subset of prostate cancer that is rare at time of diagnosis but increasingly more frequent following emergence of resistance to AR-targeted therapies and thus represents one of the mechanisms of AR-resistance in CRPCs. A recent study further clarified the molecular mechanisms operating in SCNCs and responsible for resistance to AR targeting [243]. Thus, Aggarwal and coworkers have shown that SCNC tumors are characterized by (i) a typical SCNC gene expression signature involving low or absent expression of AR signaling, elevated expression of genes associated with small cell morphology, elevated expression of CDKN2A and E2F1, low expression of NOTCH2, and elevated expression of the neural transcription factor ASCL1; (ii) tumors with the SCNC phenotype bear reduced frequency of AR gene locus amplification (40% vs. 72) and of *AR* enhancer amplification (20% vs. 83%); and (iii) even SCNC tumors displaying *AR* amplification exhibit very low AR expression; (iv) SCNC tumors more frequently harbor biallelic *RB1* inactivation (60% vs. 9.4%) [243]. At the moment, it is still unclear whether the process determining the emergence of SCNC is related to transdifferentiation from a pre-existing adenocarcinoma or from clonal expansion.

### 4.5. PTEN Gene Abnormalities

*PTEN*, a tumor suppressor gene acting as a negative regulator of the PI3K/AKT/mTOR pathway, frequently undergoes copy number loss as an early event in prostate cancer development and its loss is correlated with progression to castration-resistant disease. In fact, more than 40% of metastatic prostate cancers have *PTEN* mutations and up to 70% of the late stage prostate cancer samples exhibit loss of PTEN function or activation of the PI3K signaling pathway [244]. Loss of PTEN is associated with increased risks of biochemical recurrence and metastasis [244]. As above discussed, comprehensive genomic profiling of metastatic castration-resistant prostate cancers, *PTEN, TP53*, and *RB1* alterations have been shown to be enriched in resistant tumors [61,67]. A recent study compared the frequency of the alterations of these three genes in a large group of prostate cancers subdivided into three subgroups–L-CSPC (localized castration-sensitive prostate cancer), M-CSPC (metastatic castration-sensitive prostate cancer), and M-CRPC (metastatic castration-resistant prostatic cancer)–representing three stages of disease progression [245]. This analysis showed (a) a progressive increase of the frequency of alterations of these three genes and this phenomenon was still more evident taking into account the frequency of co-alterations of at least two of these genes; (ii) no significant changes at the level of the mutational tumor burden; and (iii) a very pronounced increase in the percentage of copy number altered genome (Figure 5) [245].

In genetically engineered mouse models *PTEN* loss was found to cooperate with TMPRSS-ERG fusion, c-myc upregulation and loss of function of the *NKX3.1* homeobox gene in promoting prostate tumorigenesis. Finally, *PTEN* loss may contribute also to the development of castration resistance. Recent studies have shown a direct link between *PTEN* loss and the development of castration resistance. In fact, it was shown that *PTEN* loss suppresses androgen-responsive gene expressions by modulating androgen receptor transcription factor activity [246]. Using a series of AR reporter and *PTEN* knockout compound mice provided evidence that *PTEN* loss directly represses endogenous *AR* expression in prostatic epithelial cells [247]. Prostate cancers lacking PTEN protein exhibited a shorter survival to abiraterone acetate treatment than patients with normal *PTEN* expression [248]. These observations strongly suggest the existence of a PI3K and AR crosstalk as a mechanism of prostate cancer development and progression. A recent study provided evidence about a strong association between hypoxia gene signature and allele loss of *PTEN*; *PTEN* loss was part of a focal deletion also including *RNLS* and *ATAD1* genes [249]. Tumors with allelic *PTEN* loss also exhibited elevated prostate genomic alterations [249]. Patients with prostate cancers bearing both *PTEN* loss and tumor hypoxia are at higher risk of relapse within 2 years, a feature associated with increased prostate cancer-specific mortality [249].

It is important to note that in mouse models *PTEN* loss alone results in prostate intraepithelial neoplasia (PIN) which, following a long latency, can progress to high-grade adenocarcinoma, albeit with minimally invasive and metastatic features. Therefore, additional mutational events must occur to allow cancer progression. A recent study showed that the strong induction of the TGFβ/BMP-SMAD4 signaling axis elicited by *PTEN* loss represents a constraint to cancer progression in *PTEN*^-/-^ animals. However, the concomitant *PTEN* and *SMAD4* loss induces prostate cancer development [250]. Studies on tumor specimens corresponding to various tumor stages support the clinical relevance of these observations. On the other hand, the concomitant loss of *PTEN* and *TP53* in prostate epithelial cells leads to invasive prostate cancer with features of human prostate cancer [251]. The study of *PTEN^-^/TP53^-^* mouse model of primary prostate cancer cells and tumor xenografts allowed to show that hexokinase 2-medaited aerobic glycolysis is required for *PTEN^-^/TP53-*deficiency-driven tumor growth [251]. Recent studies suggest that *ATF3* (Activating Transcription Factor 3), a transcription factor responding to diverse cellular stresses and regulating oncogenic activities, acts as a tumor suppressor for the subset of prostate cancers harboring dysfunctional *PTEN* [252]. In fact, *ATF3* promotes *AKT* activation and prostate cancer development in PTEN knockout mouse models [251]. In spite the frequent *PTEN* loss, PI3K inhibitors are largely inactive in metastatic prostate patients. The modest efficacy of these inhibitors was attributed to various factors, including the coexistence of other mutations in these tumors and relief of feedback inhibition of physiologic signaling in tumors treated with these inhibitors. A recent study clarified the molecular mechanisms responsible for resistance of *PTEN-*mutated prostate cancer cells to PI3K inhibitors [253]. In fact, Schwartz and coworkers showed that, in PTEN-mutated prostate cancers, PI3Kalpha activity is suppressed and PI3K signaling is mediated by PI3Kbeta; a selective PI3Kbeta inhibitor only transiently inhibits AKT/mTOR signaling in these cells because it lives feedback inhibition of IGF1R and, through this mechanism, causes PI3Kalpha activation and a rebound in downstream signaling [254]. This rebound is blocked by the combined addition of both a PI3Kalpha and a PI3Kbeta inhibitor; however, in *PTEN*-deficient mouse models of prostate cancer this efficient PI3K inhibition causes a marked activation of AR activity [254]. Triple therapy with an AR inhibitor and two PI3K inhibitors resulted in near complete suppression of AR-dependent prostate tumors in vivo [254].

In addition to *PTEN* alterations, alterations of other members of the PI3K signaling pathway are frequent in prostate cancer, involving alterations of *PI3KCA* (13%), *PIK3R1* (6%), *NF2* (3%), *AKT1* (1.5%), and *NF1* (1.5%) [255]. In advanced prostate cancers, PI3K-pathway alterations are significantly associated with *TP53* and *AR* mutations [256]. Thus, the PI3K pathway is closely associated with the progression to CRPC after ADT, via mechanisms involving a complex interaction with the AR pathway [257]. Since the interactions between the AR and the PI3K pathways may be a mechanism of resistance to ADT, PI3K inhibitors may represent a potential therapy in CRPC patients [257,258]. However, clinical trials using PI3K inhibitors as monotherapy failed to achieve clinically significant responses in prostate cancer [258]. The unsuccessfull results of PI3K inhibitors as monotherapy in CRPC is related to two different mechanisms: relief of nnegative feedback loops emanating from pathjway effectors of PI3K, thus leading to rebound activation or reactivation of the PI3K activation, and activation of alternative survival pathways [258]. These findings have strongly supported the view that PI3K inhibitors always combined with inhibitors of other pathways and, particularly, with AR inhibitors [259,260]. Thus, it is not surprising that, as supported by preclinical models in prostate cancer models, several ongoing clinical trials are evaluating Abiraterone or Enzalutamide in association with different PI3K inhibitors [259,260].

Recent studies have explored the contribution of *PTEN* loss to prostate cancer development. A first study showed that *PIK3CA* mutation correlates with poor prostate cancer prognosis and induces prostate cancer development in murine models [261]. Furthermore, *PI3KCA* mutations and *PTEN* loss coexist in prostate cancer and cooperate in mouse models to accelerate prostate cancer development and facilitate the progression to castration-resistant metastatic tumors [261]. A second study described a peculiar property of prostate cancer cells. In fact, it was shown that *PTEN*-deficient prostate cancer cells proliferate even in the presence of low concentrations of nutrients by scavenging necrotic debris and extracellular proteins through a process of micropinocytosis requiring AMPK activity [262]. This observation suggests that blocking micropinocytosis by inhibiting AMPK or PI3K may represent an effective therapeutic strategy, particularly in combination with therapies that cause nutrient stress [262]. However, the effects of AMPK on the development and proliferation of prostate cancer are complex and have originated contradictory results. AMPK expression and activity are increased in prostate cancer tissue compared to normal prostate tissue [262]. AMPK appears to be a target of the metabolic effect of androgens in prostate cancer cells enhancing glycolysis, glucose, and fatty acid oxidation [262]. AMPK-mediated metabolic changes increase ATP intracellular levels and stimulate mitochondrial biogenesis, thus promoting prostate cancer cell growth [262]. At variance3 with these findings, a recent study showed that deletion or pharmacological inhibition of calcium/calmodulin-dependent protein kinase 2 (CAMKK2) protects against prostate cancer development in preclinical mouse models lacking expression of prostate-specific PTEN [263]. CAMKK2 is induced in prostate cancer cells by AR signaling and CAMKK2 in turn activates AMPK. AMPKβ1 was identified from a siRNA screen as a specific candidate required for prostate cancer cell growth; *AMPKβ1* expression is increased following *PTEN* deletion [263]. Inhibition of AMPK, via genetic deletion of *AMPKβ1*, increases disease progression in PTEN-null prostate cancer models [263]. *AMPKβ1* could exert a protective effect on prostate cancer progression in vivo.

Farnesyl disphosphate synthase (FDPS), a mevalonate pathway enzyme, is highly expressed in many tumors, including prostate cancer. FDPS is significantly more expressed in prostate cancer tissues than in the corresponding normal tissue; FDPS expression is associated with increasing Gleason score, *PTEN* deficient status, and poor survival of prostate cancer [264]. FDPS overexpression synergizes with *PTEN* deficiency in *PTEN* conditionally knockout mice in inducing prostate cancer development [264]. Biochemical studies have shown that FDPS exerts an oncogenic role in PTEN-deficien prostate cancer through GTPase/AKT axis [264].

Whole genome sequencing of seven cases of prostate cancer revealed that, in addition to mutational events occurring in the *PTEN* gene due to breakpoints, rearrangements disrupting the *MAGI2* gene, encoding for a PTEN-interacting protein, have been observed [53]. Knockdown experiments have shown that loss of *MAGI2* expression induces AKT phosphorylation; MAGI2 levels are decreased in prostate cancer and correlate with NKX3.1 level. This observation suggests that in prostate cancer multiple functionally recurrent mutations disrupt different gene in the PI3K/AKT pathway. MAGI2 levels are higher in HGPIN than in normal prostate or benign prostatic tissue [265,266]. MAGI2 level is decreased during prostate cancer progression and is a predictor of biochemical recurrence [267].

### 4.6. NKX3.1

NK3 homeobox 1 (NKX3.1) is a transcription factor expressed in the prostate epithelium essential for maintaining prostate cell fate and suppressing prostate cancer initiation. NKX3.1 is ubiquitously expressed at the level of luminal cells, but, following androgen deprivation, its expression becomes restricted at the level of a population of luminal stem cells, known as castration-resistant NKX3.1-expressing cells (CARNs). During prostate cancer initiation, NKX3.1 expression is frequently lost in both mouse models and naturally occurring human tumors. Understanding how NKX3.1 expression is regulated in vivo is of fundamental importance to better understand the mechanisms of prostate stem cell specification and cancer initiation. Androgens cell-autonomously activate NKX3.1 expression through androgen binding to the 11-kb region in both luminal cells and CARN cells [268]. In *PTEN^-/-^* prostate cancers, loss of NKX3.1 expression is mediated at the transcriptional level through the 11-kb region, despite functional androgen receptor is present in the nucleus of these cells [268].

Downregulation of the *NKX3.1* homeobox is considered a critical and frequent event in prostate cancer progression (*NKX3.1* gene copy losses in prostate cancer are much more frequent in castrate resistant disease than in localized disease). Loss of function of NKX3.1 in mouse prostate determines the downregulation of genes that are essential for prostate differentiation [269]. Gain of function of NKX3.1 in a fully differentiated nonprostatic mouse epithelium (seminal vesicle) induces a transdifferentiation to prostate in renal grafts in vivo [254]. In human prostate cells the biolofic activity of NKX3.1 requires its interaction with the G9a histone methyltransferase through the homodomain [269]. *NKX3.1* deficiency in mouse induces a late occurrence of prostate cancer development. Induction of prostatitis in *NKX3.1* mutant mice consistently accelerates prostate cancer initiation, favoring aberrant cellular plasticity and impairment of cellular differentiation [270].

Cancer progression is characterized by the progressive reduction of NKX3.1 level [271]. *NKX3.1* acts as a tumor suppressor for prostate cancer as supported by the observation that mice *NKX3.1* heterozygous and homozygous mutants display frequently prostate tumor formation. Interestingly, in the androgen-deprived prostate, NKX3.1 is expressed at the level of a rare population of prostate epithelial cells that seem the cells responsible for tumor development in some prostate mouse cancer models, and therefore is considered a marker of prostate stem cells. Experiments carried out in mouse prostate tumor cells indicate a role for NKX3.1 in DNA protection from oxidative-mediated damage. Analysis of the functional spectrum of possible NKX3.1 activities in the context of the biology of prostate cells suggests a role for this protein in the activation of a transcriptional program that maintains the differentiation status of luminal cells; disruption of *NKX3.1* gene contributes to prostate tumorigenesis by permitting luminal cell dedifferentiation [272]. Loss of *PTEN* causes reduced expression of NKX3.1 in prostate cancer; furthermore, restored NKX3.1 level counteracts prosurvival and proliferative effects of *PTEN* loss [272]. In addition to this effect, NKX3.1 expression induces increased p53 acetylation and half-life [273].

Prostate cancer development displays a strong genetic component. NKX3.1 is involved also in the mechanism of predisposition and aggressiveness of prostate cancer. Genome-wide association studies (GWAS) have identified rs11672691 at 19q13 associated with aggressive prostate cancer and with predisposition to prostate cancer [274,275]. This mutation is located on chromosome 19 within an intron of a non-coding RNA, *PCAT19*, much more expressed in prostate cancer than in normal prostate tissue [274,275]. The A to G mutation within this locus has two important biological consequences: it reduces the affinity for NKX3.1 [274] and creates a higher-affinity binding site for the transcription factor HOXA2 [275].

### 4.7. MYC

MYC oncogene is frequently overexpressed in prostate tumors as a consequence of either somatic amplifications (8q24, advanced prostate cancers) or as consequence of deregulated expression (prostate intraepithelial neoplasia). MYC expression levels, as well as PTEN status and Ki67 expression in primary tumor samples are strong predictors of progression-free survival, more accurately than clinical factors [276].

According to these findings it was suggested that *MYC* hyperexpression could play a relevant role in prostate cancer initiation. This hypothesis was supported by experimental studies showing that *MYC* hyperexpression in prostate epithelial cells determines the formation of prostate intraepithelial neoplasias and induces a downmodulation of NKX3.1 expression [277]. Various molecular mechanisms are responsible for Myc overexpression in prostate cancer cells. Myc protein stabilization seems to be a major mechanism contributing to Myc overexpression in this tumor. A mechanism of Myc stabilization could be related to the capacity of Myc protein to directly interact with Rho-associated kinase 1 (ROCK1): this interaction induces Myc phosphorylation, resulting in stabilization of the protein and activation of its transcriptional activity [278]. Some recent studies have elucidated peculiar mechanisms of prostate cancers overexpressing c-Myc. Thus, Priolo and coworkers have shown that prostate cancer associated with predominant MYC overexpression display dysregulated lipid metabolism, while tumors with predominant AKT1 activation were associated with accumulation of aerobic glycolysis metabolites [279]. In Myc-driven prostate cancers, EZH2 histone methyltransferase expression is induced and causes the repression of Interferon gamma receptor 1, with consequent loss of tumor suppressor signaling and reduced apoptosis due to the activation of this receptor. Therefore, the combination of EZH2 and IFN-gamma -targeted therapy could represent a potential strategy in the treatment of patients with advanced cancer driven by MYC [280]. *MYC* overexpression in prostate cancer is responsible for the increase of nucleolar number and size and of a nucleolar program of gene expression in prostate epithelial cells: these effects are in part mediated through the overexpression of fibrillarin, a *MYC* target gene; this protein is overexpressed in prostate cancer, as well as in HGPIN [281].

Prostate cancer cells, as well as the large majority of tumor cells, exhibit increased telomerase activity. *MYC* overexpression is a key driver of increased telomerase activity of prostate cancer cells, a conclusion supported by *MYC* gene overexpression in normal epithelial prostate cells or *MYC* knockdown in prostate cancer cells [282]. A recent study explored the interplay between MYC and AR signaling, providing evidence that MYC antagonizes AR transcriptional activity through a mechanism mainly involving co-occupation of a significant number of AR-binding sites, and that these sites exhibit enhancer-like characteristics [283]. These findings support the view that MYC overexpression deregulates the AR transcriptional program [283].

Few studies have explored the mechanisms responsible for MYC activation in prostate cancer cells. A recent study suggested that the endoplasmic reticulum stress/unfolded protein response (ERS/UPR) pathway, a survival mechanism activated by tumor cells to face an increased protein synthesis request in conditions of low oxygen availability, may play a major role in the activation of MYC expression in prostate cancer cells [284]. Furthermore, androgen signaling activates the IREα (inositol requiring enzyme 1α) arm of UPR [285]. Genetic inhibition of *IRE1α* or *XBP1* (X-box-binding protein 1, a direct target of IRE1α, activated by mRNA cleavage) inhibits prostate cancer growth in vitro and in vivo [284,285]. Interestingly, at the bq24 locus maps several prostate cancer susceptibility genes; however, despite the close proximity to the *MYC* locus, no direct association was found between 8q24 risk alleles and *MYC* expression in normal and tumor human prostate tissues [286].

BET proteins (BRD 2/3/4) act as critical coactivators of AR-mediated gene transcription and their targeting inhibits AR-mediated gene transcription and inhibits the prolifewration of CRPC. Treatment with BETY pharmacologic inhibitors, such as JQ1, determines suppression of c-MYC transcription. BET inhibition by JQ1 downregulates MYC transcription. BET inhibition by JQ1 downregulates *MYC* transcription, followed by genome-wide downregulation of MYC-dependent target genes [287].

In 2014, Assangani and coworkers provided evidence that JQ1 and BET-762, two selective small-molecule inhibtors targeting the amino-terminal bromodomaibns of BRD4, inhibit the proliferation of AR-competent CRPC cells, through a mechanism involving the inhibition of AR recruitment to target gene loci [288].

BET domain inhibitors are potentially interesting anti-CRPC for their capacity to suppress c-MYC. Recently, Coleman and coworkers, using a panel of CRPC cell lines, showed that suppression of MYC expression by BET bromodomnaion inhibition strongly correlates with sensitivity to BET bromodomain inhibitor JQ1. Cotargeting MYC together with bromodomainb inhibition in cells in which JQ1 fails to suppress MYC expression induce sadditive antitumor effects [289].

JQ1 is an attractive candidate for clinical applications but is limited by toxicity and off-target effects. This finding clearly indicates the need for developing new BET inhibitors with greater specificity. In this context, particularly interesting are the properties of a new class of BET bromodomain inhibitors able to induce BET degradation: these drugs bind both to E3 Ubiquiting ligase Cereblon and to BET proteins [290]. Interestingly, ARV-771—a small-molecule pan-BET degrader based on proteolysis-targeting chimera technology—displayed a consistently improved efficacy in cellular models of CRPC, compared to classical BET inhibition [290]. Particularly, CRBN-mediated BET degraders have increased specificity and efficacy in CRPC, as compared to traditional BET inhibitors or other types of degraders [291]. Importantly, treatment with the CRBN-mediated degraders on prostate cancer cells induces a pronounced inhibitory effect on both AR and MYC signaling axes and elicited a markedly inhibitory effect on cell growth and stimulatory on cell apoptosis [291].

Furthermore, BET inhibitors act as blocking competitors of the transcription factor GATA2, playing an essential role in the regulation of AR-Vs expression, and, through this mechanism, these drugs may have a potential role in the treatment of CRPC expressing AR-Vs [292].

### 4.8. RB and TP53

Recent studies have involved loss of the retinoblastoma (RB) tumor suppressor in prostate cancer progression. In fact, through the analysis of large sets of primary prostate cancer tumor samples it was provided evidence that RB expression was maintained in in situ prostate carcinomas, but it is lost in the majority of metastatic prostate cancers [293]. In line with these observations comparative genome hybridization studies showed that *RB* loss is preferentially observed in advanced prostate cancer [293]. A recent study explored intrapatient molecular heterogeneity in patients with metastatic castration-resistant prostate cancer: these patients displayed a high prevalence of *RB1* genomic aberrations, with structural variants, including rearrangements (such as intragenic tandem duplication), being common [67]; RB1 immunohistochemistry showed a heterogeneous expression in 28% of cases [67]. Intrapatients genomic and expression heterogeneity favor *RB1* aberrations as late, subclonal events increasing in frequency, due to treatment selective pressures [67].

Knockdown of *RB* in human prostate tumor cell lines did not affect the rate of tumor growth in xenografted mice under standard conditions; however, following castration, RB deficient prostate tumor cells displayed an accelerated tumor growth [294]. This intriguing behaviour of prostate tumor cells was due to marked increase of androgen receptor expression induced by *RB* loss and consequent expression of AR-target genes [294]. In mouse models, the loss of *RB* alone causes prostatic hyperplasia, but not prostate cancer; the combined *RB* and *TP53* loss induces prostate cancer formation. In a more recent study, it was shown that the combined loss of *RB* and A*kap12*, a gene shown to function as a metastasis suppressor in prostate cancer, results in prostatic intraepithelial neoplasia that fails to progress to prostate cancer [294]. Surprisingly, a high percentage of these tumors exhibited metastases to distant lymph nodes [294].

Other recent studies have further explored prostate cancer models involving *RB* inactivation. In one study, it was shown that *RB1* loss facilitates lineage plasticity and metastasis of prostate cancer initiated by P*TEN* mutation [295]. Additional loss of TP53 induces resistance to antiandrogen therapy [295]. The tumors developed in mice resembled the neuroendocrine variant of prostate cancer, expressing increased levels of epigenetic regulators *SOX2* and *EZH2* [295]. The study of a mouse model of androgen resistance showed that tumors can develop resistance by a shift from androgen receptor-dependent luminal cells to AR-independent basal cells [296]. This shift is enabled by loss of *RB1* and *TP53* function and is mediated by increased expression of *SOX2* [296]. Loss of *RB* function in multiple murine models of prostate cancer alters cytoskeletal organization, induces epithelial to mesenchymal transition and induces invasion and metastases [297].

TP53 mutations seem to play an important role in the transition from hormone-sensitive prostate cancers to CRPCs. In line with this view, *TP53* mutation is the most significantly enriched genetic event in CRPC compared to androgen-dependent prostate cancer [67]. Recent studies in various prostate cancer mouse models support an active role of *TP53* mutations in prostate cancer progression: (i) the loss of *TP53*, together with *PTEN* mutations and/or *RB1* loss, was found to shape cell lineage plasticity and reprogramming, thus promoting resistance to androgen deprivation [295,296], and (ii) loss-of-function of *TP53* promotes the development of castration-resistance in prostate cancer cells through two different mechanisms involving potentiation of androgen-independent cell growth and promoting genome instability [298].

*AR* and *TP53* are among the most frequent mutations observed in mCRPC, amounting to more than 60% and 50% of patients, respectively. The role of *TP53* mutations in stratifying mCRPC patients remains undefined. A recent study explored in a group of prostate cancer patients undergoing androgen deprivation therapy the possible impact of *TP53* mutations on patient prognosis, showing that only alteration in *TP53*, but not in AR, was an independent predictor of poor prognosis when compared with clinical covariates [299]. The *TP53* mutational status helped to stratify patients into good, intermediate and poor prognosis [299]. These findings were confirmed in another recent study showing that the percent of *TP53*-positive tumor cell nuclei by immunohistochemistry, a reliable measure of the presence of *TP53* abnormalities in tumor cells, associates with shorter time to biochemical relapse and higher incidence of metastatic relapse and of prostate cancer-related mortality [300]. Therefore, TP53 is a stratification factor and may help to identify patients who will mostly benefit from adjuvant therapy.

Interestingly, a recent study carried out on three large sets of tissue microarrays of prostate cancer patients, including intermediate- and high-risk tumors, of European-American and African-American ancestry, provided evidence that the presence of *TP53* missense mutations in these tumors was associated with increased T cell density [301]. Particularly, CD3^+^ and CD8^+^, but not FOXP3^+^ T cell densities, were significantly higher in tumors with TP53 nuclear accumulation, compared to those without [301]. These findings have potential implications for future immunotherapy studies [301].

### 4.9. LRF

*LRF* is a proto-oncogene displaying an important oncosuppressive role in the prostate. Prostate-specific inactivation of *LRF* determines a marked acceleration of *PTEN*-loss-driven prostate tumorigenesis, largely due to a bypass of *PTEN*-loss-induced senescence (PICS) [302]. *LRF* expression decreases during prostate cancer progression and LRF protein expression is lost in about 50% of advanced prostate cancers, particularly those characterized by *PTEN* loss (85%) [302]. Inactivation of *LRF* in vivo leads to RB downregulation, PICS bypass, and invasive prostate cancer [302]. LRF physically interacts with SOX9 and functionally antagonize the activiy of this transcription factor [302]. Loss of *LRF* in prostate cancer cells activates SOX9, a key driver of aggressive prostate cancer by promoting invasion, cell fate and cytoskeleton alterations and epithelial to mesenchymal transition [303]. SOX9 also drives prostate cancer development through WNT pathway activation; treatment of SOX9-expressing prostate cancers with a WNT synthesis inhibitor reduces WNT pathway signaling and tumor growth in murine xenograft models [304].

### 4.10. CDK12

A recent study provided evidence that prostate cancers bearing biallelic *CDK12* deletion may represent a peculiar tumor subtype. The occurrence of this abnormality is more frequent in metastatic (5–7%) than in primary (1%) prostate cancer [305]. *CDK12*-mutant prostate cancers are baseline diploid and have an excess of focal tandem duplications; furthermore, these tumors have a peculiar transcriptional profile [290]. The *CDK12* gene was related to control of genomic stability; however, large-scale copy number gains were frequent in the *BRCA2*- and *ATM*-deficient cases, compared to *CDK12*-mutant cases [305]. *BRCA2* and *ATM* mutated, as well as *ETS*-fusion-positive prostate cancers have the highest percentage of gains, while the majority of *CDK12*-mutant tumors do not have any change in ploidy [305]. A peculiar molecular property of these tumors consisted in a high number of focal tandem duplications (ITD): these molecular events result in highly recurrent gains of genes involved in the cell cycle and DNA replication [305]. FTDs induce increased gene fusions, associated with elevated neoantigen burden (originated from fusion-induced chimeric open reading frames) and elevated tumor T cell infiltration/clonal expansion [305]. In fact, CDK12-variant tumors display higher overall levels of T cell infiltrating lymphocytes and greater numbers of expanded T cell clones than other prostate cancer subtypes; furthermore, these tumors display also increased expression of some chemokines and their receptors [305]. Preliminary data from four prostate cancer patients with *CDK12*-altered prostate cancers showed in two of these patients, response to treatment with the immune checkpoint blockade therapy [305].

### 4.11. PLZF

The promyelocytic zinc finger (*PLZF*), also known as *ZBTB16* (zinc finger and BTB domain containing 16) is a transcription factor involved in the control of many biological functions, including cell proliferation and differentiation, stem cell maintenance, and innate immune cell development. In several tumor cell types, including prostate cancer, PLZF was implicated in tumor progression as a tumor suppressor [306]. A recent study reported recurrent (15% of cases) homozygous deletions of the androgen-regulated PLZF transcription factor [67]. PLZF seems to play an important role in androgen resistance of prostate cancer cells. PLZF makes part of a molecular circuit involving KLK4 (Kallikrein related peptidase 4), PLZF, AR, and mTOR: KLK4 interacts with PLZF and decreases its stability; PLZF in turn interacts with AR and inhibits its function as a transcription factor and activates the expression of an inhibitor of mTORC1 [307]. Thus, this activity regulates both AR and PI3K signaling [307]. Studies on the expression of PLZF in prostate cancers showed a loss of expression in 26% high-grade primary prostate cancers and in 84% of metastatic prostate cancers [308]. Experiments of enforced *PLZF* gene expression in prostate cancer cells provided functional evidence that knockdown of *PLZF* expression promotes a CRPC phenotype; reintroduction of *PLZF* expression was sufficient to reverse androgen-independent growth mediated by PLZF depletion [309].

Intriguingly, PLZF expression is positively regulated by androgens: androgen deprivation therapy is able to induce a decrease of PLZF expression, a condition that could have a detrimental effect on the growth of prostate cancer cells [309].

A recent study explored, in detail, PLZF expression in prostate cancers showing that PLZF mRNA expression was lower in tumors with *PLZF* deletions; there was a strong, positive association between intratumoral AR signaling and PLZF expression; PLZF expression was lower in tumors with PTEN loss; low *PLZF* expression was associated with higher MAPK signaling; and patients with clearly low *PLZF* expression were the most likely to develop lethal prostate cancer, independently of clinicopathologic features [310]. These observations support the conclusion that low expression of PLZF is associated with a worse prognosis in primary prostate cancer.

## 5. Racial Influences on Prostate Cancer Genomics

Some recent studies have explored the occurrence of molecular differences in prostate cancer between African American (AAM) and Caucasian Men (CaM). In this context, Khani and coworkers recently reported a higher frequency of *ERG* rearrangements (42.5% vs. 27.6%), *PTEN* deletion (approximately 20% vs. 7%), and *SPOP* mutation (10% vs. 4.5%) in CaM than in AAM [311]. In contrast, *SPINK1* overexpression is higher in AAM prostate cancers (~24%) than in CaM cancers (about 8%) [311]. SPINK1 is similar on a structural point of view to EGF and activates the EGFR on the surface of prostate cancer cells, inducing their growth [312]. *SPINK1* was overexpressed in an aggressive subtype of *ETS*-negative prostate cancers [312]. However, a recent study, confirmed that *SPINK1*-positive prostate cancer is more prevalent in AAM than CaM, but, contrary to previous observations, failed to show any significant association with worse pathologic or oncologic outcomes after radical prostatectomy in either AAM or CaM [313].

Petrovics et al. compared the genomic profile of localized prostatic cancers between CaM and AAM and identified a recurrent deletion of LSAMP (Limbic System Associated Membrane Protein) gene on chromosome 3q13.31, prevalent in prostate cancers of AAM compared to CaM (26% vs. 7%); this gene deletion was associated with rapid disease progression [314]. A high frequency (20%) of LSAMP deletion was reported also by Ren and coworkers in a Chinese cohort of prostate cancer patients [314].

Lindquist and coworkers have analyzed the genomes of 24 AAM with aggressive prostate cancer with Gleason grades ≥7 and confirmed in these patients a low prevalence of TMPRSS2-ERG fusions (20%) and of PTEN deletions (8%) [315]. Interestingly, this study identified a novel gene fusion involving CDC27-OAT (17%) and copy number alterations involving the amplification of *MIR6723*, *PCBD2*, and *TXNDC15*, and the deletion of *EBF2* [315].

A genomic analysis of localized prostate cancers on a group of 102 AAM cases led to the identification of recurrent loss-of-function mutations of *ERF*, an *ETS* transcriptional repressor, in 5% of cases [119,316].

These studies have shown an overall tumor mutational burden (TMB) similar in localized prostate cancers of CaM and AAM patients [312,315]. However, recent analysis by whole genome sequencing of high-risk localized primary prostatic cancer in African men (AM) showed an elevated tumor burden, with a 1.8-fold increase in small somatic variants in African versus European-derived tumors [317]. Furthermore, an increase in oncogenic driver mutations in African tumors was observed [317]. TM*PRSS2-ERG* or any *ERG* fusions were absent among African prostate cancers [317].

Recently, Tonon and coworkers reported the mutational profile of 25 localized prostate cancers from African Carribean men, and compared it to the findings observed in 15 French Caucasian prostate cancer patients. African Carribean tumors were characterized by the more frequent deletion at the level of 1q41–43, encompassing the *PARP1* gene, involved in DNA repair and a higher proportion of intrachromosomal rearrangements, including duplications associated with *CDK12* truncating mutations [318]. Transcriptomje studies show an overexpression of genes involved in AR activity in African Carribbean prostate cancers and of PVT1, a long non-coding RNA located at 8q24 [318].

The prostate cancer-related mortality is 2–3 times higher in AAM compared to CaM [319]. Recent studies based on the analysis of prostate cancer patients according to the Gleason score, the best independent predictor of prostate cancer outcomes, have shown that black AA patients with Gleason 6 score had a higher risk of prostate cancer death compared with nonblack patients, while no significant difference was observed with Gleason 7 to 10 Disease [320,321]. These observations suggest that AAM have a greater tendency to develop tumors with aggressive phenotypes. Gene expression studies have also supported the aggressiveness of AAM prostate cancers. Thus, six biomarkers, including ERG, AMACR, SPINK1, NKX3.1, GOLM1, and AR predict higher aggressiveness of AAM than CaM prostate cancers; then, CaM had triple-negative (ERG-negative/ETS-negative/SPINK-1-negative) disease (51% vs. 35%) [322]. Another recent study provided evidence that dysregulated gene expression predicts tumor aggressiveness in AAM prostate cancers: this study identified 362 differentially expressed genes in AAM involved in regulating signaling pathways associated with tumor aggressiveness. In PCA tissues, NKX3.1, APPL2, TPG5, ALDH1A3, and AMD1 transcripts are significantly upregulated. Immunohistochemistry studies confirmed the overexpression of TPD52 and LTC45 in AAM compared to CaM prostate cancer [323].

The Shared Equal Access Regional Cancer Hospital (SEARCH) Database study group recently performed two studies on a large set of AAM and CaM undergoing radical prostatectomy for prostate cancer. Among men undergoing radical prostatectomy at equal-access centers, alhough black men had an increased risk of biochemical recurrence, they had simila risks of aggressive disease, recurrence, metastasis, and prostate cancer-related death compared with white men, and the risk of biochemical recurrence was similar after taking into account risk parameters [324]. In the second study, it was shown that among men who received ADT post-biochemical recurrence after radical prostatectomy, racial differences were not a predictor element of metastases or other adverse events [325]. The findings of these two studies support the view that racial differences in prostate cancer mostly rely on early stages of disease development.

## 6. Gene Expression Profiling Studies

D’Amico and coworkers in 1998 developed a classification system of prostate cancer based on clinical parameters (PSA, Gleason score and clinical staging) to group these patients into low-, intermediate-, and high-risk of relapse after therapy with curative intent [326]. The most commonly used system to simple classify the clinical risk of prostate cancer patients is the simplified tghree-tiers (slow-, intermediate-, and high-risk) NCCN (National Comprehensive Cancer Network, Fort Washington, PA, USA) risk groups, based ongrouping of pretreatment PSA, Gleason Score, and clinical stage. A great limitation of this classification derives from the lack of integration with genomic data, which could be useful to better stratify the patient risk and to define a more personalized risk assessment.

In order to improve the capacity of the PSA screening to predict the evolution of prostate cancer, a number of active surveillance strategies, mainly based on gene expression studies, have been introduced for low-risk prostate cancer patients. The introduction of these additional evaluations of early prostate cancers is important because the simple PSA screening leads to overdiagnosis of individuals needing aggressive treatment [327]. Active surveillance of low-risk patients imply multimodal serial monitoring, involving also a number of biopsy tissue based on initial research studies, and then on commercial kits derived from these studies: Myriad Prolaris Cell Cycle Progression (CCP) score, based on RNA expression signature derived from cell cycle proliferation genes [328,329]; Oncotype DX Genomic Prostate Score, based on a 17-gene assay [330]; and Genome DX Decipher Genomic Classifier, based on a genomic classifier [331]. Decipher is 22-gene genome classifier and its capacity to predict clinical outcome was supported by several clinical studies. Thus, the genomic classifier predicted metastasis on multivariate analysis in a high-risk population after radical prostectomy: the cumulative incidence of metastasis at 5 years after radical prostectomy was 2.4%, 6.0%, and 22.5% in patients with low, intermediate, and high classification scores, respectively [332]. The Genomic Classifier Scoring system, together with clinical nomograms, allows the identification of patients most at risk for rapid metastatic progression [333]. The Decipher Genomic Scoring system improves post-prostatectomy risk stratification in cohorts of intermediate–high risk men (decipher index correlated with increased cumulative incidence of biochemical recurrence, metastasis and prostate cancer-related mortality) [334]. The meta-analysis of five studies, involving a total of 975 patients, including 855 patients with individual patient-level data, showed tha Decipher Genomic improves the prognostication post-prostectomy, as well as within all clinicopathologic, demographic, and treatment subgroups [335].

Prostate cancer is heterogeneous and multifocal, with the presence of multiple, genomically independent tumors identified in ~80% of patients undergoinhg radical prostectomy for localized prostate cancer. This heterogeneity represents a potential problem and an impediment for the development of single bioipsy-based prognostic biomarkers and for the development of of precision medicin approaches based on single biopsy analysis. This problem is particularly challenging in low-grade prostrate cancers, due to the limited volume of cancerous biopsy tissue available for analysis. The problem of multifocality of prostate cancer is very relevant and challenging because recent studies have shown transcriptional differences between multifocal low- and high-grade prostate cancer [37]. Salami and cowiorkers have directly analyzed the impact of tumor multifocality on the tumor scoring using the most frequently used gene expression prognostic signatures, including Cell Cycle Progression, Oncotype DX Genomic, and Genome DX Decipher Genomic Classifier systems, and showed that low-grade tumor foci present in tumors classified as low-risk have a similar gene expression profile if are present in prostates with or without coexisting higher grade tumor foci [40,336]. This finding demonstrates that prognostic RNA expression assays performed on low-grade prostate cancer specimens may not provide appropriate information on the presence of coexisting unsamplked aggressive tumor foci [40,336]. A recent study by Cooperberg and coworkers evaluated the heterogeneitybin terms of clinical charactereistics and of genomic risk scores in a large group of low-risk prostate cancer patients. These cases were compared to a large group of high-grade prostate cancer patients. Average genomic risk (AGR) was determined from 18 published prognostic signalutres and resulted to be associated with pathological and biochemical outcomes [337]. In contrast, an unsupervised clustering analysis of the hallmark gene set scores, that were enriched for Luminal A, Luminal B and Basal subtypes, but was not related to patient’s outcomes [337]. These observations support the usefulness of the genomic characterization of low-risk prostate cancer patrients that can help to stratify these patients into those amenable only to surveillance and those requiring immediate treatment [319]. Importantly, the integration of a commercially available genomic classifier in combination with standard clinicopathologic variables generates an easily usable clinical-genomic risk styratification suitable to identify prostate cancer patients at low-, intermediate-, and high-risk for metastasis [338]. This system could be integrated into current guidelines to better stratify patients according to risk of disease progression [338]. Zhao and coworkers have reported the development of a 24-gene signature predictive of response to postoperative radiotherapy in prostate cancer: patients with high PORTOS (Postoperative Radiation Therapy Outcome Scores) had a lower incidence of distant metastasis than did patients who did not have radiation therapy (at ten years, 5% vs. 63% of metastasis rate, respectively) [339]. Importantly, the conventional prognostic tools Decipher and cell cycle progression signature were unable to predict response to radiotherapy [339].

Van Eeden and coworkers reported the impact of a biopsy-based 17-gene genomic prostate score, based on the gene expression Oncotype DX Prostate Cancer assay, to predict prostate cancer and prostate cancer death in surgically-treated men with clinically localized disease [340]. The test provides a Genomic Prostate Score (GPS) result, with a score from 0 to 100, with increasing scores indicating more aggressive disease [340]. This test has been validated as a strong, independent predictor of adverse pathology and biochemical recurrence after radical prostectomy in men with low- and intermediate-risk prostate cancer [340,341]; furthermore, GPS is a strong, independent predictor of long-term uoutcomes in clinically localized prostate cancer treated with radical prostecxtomy [340].

Other investigations have explored the profile of gene expression in prostate cancer with the specific aim of identifying new classification and prognostic criteria and genes or metabolic/signaling pathways that could be targeted. Initial studies based on the exploration of the expression of 26,000 genes have led to the identification of three tumor subtypes: subtype I is characterized by a gene expression pattern resembling in part that expressed in normal prostate and by a low recurrence, and subtypes II and III exhibit, in part, a similar pattern of gene expression, including genes related to the extracellular matrix, cell proliferation, and metabolic activity. High-grade advanced stage and metastatic tumors were mostly represented in subtypes II and III [342]. The two genes whose expression mostly predicted prognosis were *A2GP1*, a zinc alpha-2-glycoprotein whose expression characterized tumors of the subtype I, associated with low/mild aggressive development: *MUC1*, encoding the Mucin1 transmembrane protein, whose expression was observed in subtypes II and was associated with aggressive clinopathological features [342]. A subsequent study of the same authors was devoted to analyze the occurrence of genomic alterations in the three prostate cancer subtypes: 6q15 and 5q25 deletion was associated with subtype-1 gene expression pattern; subtype-2 was characterized by deletions at 8p21 (*NKX3-1*) and 21q22 (resulting in *TMPRSS2-ERG* fusion); and subtype-3 displayed frequent DNA copy number alterations and, particularly gains at 8q24 (*MYC*) and 16q13, and losses at 10q23 (*PTEN*) and 16q23 [343]. According to these findings it was suggested that prostate cancers develop through a limited number of genetic pathways [343]. More recently, Markert et al. [344] have reexplored the expression signature of prostate cancers, with the specific aim of identifying a cancer subset, characterized by an “embryonic-like” expression pattern. Particularly, these authors have classified prostate cancers according to their mRNA microarray signature profiles indicating stem cell expression patterns (stemness), inactivation of the tumor suppressors *PTEN* and *TP53, TMPRSS2-ERG* fusion, and activation of some oncogenic pathways [344]. Unsupervised clustering identified five tumor subsets: a group of tumors displayed stemness-like signatures, associated with *PTEN* and *p53* inactivation and had a very survival outcome; a second group is characterized by *TMPRSS2-ERG* fusion and is associated with intermediate survival outcome; three other groups were associated with benign outcome [344]. Ben-Porath et al. [345] introduced a peculiar approach for the gene expression microarrays of human tumors by studying the concordant expression of sets of genes known to be over- or underexpressed in embryonic stem cells and demonstrated that this embryonic signature corresponded to a subclass of breast cancers. This approach allowed identification of a subgroup of prostate cancer manifesting stem-like signature, associated with *p53* and *PTEN* inactivation and very poor survival outcome [345]. In a more recent study Rye and coworkers have analyzed gene signatures in a group of prostate cancer patients and showed the existence of only two groups of tumors: the poor prognosis group was characterized by enrichment in embryonic stem cell, *ERG*-fusion, and MYC-rich signatures; the other group was associated with good prognosis [346].

Molecular studies have led to the characterization in mCRPC patients of a subgroup comprising about 10% of cases with an overexpression of the *SPINK1* gene, in association with negativity for *ETS* fusions [312]. *SPINK1* overexpression is associated with a negative prognosis in patients with mCRPC, but not in those with localized disease [347]. The mechanisms responsible for *SPINK1* overexpression are unclear, but it evident that it is not related to chromosomal rearrangement, deletion or amplification. A recent study based on transcriptome analysis of prostate cancer tissues showed that *SPINK1* overexpressin prostate cancers are characterized by a peculiar gene expression signature, involving the expression of genes typically expressed in the gastrointestinal tract, such as albumin gene [348]. The gastrointestinal gene signature was orchestrated by the transcription factors *HNFG4* and *HNF1A*; the induction of *HNFG4* and *HNF1A*-mediated pathway is required to sustain the proliferation of *SPINK1*-overexpressing prostate cancer cells, independently of AR signaling [348]. A recent study provided evidence that *SPINK1* overexpression in CRPC may be related to an epigenetic mechanism involving downmodulation of two miRs, miR-338-5p and miR-421, targeting *SPINK1* [349]. *SPINK1*-oveexpressing prostate cancer displays reduced miR-338-5p and miR-421 expression; enforced expression of miR-338-5p and miR-421 in SPINK1-overexpressing prostate cancers induces increased *EZH2* expression; functional studies support a role for *EZH2* as a mediator of epigenetic silencing of miR-338-5p and miR-421 ]349]. miR-338-5p and miR-421 expression can be repristinated in *SPINK1*-overexpressing prostate cancers through epigenetic drugs affecting *EZH2* expression or through synthetic mimics [349].

A basal population within the human prostate possess stem cell feature: this cell population is able to generate all three epithelial populations of prostate and act as tumor-initiating cells though enforced expression of some oncogenes commonly altered in prostate cancer. Based on these observations, Smith and coworkers isolated Trop2^+^ CD49f^high^ human basal prostate cells and identified in these cells a basal stem cell signature: metastatic prostate cancer was enriched for this signature [350]. Using a dataset comprehensive of different metastatic prostate cancer phenotypes, showed that metastatic small cell carcinoma was the most enriched for this signature [350]. Small cell prostate carcinoma is a prostate cancer subtype with neuroendocrine differentiation properties, rare among organ-confined prostate cancers (~1%), but frequent among metastatic prostate cancer (20–25%).

Carcinomas originate from epithelial tissues, which have basal and apical (luminal) prientations. Consequently, tumors originating from these tissues may exhibit a preferential luminal or basal differentiation. Understanding of the biological features related to “luminal-ness” or “basal-ness” of the epithelial tissues is important because it may affect the prognosis and response to treatment of these tumors. The *PAM50*, gene expression classifier which has been used to group breast cancers into Luminal A, Luminal B, Basal, and Her2-like subsets; the luminal breast cancer subtypes express higher levels of estrogen receptor and progesterone receptor and are more responsive to hormonal therapy [351]. Zhao and coworkers have used this classifier system analyzed 3782 localized prostate cancer samples and found that these tumors clustered among three different groups: Luminal A, Luminal B, and Basal: the basal tumors displayed the typical CD49f signature, while the luminal tumors were enriched in luminal markers, including KRT1B, NKX3.1, and AR [212,352]. Furthermore, these investigators explored also the predictive role of the classifier for sensitivity to androgen deprivation therapy, showing that Luminal B subtype, but not Luminal A and Basal subtypes did not benefit from androgen deprivation therapy [352].

The analysis of gene expression datasets showed that prostate cancers can be subdivided into three distinct subtypes: PCS1–3; PCS1 and PCS2 cancers reflect luminal subtypes, while PCS3 corresponds to a basal subtype [351]. Importantly, PCS1 tumors progress more rapidly to metastasis than PCS2 or PCS3 [353]. The PCS1 exhibits high activation scores for *EZH2*, *PTEN*, *PRF* (cell proliferation), ES (stemness), AV (neuroendocrine differentiation), and AR-V pathways; PCS2 group is characterized by high activation of *ERG*, *AR*, *FOXA1*, and *SPOP1* pathways; the PCS3 group is characterized by high activation of RAS, PN (proneural), MES (epithelial–mesenchymal transition) pathways [353]. PCS1 and PCS2 were characterized by expression of luminal genes, such as *EZH2*, *AR*, *MK167*, *NKX3.1*, *KLK2/3*, and *ERG*, while PCS3 show expression of basal genes, such as *ACTA2*, *GSTP1*, *IL6*, *KRT5*, and *TP63* [353].

Given the key role of the development of castration resistance during prostate cancer progression, the study of the expression of AR-responsive genes was of considerable interest to better define subsets of prostate cancers and to predict their progression to androgen independency. These studies were triggered by an initial important finding obtained by Wang and coworkers, showing that the role of AR in androgen-independent prostate cancer is not to direct androgen-dependent gene expression program without androgens, but rather to induce and maintain a different program inducing androgen-independent cell growth [354]. A subsequent study performed a comprehensive analysis of AR binding sites: some of these sites identified in untreated prostate cancers are lot in tumor-responsive samples, and a part of which was regained with the development of castration-resistant disease [355]. The tissue-specific ARBS identified in CRPCs are associated with in vivo-regulated genes and converge on distinct transcription factor networks [355]. Overlapping ARBS and histone marks in CRPCs allowed to identify a 16-gene signature enriched in STAT, MYC, and E2F binding sites; this gene signature showed the capacity of monitoring disease progression and identified potential targets for therapeutic intervention [356]. Agonist-liganded human AR and antagonist-liganded AR bind to two distinct different motifs, leading to distinct transcriptional outcomes in prostate cancer cells [357]. This approach allowed to identify AR binding sites that differentiated normal prostate tissue from cancer, associated with onset and progression of prostate cancer [357].

Aberrantly expressed proteases are important biomarkers for many tumors, including prostate cancer. PSA is a protease in the kallikrein family (KLK3), regulated by androgens; KLK2, another member in this family, is another very promising biomarker for prostate cancer. A recent study provided a new classification of prostate cancer using a protease library; in fact, Dudani and coworkers have identified a panel of prostate cancer proteases through transcriptomic and proteomic analysis and, using this panel, it was developed a nanosensor library that measures protease activity in vitro using fluorescence and in vivo using urinary readouts [358]. This nanosensor library was able to classify aggressive prostate cancer [358].

The detection of membrane-associatedd prostate cancer antigens is of fundamental importance also for the development of in vivo imaging or for therapeutic targeting of prostate cancer cells. In this context, Gallium-68 prostate-specific membrane antigen (PSMA) positron emission tomography (PET) has been developed and increasingly used for accurate staging of high-risk localized, advanced, and metastatic prostate cancer [359]. The meta-analysis of 37 clinical studies carried out using this agent supported its high bsensitivity and specificity in in vivo detecting prostate cancver [359]. Interestingly, for PSA categories 0–0.19, 0.2–0.49, 0.5–0.99, 1–1.99, and ≥2 ng/mL, the percentages of positive scans were 33%, 45%, 59%, 75%, and 95%, respectively [359]. Lee and coworkers have integrated transcriptomic and cell-surface proteomic data generated from a panel of prostate cancer cell lines to identify cell-surface markers associated with prostate cancer adenocarcinomas and neuroendocrine prostate cancers, respectively [360]. CECAM5 appeared to be a promising target for cell-based immunotherapy [360].

A recent study provided evidence that a molecular portrait of epithelial–mesenchymal transition (EMT) in prostate cancer cells is associated with clinical outcome. [343]. EMT is a complex set of phenotypic changes that contribute to cancer progression and therapy resistance. It is a dynamic process, inbvolving plasticity of tum or cells able toundergo EMT or MET. The presence of EMT in primary tumor cells is very difficult to define and the detection of simple mesenchymal markers (i.e., presence of E-casdherin and absence of vimentin) are not sufficient to demonstrate the presence of prostate cancer cells that bhave undergone an EMT. To bypass these limitatrions, Stylianou and coworkers have identified a transcriptional profiling of prostate cancer cells oscillating between EMT and MET. This gene signature identified patients with poor prognosis in primary prostate cancer [361].

Although RNA analysis studies have contributed to identify prognostic groups of prostate cancer patients and to predict disease recurrence, the integration of genomic profiling studies (genetic and transcription data) may be of some help to predict prostate cancer patients’ overall risk. Thus, Taylor and colleagues have shown that three important biochemical pathways, RAS/RAF, PI3K, and RB1, exhibit alterations in 34–43% of primary tumors and 74–100% of metastatic tumors [79]. Importantly, in this study it was shown that DNA copy number profiling was significantly associated with clinical outcome [79]. In this analysis, they identified a subgroup of patients with good prognosis associated with tumors that do not carry any somatic gene copy number alteration or aneuploidy [79]. Using a multicolor fluorescence in situ hybridization approach, the copy number alterations of six genes known to be involved in aggressive prostate cancer (*PTEN*, *MYC*, *MEN1*, *PDGFB*, *CTTNBP2*, and *TBL1XR1*) was explored in a group of recurrent and in a group of nonrecurrent prostate cancer patients [362]. *PTEN* loss and *TBL1XR1* gain were the most two frequent aberrations observed in progressors, detecting 86% of these patients [362]. *PTEN* loss is probably the single most reliable genetic alteration associated with prostate cancer progression: inactivation of *PTEN* by mutation or deletion was identified in ~20% of primary prostate cancers at radical prostectomy and in as may as 50% of castration-resistant prostate cancers [363]. A study based on more than 7000 patients undergoing radical rpostectomy showed that GS6, GS7 (3 + 4), GS7 (4 + 3), and GS8 showed *PTEN* loss in 14%, 21%, 38%, and 41%, respectively [364]. PTEN protein loss by immunohistochemistry predicts upgrading of GS6 on biopsy to GS7 [365]. Therefore, *PTEN* loss is a genetic marker to distinguish indolent from aggressive disease in patients with clinically localized prostate cancers [363].

Recently, Tomlins and coworkers have performed a study of gene expression profiling of 1577 prostate cancers representing integration between gene expression and gene alterations and showing that they can be classified into four molecular groups: 45% as ERG^+^, 9% as ETS^+^, 8% as SPINK1^+^, and 38% as triple-negative [366]. Multivariate analysis showed that ERG^+^ tumors were associated with lower preoperative prostate-specific antigen and Gleason scores, but higher extraprostatic extension; ETS^+^ tumors were associated with seminal vesicle invasion; finally, SPINK1^+^/triple-negative tumors displayed higher Gleason scores [366]. However, in spite of these differences, clinical outcomes were not significantly different in these three different tumor subtypes [366]. The identification of these molecular subgroups, although not associated with patient’s outcome, may be useful to identify subgroups of patients amenable to different therapeutic approaches. Thus, a recent study showed in a group of patients with metastatic castration-resistent prostate cancer treated with docetaxel chemotherapy, those with ERG^+^ primary tumors have a two times increased risk of therapy resistance than those with ERG^-^ tumors [367]. Furthermore, preclinical studies support the targeting of PARP in ERG^+^ or ETS^+^ tumors and targeting of EGFR in SPINK1^+^ tumors [367].

The study of copy number alterations allowed to identify four tumor prostate cancer subtypes with prognostic implications: subtype 1 characterized by gain of chromosome 7; subtype 2 characterized by loss of 8p and 16q (subtype 2 and 3 share many common genetic alterations); and subtype 4 is called the tumor subtype of so-called quiet genomes, due to the presence of few genetic alterations. Patients with subtype 4 have a significantly better prognosis than those of the other tumor subtypes; subtype 1 had an intermediate prognosis and subtypes 3 and 4 displayed a similar prognosis less favorable than the other two tumor subtypes [368]. Importantly, this study showed that the evaluation of the percentage of genome alteration (PGA) was a strong prognostic parameter: patients with a low PGA score have a markedly better prognosis than those with high PGA score [368]. The evaluation of PGA together with the evaluation of a hypoxia signature still improved the prognostic stratification of prostate cancer patients [368].

A recent study provided clear evidence that in localized prostate cancers the monoclonality or polyclonality of genetic alterations, including CNAs, is the major determinant of tumor progression, in that moinoclonal tumors rarely relapse, while polyclonal tumors frequently relapse [63].

Recently, Stelloo and coworkers have performed a strudy based on the integration of genetic information together with epigenetic and gene expression data and stratified patients accordingly [369]. Integrative molecular subtyping allowed to identify three major subtypes of which two were *TMPRSS2-ERG* dictated, while a third subtype was characterized by low chromatin binding and activity of AR, high activity of FGF and WNT signaling, positivity for neuroendocrine genes, negativity for genes characteristic of poor-outcome associated luminal B-subtype and low mutational burden [369].

Studies on metastatic prostate cancers have reported the similarity of tumors and metastases from the same patient [74], thus supporting the rationale of investigating agents capable of targeting metastatic progression in advanced prostate cancer. The investigation of metastatic progression in a genetically engineered mouse model of prostate cancer was used for the isolation of tumor and metastic cells and for the definition of a molecular signature of metastasis progression; particularly, cross-species computational analyses, comparing a mouse signature with a comparable human signature of metastatic prostate cancer identified master regulators of the metastatic process, highly enriched for genes that are predicted to function as regulators of the epigeneome, including modifiers of DNA and histone, or remodeling chromatin architecture [370]. The analysis of a gene signature based on eight metastasis regulator genes predicted disease outcome in different cohorts of prostate cancer patients [352]. Among the various metastasis-related genes, the highest level of metastasis-related activity was observed for the histone methyltransferase, Nuclear Receptor Binding SET Domain Protein 2 (*NSD2*), robustly expressed in lethal prostate cancer [370]. *NSD2* silencing inhibited metastasis of mouse allografts in vivo [370].

Proteomic and phosphoproteomic analyses have the potential to contribute to a better understanding of the molecular pathways involved in prostate cancer development and progression, identification of new disease biomarkers and definition of kinase signaling networks present in prostate cancer cells at various stages of development [371,372]. The informations related to proteomic and phosphoproteomic studies are complementary to those originated by geniomic and transcrptomic approaches [371,372]. Proteomic and phosphoproteomic technologies have made significant progresses in the last years, with the development of basically two different platforms: platforms based on mass spectrometry and platforms based on antibodies [371,372]. Global discovery proteomics and phosphoproteomics are predominantly performed using liquid chromatography–tandem mass spectrometry coupled with data-dependent and data-independent acquisition [371,372]. However, many technical obstacles must be bypassed before the proteomic and phosphoproteomic technologies can be introduced in routine clinical practice as a support for identification of the optimal patient’s therapy.

A number of studies was focused to analyze phosphoproteomics in prostate cancer. In a fisrt study, Drake and coworkers explored the tyrosine phosphoproteome in mCRPC and identified several activated tyrosine kinases (EGFR, SRC, ALK, MAPK 1/3, and RET) in these tumors, showing interpatient heterogeneity, but consistent similarity at the level of metastatic sites within the same patient [373]. This finding is important because supports the potential clinical utility of phosphoproteomic analysis based on a single metastatic biopsy [373]. In a second study, the same authots have performed an integrated analysis of phosphoproteomic data with genomics and transcriptomica. Using this approach, six major phosphorylation pathways were enriched in CRPC tumors after incorporation with phosphoproteomic data [374]. The need of an integration approach including phosphoproteomic analysis clearly shown by some pathways such as AKT/mTOR/MAPK and cell cycle cycling were found to be enriched in mCRPC only when phosphoropeomic analysis was included [374]. It is im portant to point out that using this approach every evaluate patient displayed at least four phosphorylation hallmarks making the prioritization of kinase pathway very difficult; however, the inclusion of an analysis with targeted kinase inhibitors allowed to stratify individual patients for patient-specific kinase hierarchies [374]. This analysis showed two remarkable findings: (a) not all patients with the same cancer hallmark pathway are predicted to display the same response to kinase inhibitors and (b) in the majority of patients the cell cycle pathway is prominent, thus suggesting that CDK4/6 inhibitors may have clinical efficacy and could be used in combination with other antitumor agents [374].

Various studies have explored the proteome of prostate cancer. An initial study by Iglesias-Galo et al. explored the genome-scale proteomic profile of primary prostate cancer [375]. Tumor tissues, compated to normal prostatic tissues, exhibited elecvated expression of proteins involved in multiple anabolic processes, including protein and fatty acid synthesis, ribosomal biogenesis and protein secretion [357]. Some proteins resulted to be overexpressed in prostate cancers, including carnitine palmoyltransferase 2 (CPT2), coatomer protein complex, subunit alpha (COPA), and mitogen- and stress-activated protein kinase 1 and 2 (MSK ½) [357]. The same authors explored the proteomic profile of bone metastatic prostate tumors from 22 patients, showing a proteomic pattern more heterogeneous than thet observed in primary tumors, associated with increased expression of proteins involved in cell cycle progression, DNA damage response, RNA processing and fatty acid β-oxidation, and reduced expression of proteins involved in cell adhesion and carbohydrate metabolism [376]. Interestingly, within bone mestastases, two phenotypic subgroups were observed: BM1, expressing higher levels of AR targets, mitochondrial and Golgi apparatrus-resident proteins; BM2, associated with increased expression of proliferation and DNA repair-related proteins [77,376].

Latonen and coworkers reported the first integrative proteomic analysis through high-throughput mass spectrometry of tumor biopsies derived from benign prostatic hyperplasia (BPH), primary prostate cancer, and CRPC [377]. Each sample group displayed a distinct protein profile [377]. The integrative analysis provided evidence that gene copy number, DNA methylation, and RNA expression do not predicteds the observed proteomic changes observed in prostate cancer tissues [377]. Particularly, the study of the proteomic profile allowed to identify previously unrecognized molecular and pathway changes, such as those corresponding to two metabolic shits at the level of the citric acid cycle (TCA cycle): a first shift occurs during the progression of BPH to localized prostate cancer and involves the upregulation of most of TCA enzymes and, particularly, of aconitase 2 (ACO2); a second shift occurs during the progression of localized prostate cancer to CRPC and is charactewrized either by the maintenance of high levels of some enzymes (citric synthase (CS) and fumarate hydratase (FH)) and downregulation of other enzymes (ACO2, oxoglutarate dehydrogenase, OGDH, and succinate-CoA ligase alpha subunit, SUCLG1) [377]. A notable exception is represented by the enzyme malate dehydrogenase 2 (MDH2), whose levels continue to increase during the second shift, an event probably related to decreased expression of miR-22 and miR-205 [377].

A recent study reported the profiling of the genomes, epigenomes, transcriptomics and proteomes of localized, intermediate-risk prostate cancers [378]. This analysis provided evidence for the existence of four clusters of proteins (p1 to p4) and five clusters of patients (C1 to C5); protein clusters P1 and P3 are enriched in proteins encoded by immune-related genes; and C2 and C3 are associated with an increased rate of biochemical recurrence [378]. Interestingly, proteomic subtypes were distinct from genomic and AR signatures subgroups [378]. The only molecular subgroup characterized by a peculiar proteomic profiling is represented by the *ETS* gene fusions group: 145 mRNAs and 68 proteins were significantly associated eith *ETS* gene fusion status [378]. In this subgroup, at protein level, particular relevant was the peculiar enrichment in genes associated with carboxylic caid metabolism, suggesting a link between *ETS* fusions and lipid metabolism and in genes associated with intra- and extracellular vesicles, thus supporting a link between *ETS* fusions and cell migration [378].

Interestingly, in addition to coding RNAs, two long non-coding RNAs (lncRNA) were also found to be potential biomarkers for prostatic cancer. Thus, *PCA3* is a prostate cancer-specific lncRNA biomarker, overexpressed in prostate cancer tissues and in HGPIN [379]. *PCA3* is detectable in urine and its clinical utility was explored in several studies: the clinical sensitivity and specificity of urinary *PCA3* evaluation for prostate cancer detection are 58–82% and 72–79%, respectively [380,381]. Its detection, in combination with PSA or *TMPRSS2-ERG* RNA, showed some clinical utility [382]. *SCHLAP1* is a lncRNA discovered during a bioinformatic analysis carried out to identify lncRNAs selectively upregulated in prostate cancer [383]. *SCHLAP1* is not expressed in other cancers or in any normal tissue [384] and is highly overexpressed in patients with aggressive prostate cancer compared to localized prostate cancer [385]. *SCHLAP1* is a biomarker to predict metastasis development and to discriminate high-risk from low-risk prostate cancer [384] and its dysregulation was found to be associated with aggressive intraductal and cribiform pathology of prostate cancer [386].

## 7. Association of Genomic Abnormalities with Patient Clinical Outcomes

Many studies have consistently contributed to define the genomic landscape of metastatic castration-resistant prostate cancer, but vey few studies have explored the association of genomic abnormalities with patient clinical outcomes and with additional features prognostically relevant, such as tumor histology and transcriptional profile. In this context, a recent study provided a comprehensive integrative analysis of genomic abnormalities, transcriptomic profiles, histology and clinical outcomes of 429 patients with mCRPC [387]. For 128 mCRPC patients treated with Abiraterone or Enzalutamide, the association with DNA- and RNA-based genomic alterations with clinical outcomes was explored. Of all these alterations, only RB1 alterations were associated with reduced overall survival, whereas alterations in *RB1, AR*, and *TP53* were associated with shorter time on treatment with Abiraterone or Enzolutamide [387]. These observations have clearly indicated RB1 genomic alteration as a parameter predicxting a negative outcome and a reduced survival [387]. These observations strongly support future studies aiming to explore better the mechanisms of resistance to AR therapies induced by loss of *RB* and to identify potential active new drugs.

Another study performed on behalf of the West Coast Prostate Cancer Dream Team explored genomic drivers of poor prognosis and of Enzolutamide resistance in a group of 256 mCRPC patients [388]. Two or more *TP53* DNA alterations were observed in 47% of patients, and two or more DNA alterations in *PTEN* and *RB1* were observed in 36% and 12% of patients, respectively; the combination of these alterations showed that 23% of patients had two or more alterations in at least two of these genes, 17% had alterations in each *TP53* and *PTEN*, and 7% had no DNA alterations in any of these three genes; furthermore, *MYC* was amplified in 38% of cases and 86% of patients had AR gain of function [388]. Among these various DNA alterations, only DNA alterations in RB1 have a clear negative impact on overall survival: the median OS for patients with two alterations in *RB1* was 14.1 months, compared to a median OS of 42.0 months for patients without *RB1* alterations [388]. mCRPCs with *RB1* alterations had also a distinct transcriptomic profile with expanded E2F1 function and lower AR activity [388]. Furthermore, transcriptome profile analysis showed that the WNT/β-catenin pathway is the mostly enriched pathway among Enzalutamide-resistant patients [388]. Finally, multivariate analyses demonstrated that RB1 and CTNNB1 alterations are prognostic after accounting for clinicopathologic variables [388].

## 8. Sensitivity of Prostate Cancer to Immunotherapy

The understanding of the immune-related antitumor mechanisms greatly benefitted from the discovery of immunecheck inhibitors (such as PD1, PD-L1, and CTLA-4) and the development of monoclonal antibodies targeting these inhibitory receptors and thgeir ligands [389]. The immune pathways and the tumor cell features that determine the sensitivity to immune check inhibitors are yet not carefully characterized, but several markers, such as PD-L1 expression by tumor cells, mutational burden (a phenomenon particularly evident in tumors with microsatellite instability and with generation of many neoentigens related to frequent gene coding mutations), and presence of lymphoid infiltrations at the level of tumor microenvironment [390].

Five studies have explored PD-L1 expression in prostate cancer [391,392,393,394,395]. A recent meta-analysis of these studies showed that PD-L1 was expressed in 35% of prostate cancers; PD-L1 expression and PD-L1 methylation were both associated with poor biochemical recurrence-free survival and PD-L1 was expressed at high levels preferentially in high Gleason score tumors and AR-positive cases [392]. In contrast to these findings, PD-L1 had only a weak correlation with age, pathologic stage, lymph node metastasis and preoperative PSA levels [364]. Some studies provided evidence PD-L1 was more expressed in castration-resistant prostate cancer (32.1% positive tumors) than in primary prostate tumors androgen-sensitive (7.7% of positive tumors) and was particularly pronounced in neuroendocrine prostate cancers [391].

Some studies have explored the mechanisms regulating the expression of PD-L1 on prostate cancer in relation to specific molecular abnormalities [396]. PD-L1 expression can be regulated al both transcriptional and post-translational levels. A recent study showed that PD-L1 abundance on prostate cancer cells is regulated by cyclin D. CDK4-mediated phosphorylation of SPOP and the cyclin 3-SPOP E3 ligase via proteasome-mediated degradation [125]. SPOP mutation compromises of ubiquitin-mediated PD-L1 degradation leading to increased PD-L1 levels and reduced numbers of infiltrating lymphocytes in prostate cancers [125]. Another recent study showed that phosphorylated RB inhibits NF-kB activation and PD-L1 expression: in patients samples phosphorylated RB inversely correlates with PD-L1 levels [397]. Interestingly, the expression of a phosphorylation-mimetic peptide in prostate cancer cells suppresses radiotherapy-induced upregulation of PD-L1 and augments the therapeutic efficacy of radiation in vivo [397].

Few studies have explored PD-L2 expression in prostate cancers. In this context, particularly relevant was a recent study by Zhao and coworkers who performed a very large gene expression screening based on the analysis of more than 9000 prostate cancer samples and observed, through unsupervised hierarchical clustering of hallmark oathways, the existence of an immune-related tumor cluster [398]. However, deconvolution of the data for single immune cell types showed that mast cells, netural killer cells, and dendritic cells conferred improved distant metastasis-free survival (DMFS), while macrophages and T cells conferred negative DMFS [398]. At the level single immune-related genes, PD-L1 expression was not prognostic, while PD-L2 expression strongly correlated with immune-related pathways, expression of radiation response pathways, and response to postoperative radiation therapy [398]. These observations support a potential role for PD-L2 targeting alone or in combination with radiation as a potential therapeutic strategy. Interestingly, in this study, it was observed an inverse correlation between the immune pathways with the AR response pathways [398], a finding observed also in another recent gene expression study based on the anlysis of CRCP bone metastases [399]. In fact, according to this last study, bone metastase of CRPC can be subdivided into two subgroups: (a) more frequent (~80% of cases), displaying high AR activity and metabolic activities and low immune responses, and (b) more rare (~20% of cases), displaying low AR activity and low metabolic activities and high immune responses [399].

Prostate cancer is a tumor considered immunogenic and characterized by the absence of infiltrating T-lymphocytes in the tumor microenvironment [400]. However, in spite this negative view, the antitumor vaccine for solid tumors, Sipoleucel-T, was approved in 2010 for treatment of metastatic prostate cancer. This vaccine is an example of persobnalized medicine and is based on patient’s own antigen-presenting cells activated with PAP2024, a fusion protein of PSA and GM-CSF; this vaccine increased overall survival compared to placebo in metastatic prostatic cancer patients and was approved for treatment of these patients [401,402].

Several clinical studies have explored the clinical efficacy of various immune check inhibitors in prostate cancer patients. Usually, immunotherapies using single-agent have been not successful in metastatic prostate cancer, as well as in other genitourinary neoplasia, such as bladder cancer and renal cancer. Thus, Ipilimumab, a monoclonal antibody targeting CTLA-4, used as a single agent was basically negative for the treatment of mCRPC, with only few patients responding to this treatment, usually exhibiting low tumor burden [403]. Ipilimumab single agent was unable to improve the survival of mCRPC patients [404]. Interestingly, the treatment with Ipilimumab and Nivolumab (anti-PD-1) may induce a greater response rate among patients with metastatic prostate cancer, with responding patients only among those exhibiting homologous repair deficiency [405].

PD-1 targetingh therapy was associated with a low rate of responding patients in metastatic CRPCs. A pilot study based on only a small group of 10 patients with mCRPC progressing upon Enzalutamide therapy showed a response rate of 30% [406]. The results of a large phase III trial (KEYNOTE-199) were presented at the 2018 ASCO Annual Meeting [407]: in this study, a PD-1 blocking monoclonal antibody, Pembrolizumab, was administered to docetaxel-refractory mCRPC cohort 1 included patients with measurable tumors positive for PD-L1; patients with measurable tumors negative for PD-L1 and cohort 3 included patients with nonmeasurable tumors. Anti-PD-L1 monotherapy elicited a limited antitumor activity, with a response rate of 3–5% among patients with measurable disease [407]. No differences were observed between the PD-L1^+^ and PD-L1^-^ cohorts [408]. Eleven percent of all patients displayed a PSA decline >50%. Importantly, the tumors of responsding patients displayed DNA damage repair pathway alterations (*BRCA1*, *BRCA2*, and *ATM* mutations), supporting the view that defects in this pathway are biomarkers of anti-PD-1 response [407]. Thus, although some cancer patients respond to therapy with Pembrolizumab monotehrapy, the response rate is significantly lower than that observed in other genitourinary tumors, such as kidney (~40% of responding patients) or bladder cancer (~ 25% of responding patients).

There is a rationale to associate androgen deprivation therapy with anti-PD-1 treatment. In fact, studies of manipulation of androgen activity either with androgen inhibitors or by castration, striongly support the view that androgens have an immunosuppressive effect [378]. Furthermore, the various observations showing that androgens have immunosuppressive effects on CD4^+^ T_H_1 cells further support the view that removal of androgens by castration enhances immune function [408].

A phase II clinical study evaluated the clinical activity of an anti-PD-1 agent in patients progressing on the second-generation antiandrogen Enzatulamide. Initial results published from this study showed that ten of the patients enrolled in this study exhibited a reduction in circulating PSA levels [376]. A moree extend ed report on this study provided evidence that 18% of patients displayed reductions (>50%) of PSA levels and 25% of patients exhibited a detectable radiographic reduction of tumors [409]. The results of this study support to further explore the therapeutic potential of immunocheck inhibitors in combination with androgen deprivation. In line with this view, a randomized phase III trial is evaluating the role of a PD-1 inhibitor in prostate cancer treatment by treating patients progressing with an androgen synthesis inhibitor Abiraterone acetate with either Enzatulamide monotherapy or the combination of Enzatulamide with the anti-PD-1 antibody Atezolizumab.

A recent study explored the mechanisms responsible for resistance of prostate cancer to Ipilimumab. Thus, Gao and coworkers explored the effect of Ipilimumab administered together with androgen-derpivation before surgery in patients with localized prostate cancer and observed increased PD-1, PD-L1, and VISTA expression in prostate cancer [410]. Particularly, significant increases in PD-L1, as well as in tumor-infiltrating immune cells, including CD4^+^, CD8^+^, ICOS^+^, CD45RO^+^, granzyme-B^+^, and CD68^+^ cells are detected in post-treatment tumors which were not seen in the control group treated with androgen deprivation therapy alone [410]. Furthermore, an increase in VISTA, another immune chekpoint, was detected in the tumor microenvironment [410]. These observations support the view that Ipilimumab promote an immune response with a concomitant upregulation of PD-1 and VISTA, as adaptive resistance mechanisms [410].

Recent studies have analyzed the immunogenomic profile of prostate cancer. Thus, Navas Rodrigues and coworkers have shown that up to 8% of mCRPC have evidence for defective MMR, associated with loss of some MMR effectors, such as *MSH2*, *MSH6*, *MLH1*, or *PMS2*; dMMR was associated with poor overall survival compared to MMR-proficient prostate tumors (3.8 versus 7.0 years) [154]. Two mutational signatures of dMMR CRPCs were detected, charcaterized by increased expression of immune cells, immune checkpoint, and T cell-associated transcripts [154]. Wu et al. [290] found that the increased mutational burden in CPPC is associated with homologous recombination deficiency, caused by translocations, or with *CDK12*-mutated tumors, caused by focal tandem duplications [305]. Tumors bearing biallelic *CDK12* loss have T cell infiltration and increased numbers of expanded T cell clones [305].

Immune responses generated in the tumor microenvironment influence the response of prostate cancer to androgen deprivation therapy. Thus, various studies have shown that the immunosuppressive microenvironment of prostate tumors decreases the response to androgen deprivation therapy. Thus, Calcinotto and coworkers have shown that IL-23 secreted by neutrophil-myeloid-derived suppressor cells (MDSCs) contribute to generate resistance to androgen deprivation therapy [411]. MDCSs suppress the immune response in tumor microenvironment and promote senescence evasion and angiogenesis, thus promoting prostate cancerogenesis [412]. These findings imply the existence of a multitude of immune suppressor mechanisms present in prostate cancers that need to be countered via appropriate combination therapies.

Immunotherapy is a promising area for a subset of prostate cancers, but many challenges seem to limit its development [413]. First, the definition of a condition of dMMR remains difficult for many patients and its optimal assessment would require whole genome sequencing, a technique not readily available in current clinical practice. This conclusion is supported by studies showing that complex rearrangements of *MSH2* and *MSH6* genes are responsible for a significant proportion of somatic MMR mutations [58]. Second, most of clinical grade genomic assays do not detect copy number alterations in tumor suppressor genes, including MMR genes. These limitations are in part bypassed through immunohistochemistry analysis of the MMR proteins (MSH2, MSH6, MLH1, and PMS2). A third challenge derives from some ambiguity in the definition of MSI status and the method used to determine it. A classical method is based on the assessment by PCR of five microsatellite sequences, derived from studies on colorectal cancers. In a study carried out in 13 dMMR prostate cancers, the analysis by PCR showed that 27% of these tumors had no MSI marker shfted, 36% had two markers shifted, 36% three to four markers shifted, and none had all five markers shifted [155]. A more sensitive technique to assess MSI status in prostate cancer evaluates an extended panel of prostate cancer-relevant microsatellites analyzed by next-generation sequencing [414]. Thus, in a series of 29 MMR prostate cancers, the 60-marker next-generation sequencing method had a sensitivity of 93%, while the five-marker PCR had a sensitivity of only 72% [414].

As above mentioned, men carrying germline mutations in DNA repair genes, such as *BRCA1*, *BRCA2*, *CHECK2*, and *ATM*, are at increased risk of developing prostate cancer and of prostate cancer-related mortality [145]. *BRCA2* mediates homologous recombination and thus is not surprising that *BRCA2*-deficient cells have an increased potential for genetic intability-driven tumorigenesis. *BRCA2* germline mutations are more frequently mutated in mCRPC, than in patients with localized prostate cancers or in healthy population [149]. These observations suggest an association between germline BRCA2 mutations and aggressive prostate cancer disease. In fact, BRCA2 germline mutations are associated with a shorter metastasis-free survival and cause-specific survival and are an independent poor prognostic factor for localized prostate cancer [415,416]. Men on active surveillance with inherited mutations in *BRCA1/2* are more likely to develop aggressive prostate cancer and need to be reclassified for tumor grading in the context of the Gleason scoring [417].

Recent studies have in part clarifid the mechanisms underlying the biologic aggressiveness of tumors developing in *BRCA2* mutation carriers. Importantly, the secondary mutations and genetic alterations observed in de novo germline *BRCA2*-mutant prostate cancers are more similar to those observed in sporadic prostate cancers from advanced prostate cancers after extensive treatment than those observed in treatment-naïve sporadic prostate cancers [30,418].

Germline BRCA2-mutant prostate cancers harbor a twofold higher percentage of genomic aletrations, graeter somatic single-nucleotide variant and increased CAN burden compared to localized sporadic prostate tumors [30,418]. CNAs at the level of *MYC*, *MYCN*, *GSK3B*, *MTOR*, and *BRCA2* are more frequently detected in germline BRCA2-mutant prostate cancers than in sporadic localized cancers [30,418]. Interestingly, *BRCA2*-mutant prostate tumors more frequently display amplification of region located on chromosome 3q, encoding the WNT pathway modulator *MED12L* than sporadic prostate cancers [30,418]. These observations strongly support the view that germline *BRCA2* mutations influence the accumulation of prostate cancer-associated genetic alterations in normal prostatic epithelium, thus favoring prostate cancer development and progression [30,418].

Evidence reported to date on the response of germline *BRCA2*-mutant prostatic cancers to standard treatments is conflicting. Annala et al. have explored retrospectively the data obtained on 176 patients with mCRPC, including 22 germline DDR carrier (16 *BRCA2* carriers) and observed that the progression of gDDR on first line androgen deprivation therapy was significantly shorter that that of noncarriers [419]. Anatonarakis et al. have reported the outcomes of 172 patients with mCRPC, including 22 gRRD carriers (23% of these patients received chemotherapy before androgen-derpivation therapy): a trend toward a more prolonged PFS in gDDR carriers compared with noncarriers [420]. On a large retrospective series of 330 patients, including 60 carriers of gDDR (37 g*BRCA2*), Mateo and coworkers found no link between the gDDR status and the response to treatment with androgen deprivation treatment or taxanes [421].

Recently, the results of the study PEROREPAIR-B, the first prospective study designed to assess the prevalenve and impact of gDDR mutations in the outcomes of metastatic prostate cancer patients [422]. This study showed the prognostic role of *BRCA2* mutations for disease-related survival [422]. In this trial, 16.2% of mCRPC patients were identified as gDDR carriers, a proportion markedly higher than in healthy population [422]. The two most frequently germline-mutated genes in this population were *BRCA2* (3.3%) and *MUTYH* (3.1%) [422]. The disease-related survival was halved in *BRCA2* carriers compared to noncarriers [422]. Furthermore, the outcomes of the *BRCA2* carriers were greatly influenced by treatment sequence: these patients responde better to androgen deprivation than to taxanes, as first-line of treatment [422].

As discussed above, *BRCA2*-mutated prostate cancers are sensitive to PARP inhibitors [150,151]. Thus, PARP inhibitors are being explored as a treatment option for mCRPC in men harboring mutations in holomogous recombination DNA repair genes. Interestingly, in a recent study the outcomes for men with *BRCA1/2* mutations to those for men with *ATM* mutations beinge treateds with Olaparib, a PARP inhibitor, were compared; men with *ATM* mutations do not respond as well as men with *BRCA1/2* mutations to PARP inhibitors [423].

It is important to note that all DNA repair gene mutations are more frequent in advanced stages of prostate cancer, compared to initial stages. In this context, particularly interesting was a recent study by Mrashall and coworkers showing an association between the frequency of DNA repair gene mutations and clinical and pathological features of localized prostate cancers: thus, in Gleason score 1 and 2 prostate cancer, the frequency of DNA repair gene mutations was 5%, and in Gleason score 3 to 5 was 11%; in cT1–cT2 localized prostate cancers, the frequency of DNA repair gene mutations was 7.8% and in cT3–cT4 tumors was 15% [424]. These findings indicate that among patients with localized prostate cancer, maximum enrichment in PARP inhibitor-sensitivity occurs in patients with Gleason score ≥3 and clinical stage ≥cT3 [424].

## 9. Circular RNA and Prostate Cancer

Circular RNA (circRNA) is a novel class of noncoding RNA covalently bonded at 5′ and 3′ ends, forming a continuous loop which is more stable than linear RNAs [425]. crcRNAs are produced by precursor mRNA back-splicing of exons of thoudands of eukaryotic genes [425]. Originally, circRNAs were considered as function-less RNAs, devoided of any biological significance; however, recent studies have shown that circRNAs are functionally involved in the regulation of many biological processes and may be deregulated in some tumors [425]. circRNAs are usually expressed at low levels and frequently are expressed according to tissue-specific and cell-specific patterns [425]. Although the function of most of circRNAs remains unknown, growing evidences indicate that circRNAs may play an important role as gene expression regulators, miRNA spoinges, mRNA slicing regulators, and gene translation templates for proteins [394]. Growing evidence also suggests that circRNAs play significant roles in the development of some cancers and some circRNAs are abundantly expressed in some tumoras and their expression correlates with the severity of these tumors [425].

Very recent studies support a significant role of circRNA in prostate cancer. Thus, using a novel exomne capture RNA sequencing protocol, Vo and coworkers have profiled circRNA among more than 800 human cancer samples [426]. Using this capture sequencing, it was built a comprehensive catalog of circRNAs, MiOncCirc [426]. Using MiOncCirc, candidate circRNAs were identified [426]. circRNAs expressed in prostate cancers resulted to be tissue-specific, less tissue-specific, and ubiquitous; among circRNA transcritps that were deregulated in cancer, the majority were downregulated in cancer (with the downregulation of some circRNAsthat cannot be explained by the downregulation of the parental genes) and small subsets of circRNAs more expressaed in tumor sets that in normal counterpart (such as circular isoforms of *AKT3*, *SDK1*, *LUZP2*, *ABCC4*, and *AMACR*) [426].

In another recent study, Chen and coworkers have performed ultradeep non-Poly-A RNA sequencing on 144 localized prostate cancers, linked to their pong-term clinical outcomes. This analysis allowed identification of the circRNA profile of prostate tumors, with the finding of 76,311 distinct circRNAs [427]. The total circRNA burden correlates to disease progression in various patient cohorts [427]. Loss-of-function sceening found that 11.3% of abundant circRNAs act as potentially important regulators of cell proliferation; interestingly, for the large majority (≅90%) of these circRNAs their parental parental linear transcripts were not essential for cell proliferation regulation [427]. The function of some specific circRNA was investigated; thus, circCSNK 143 promotes prostate cancer cell growth by interacting with miR-181 [427]. This study supports also the adoption of ultradeep RNA sequencing without poly-A selection as a strategy to explore both linear and circular transcriptomes [396].

Other recent studies have reported the identification of specific circRNAs, such as circ-SMARCA5 [428] and circ-102004 [429], overexpressed in prostate cancer cells and promoting cell proliferation of these cells.

## 10. Hormonal Regulation of Prostate Cancer

The prostate is a hormonally regulated gland. During development androgen receptor-mediated events are thought to be mediated by AR expressed on stromal cells. Thus, it is believed that the binding of androgen hormones to ARs present on stromal cells causes the release of soluble factors (“andromedins”) that induce growth and differentiation of the prostate by binding to their cognate epithelial receptors. On the other hand, the tumoral transformation of a normal prostate cell to cancer is believed to occur accompanied by a switch from paracrine to cell-autonomous constitutive AR signaling. The role of AR in promoting prostate tumor growth at an early stage of tumor development is complex in that studies carried out in various experimental models have lead to the conclusion that stromal AR acts as a promoter of primary prostate tumor proliferation, while epithelial AR may act as a suppressor of tumor cell proliferation [430,431]. The role of autonomous androgen receptor signaling may be different following the different oncogenic mechanisms promoting prostate cancer development: thus, PIN developed by constitutive cell-autonomous AKT progressed independent of epithelial AR signaling; in contrast, PIN induced by paracrine FGF10 secretion was dependent on epithelial AR signaling [432]. The cell autonomous AR signaling occurring in prostate cancer seems to be mediated by autocrine release of androgens by AR-positive tumor cells and by acquisition of intracellular activation pathways. This mechanism operates also at the level of prostate CSCs [433].

There is some evidence to suggest that in prostate cancer development an important pathogenetic event could be related to abnormal steroid receptor signaling at the level of the stem cell compartment. Prostate hormonal carcinogenesis triggered by combined testosterone and estradiol treatment for several months has been established in murine models. Recent studies have shown a carcinogenic effect of steroids using human prostate regeneration models. These models were based on the recombination of prostate stem cells with rat urogenital sinus mesenchyme (UGM), followed by in vivo growth as renal grafts in immunocompromised mice. Using this approach it was reported the formation of human prostate-like tissues from embryonic stem cells mixed with rat UGM using a renal graft approach [434]. This method was recently improved reporting the generation of normal human prostate tissue starting from enriched preparations of human prostate stem cells combined with inductive rat UGM using a renal graft approach in nude mice. Using this approach, the grafted mice containing chimeric human-rat structures were exposed to elevated testosterone+estradiol doses: over a 1–4-month exposure period, the human prostate-like structures developed progressive neoplastic disease from atypical hyperplasia to PIN and high-grade prostate cancer with high local invasive properties [435]. In addition to androgen and estrogen, prolactin also through activation of Stat5 signaling may affect prostate cancer development though a direct effect on the basal stem-like compartment of prostate cells [436]. Sustained Stat5 activation was associated with the development of abnormal clusters of basal/stem cells in prostate epithelium of transgenic mice overexpressing prolactin at the level of the prostatic tissue and with the amplification of a luminal progenitor cell population, seemingly originated from amplified basal/stem cells [437]. Stat5 plays an important role in the transition of prostate cancer to its castrate-resistant state. Pharmacologic treatment of Stat5 is an efficient approach to delay castrate-resistant progression, due to the cooperation between Stat5 and Androgen Receptor to promote prostate cancer progression [438]. Jak2-Stat5 signaling inhibitors potently suppress the growth and induce apoptosis of primary prostate cancer cells and castrate-resistant prostate cancer cells, this inhibitory effect was observed in 75% of primary tumors grown ex vivo in organ explant cultures [439]. Therapeutic targeting of AR in prostate cancer using antiandrogens may be considerably enhanced by targeting of Stat5, through a stimulation of proteosomal degradation of AR liganded by antiandrogens [440]. Studies in experimental models support the concept that Stat5 signaling promotes metastatic progression of prostate cancer by inducing epithelial to mesenchymal transition and stem cell properties in prostate cancer cells [441]. The role of Stat5 in prostate cancer progression is supported by two clinical observations: (i) active nuclear Stat5 had a predictive value for early disease recurrence of localized prostate cancer treated with radical prostatectomy [442]; (ii) the Stat5 gene locus undergoes amplification during prostate cancer progression, conferring a growth advantage in prostate cancer cells [443]. A recent study showed that Stat5 induces Rad51, a key protein controlling DNA repair process, in prostate cancer cells; treatment cells with a Stat5 inhibitor sensitizes prostate cancer cells to radiation [444].

A key event in prostate cancer progression is represented by the development of castration resistance and the mechanisms responsible for this phenomenon are not completely understood. However, in spite these limitations studies carried out during these last years have shown that a high proportion of prostate cancers progress to androgen resistance through mechanisms involving the maintenance of androgen receptor-dependent signaling, such as (i) androgen receptor overexpression (amplification of androgen receptor gene copy number) and (ii) growth factor-regulated androgen receptor activation (usually due to gain of function mutations of the androgen receptor that confer greater sensitivity to androgens or ligand-independent activation); de novo autocrine/intracrine androgen production [162,445,446]. Androgen resistance is not related only to androgen receptor reactivation, but also to mechanisms related to tumor heterogeneity. Studies on advanced prostate cancers indicate that these tumors are heterogeneous, being composed by islets of cells that overexpress androgen receptors and other islets of cells that do not express androgen receptors [447,448]. Recent studies have stressed the relevance of AR heterogeneity in prostate cancer tissue. In fact, prostate cancers contain both AR^+^ and AR-low-expressing nonexpressing (AR^-/low^) and is augmented in advanced and relapsed prostate cancers [449]. Recently, Li and coworkers explored the existence of a possible link between the pattern of AR expression and response to therapy. These authors analyzed 200 prostate cancer tissues from CRPC patients and observed three different patterns of AR expression: nuclear (n-AR), mixed nuclear/cytoplasmic (n/c-AR) and low/no expression (AR^-/low^) [450]. Xenograft modeling experiments have shown that AR^+^ CRPCs are Enzalutamide-sensitive, while AR^-/low^ CRPCs are resistant [450].

The cellular mechanisms underlying the emergence of castration resistance in prostate cancer are not clearly defined. In this context, two hypotheses have been proposed: the first, based on an adaptative mechanism, implies genetic/epigenetic changes at the level of tumor cells previously androgen-dependent, and the second, based on clonal selection model, argues the emergence of castration resistance in consequence of the proliferation of a previously quiescent rare population of castration-resistant cells within an otherwise androgen-dependent tumor. In line with this last hypothesis, it was shown that early (stage I) human prostate adenocarcinomas harbor androgen-independent cancer cells with stem/progenitor-like properties [451]. These cells survive to antiandrogen therapies and may drive the subsequent divergence of disseminated CRPC [451].

## 11. Abnormalities of Metabolism in Prostate Cancer

Prostate cancer, as well as other tumors, display metabolic alterations essential for tumor development, survival of tumor cells in various environmental conditions, tumor progression, and resistance to therapy [452]. The metabolism operant in AR-driven prostate cancer is peculiar because is mainly fueled by lipogenesis and less by glycolysis and is more reliant on oxidative phosphorylation thgan most other solid tumors [453]. This topic was reviewed [452,454] and here are just outlined the most relevant studies, providing findings leading to the identification of metabolic vulnerabilities and of potential therapeutic targets.

Several other recent studies have shown that a dysregulated lipid metabolism plays an oncogenetic role in prostate cancer. The dysregulation of lipid metabolism observed in prostate cancer is complex and involves elevated de novo lipogenesis, including steroid hormone biosynthesis and beta oxidation of fatty acids [455]. An increase of de novo fatty acid (FA) synthesis is a hallmark of prostate cancer cells [452]. Prostate cancer progression is characterized by dysregulation of lipid metabolism, mediated by overexpression of fatty acid synthase (FASN), an enzyme playing a key role in de novo fatty acid synthesis [242]. This property distinguishes prostate cancer cells from normal prostate cells that rely mostly on diet-derived lipids [242]. As mentioned above, a recent study reported the development of a FASN inhibitor, IPI-9119, inducing a reduction of protein expression and transcriptional activity of FL-AR and AR-V7 [242]. On the other hand, AR induces FASN expression by a transcriptional mechanism involving both direct binding to the FASN gene promoter and activation of SREB1 (sterol regulatory element binding protein 1).

Another recent study defined a peculiar metabolic abnormality sustaining lipogenesis in prostate cancer cells. Pyruvate decarboxylase complex (PDC) is a gatekeeper multiprotein complex inducing the catalytic conversion of pyruvate to acetyl coenzyme A (acetylCoA), for entry into carboxylic acid cycle in mitochondria, includes PDHA1 as a major and key component [456]. Chen and coworkers provided evidence that the impairment of PDC and PDHA1 functions induces tumor suppression in prostate cancers [456]. The major mechanism through which PDHA1 knockdown induces prostate cancer suppression is related to abrogation of lipogenesis through a mechanism related to the nuclear localization of PDHA1A [456]. In fact, PDG controls the nuclear pool of AcetylCoA and, through this mechanism, regulates gene expression; in prostate cancer cells PDH1A regulates expression of lipid biosynthesis genes independently of mitochondrial PDC and regulates fatty acid biosynthesis in the presence of mitochondrial citrate [456]. As a consequence of this important regulatory role, pharmacologic inhibition of PDH1A inhibits the growth of prostate cancer cells [456]. Interestingly, in prostate cancer cells, PDC localizes both in the nucleus where it controls the expression of SREBF and in the mitochondria, where it supports the generation of cytosolic citrate for lipid synthesis [456]. Analysis of prostate cancers showed that both PDHA1 and the PDC activator pyruvate dehydrogenase phosphatase 1 (PDP1) are frequently activated and overexpressed [456].

Prostate cancer cells exhibit increased de novo synthesis of fatty acids and use fatty acids as the major source of carbons for lipid synthesis, while the contribution of glucose and glutamine is only minoritary [457]. This study showed also the heterogeneity of prostate cancer cells in their utilization of fatty acids [457].

Prostate cancer progression is characterized by increased rates of de novo fatty acid synthesis, independent of circulatory lipid levels [458]. The expression of the fatty acid synthase (FASN) enzyme and of the transcriptional regulator SREBP is significantly increased in prostate cancer and, particularly, in mCRPC [458]. Interestingly, several lines of evidence suggest a reciprocal link between fatty acid synthesis and AR signaling. FASN expression is transcriptionally induced by AR through activation of SREBP1 or through direct binding to FASN promoter region; SREBP1 inhibition downregulates AR levels [242]. A potent FASN inhibitor reduces the growth of CRPC cells, including those expressing AR-V7 [242].

As above discussed, PTEN deletion is a key event in prostate cancerogenesis and the study of PTEN deleted mice have consistently contributed to our understanding of prostate cancer development. Complete inactivation of *PTEN* alone in the mouse prostate leads to indolent tumors with minimal invasive properties after a long latency. This observation suggests that additional events cooperate with *PTEN* loss in driving advanced, mestastatic prostate cancer progression. The analysis of recent data of array-based comparative genomic hybridization based on the analysis of localized prostate cancers and mCRPCs showed that *PTEN* was lost in 14% of localized prostate cancers and 66% of mCRPC samples [459]. The analysis of other tumor suppressors codeleted with *PTEN* showed that *PML* was deleted in 31% of mCRPC, but was intact in localized prostate cancers; furthermore, *PML* and *PTEN* were codeleted in ~20% of cases, in association with metastatic disease; *PTEN* and *PML* loss negatively impacted on the survival of prostate cancer patients. These observations supported the exploration of the *PML* loss in mice: *PML* loss greatly potentiated the aggressiveness os *PTEN*-null tumors that became lethal and metastatic to lymph nodes [459]. Interestingly, *PML* loss causes in prostate tumors a MAPK activation that in turn elicited the activation of a sterol regulatory element-binding protein (SREBP) prometastatic lipogenic program; in vivo targeting of SREBP using a fatostatin blocked tumor growth and metastatic activity [459]. The tumorigenic role of SREBP was further supported by the observation that a high-fat diet induced lipid accumulation in prostate tumors and was sufficient to drive metastasis in a nonmetastatic PTEN-null mouse model of prostate cancer; furthermore, SREBP signature was greatly enriched in mCRPC [459].

Recent studies have supported the link between AR signaling and lipid biosynthesis in prostate cancer cells. Thus, the studies of various models of prostate cancer progression showed that lipid biosynthesis is maintained and reactivated during the progression to CRPC and increased lipid synthesis is associated with poor prognosis [460]. Blocking lipid/cholesterol biosynthesis in AR variants-expressing CRPC cells markedly reduces tumor growth through inhibition of mTOR pathway; silencing of a fattry acid elongase, ELOVL7, also leads to regression of CRPC xenograft tumors [460].

Interestingly, inhibition of cholesterol biosynthesis using simvastatin overcomnes Enzalutamide resistance in castration-resistant prostate cancer (CRPC) [461]. Mechanistically, simvastatin deacrease AR protein expression, which is further decreased in combination with Enzalutamide [461]. The deacrease in AR expression is mediated by simvastatin-induced inhibition of the mTOR pathway, whose activation is associated with increased 3-hydroxy-3-methyl-glutaryl-CoA reductase (HMGCR) and AR expression [461]. The role of aberrant cholesterol metabolism in prostate cancer cells is further supported by a study showing increased expression of the high-density lipoprotein-cholesterol scavenger receptor B1 (SR-B1) in primary and mCRPC cells [462]. While benign prostate cells are insensitive to SR-B1 antagonism, prostate cancer cells, particularly those expressing splice-variant AR, are inhibited by lowering SR-B1 expression/activity [462]. All these observations support the view that cholesterol could represent a potentially interesting target in CRPC cells.

A recent study addressed another abnormality involving pyruvate mitochondrial metabolism in pyruvate cancer cells. This abnormality involves mitochondrial pyruvate carrier subunit 2 (MPC2), a constitutive member of the MPC complex involved in the import of pyruvate into the mitochondrial matrix for incorporation into intermediary metabolism in the citric cycle. MPC is transcriptionally regulated by AR in prostate cancer cells and its inhibition restricts tumor cell proliferation and metabolic flow through lipogenesis and oxidative phosphorylation [463]. Metabolic disruption elicited by MPC inhibition activates the integrated stress response (ISR), preventing cell cycle progression and increasing glutamine incorporation into the TCA cycle [463]. Interestingly, a small molecule MPC inhibitor, MSDC0160, suitable for clinical studies, induced tumor suppression in models of AR-sensitive and castration-resistant prostate cancer [463].

Using a network-based integrative approach, it was shown that peculiar alterations in the hexosamine biosynthetic pathway (HBP) are critical for CRPC [464]. In fact, expression of the HBP glucosamine-phosphate N-acetyltransferase 1 (GNPNAT1) is markedly decreased in CRPC in comparison with localized prostate cancers [426]. Studies in experimental models of prostate cancer provided evidence that GNPNAT1 inactivation increased the aggressiveness and proliferation capacity of prostate cancer cells, through either activation of PI3K-AKT pathway in cells expressing FL-AR or by specific protein 1 (SP1)-regulated expression of carbohydrate response element-binding (ChREBP) in tumor cells expressing AR-V7 [464]. Importantly, addition of the HBP metabolite UDP-N-acetylglucosamine (UDP-GlcNAc) to CRPC cells inhibits cell proliferation [464].

As above discussed, Reina-Campos and coworkers have defined a mechanism whereby loss of PKCλ/ι, and subsequent upregulation of the serine, glycine, and one-carbon (SGOC) metabolic pathway in early-stage prostate cancer triggers the development of NEPC [99]. Particularly, this study showed that the cellular availability of the methionine-derived metabolite S-Adenosylmethionine (SAM) is one of the major mechanisms detrmining the development of NEPC. In fact, the levels of SAM are elevated in response to upregulation of the SGOC metabolic pathway and promote hypermethylation of DNA, ultimately causing the transcriptional gene expression program underlying NEPC formation [99]. Interestingly and importantly, targeting the SGOC metabolism in a suitable animal model reduced NEPC formation, providing direct support to the concept that SGOC metabolism is a potential metabolic vulnerability, therapeutically exploitable, of NEPC [99]. The findings of this study again support the concept of the possible links between alterations in cellular metabolism with prostate cancer cell biology.

The metabolism of other amino acids is also deregulated in prostate cancer. Glutamine is an essential amino acid used as a source of carbon and nitrogen for macromolecule synthesis and as a fueling system to dupport ATP production in the tricarboxylic acid (TCA) cycle through glutaminolysis, involving first conversion of glutamine to glutamate and then its metabolization to α-ketoglutarate; α-ketoglutarate may be used also to stimulate synthesis of lipids, necleotides, and amino acids. In an initial study, Wang et al. showed that the glutamine transporter ASCT2, also known as SLCA5, is highly expressed in prostate cancer samples; chemical inhibition of this receptor inhibited cell cycle progression of prostate cancer cells through E2F transcription factors and reducedbasal oxygen consumption and fatty acid synthesis, thus providing evidence that downstream metabolic function is reliant on ASCT2-mediated glutamine uptake [465]. Another study showed that two glutamine transporters—SCL1A4 and SCL1A5—whose expression is stimulated by AR, are overexpressed in prostate cancer cells and stimulate glutamine uptake by these tumor cells and are required for maximal AR-driven proliferation [466]. The upmodulation of these two glutamine receptors implies the mTOR pathway activation [466]. The screening of aggressive prostate cancer cell lines showed increased glutamine utilization and a peculiar sensitivity to the glutaminase inhibitor CB-839 [467]. A RAS inhibitor molecule, RASAL3, was epigenetically silenced in human prostate cancer-associated fibroblastic cells inducing Ras activation and micropinocytosis-mediated glutamine synthesis [468]. ADT further enhanced RASAL3 silencing, and glutamine secretion by prostatic fibroblasts [468]. Interestingly, antagonizing the uptake of glutamine restored sensitivity to ADT in a castration-resistant xenograft mode [468]. In line with these findings, prostate cancer patients on ADT with therapeutic resistance displayed elevated blood glutamine levels compared to those responsive to ADT [468].

IDH1 and ADH2 catalyze the reversible oxidative decarboxylation of isocitrate to yeld α-ketoglutarate as a part of the TCA cycle in glucose metabolism. IDH1 is localized in the cytoplasm, while IDH2 is present in mitochondria. IDH1/IDH2 genes are mutated in 1–3% of all prostate cancer cases. Interestingly, prostate cancers display the highest IDH1 expression levels across the human cancer spectrum and IDH1 expression increases during porstate cancer progression. AR exerts a positive control on IDH1 expression, stimulating the transcription of *IDH1* gene [469]. Knockdown of IDH1 blocked the effect of AR on IDH activity. IDH activity promotes prostate cancer progression. According to these observations it concluded that AR reprograms prostate cancer cell metabolism, favoring extramitochondrial IDH1-mediated IDH activity [469].

Prostate cancer hypoxia contributes to the development of tehrapy resistance mechanisms. Interestingly, it was recently shown that chronic ADT in the conditions of hypoxia induces daptive AR-independence, and therefore determines resistance to AR-targeted therapy [470]. The mechanism of this effect is mediated by Glucose-6-phosphate isomerase (GPI), transcriptionally repressed by AR in hypoxic conditions, by stimulated by AR inhibition [470]. The induction of GPI stimulates glucose metabolism and energy homeostasis under hypoxic conditions by shifting the glucose flux from the AR-dependent pentose phosphate pathway to hypoxia-induced glycolysis, thus reducing the growth inhibitory effect elicited by AR inhibitors [470]. Importantly, inhibiting GPI allows bypassing therapy resistance in hypoxia in vitro and potentiates enzalkutamide efficacy in vivo [470]. Interestingly, targeting hypoxia reduction may represent a strategy to restore T cell infiltration and to sensitize prostate cancer cell to immunotherapy with immune check inhibitors [471].

Thus, reprogramming of cell metabolism is a hallmark of prostate cancer, as well as of other malignancies. AR emerges as a main driver of the reprogramming of specific metabolic pathways that contribute to tumor growth and disease progression. These recent observations support the importance of metabolic studies because the identification of metabolic vulnerabilities of prostate cancer cells may open new avenues for novel personalized diagnostic and therapeutic approaches [452].

## 12. Prostate Stem Cells

The human prostate is a compound tubular–alveolar gland and is composed of distinct glandular subunits that each independently drains proximally into the prostatic urethra. Each glandular subunit is composed by a complex branching of stratified epithelia, composed by three cell types: basal, luminal, and neuroendocrine cells. The putative prostate stem cells are localized at the level of the basal compartment, in strict contact with the basal lamina on one side and with the stromal cells on the other side. Basal and luminal cells of the human prostate express different markers and are differentially regulated by hormones: in fact, luminal cells express cytokeratin (CK) 8 and 18, while basal cells express CK5 and CK14; basal cells do not express the AR and prostate specific antigen (PSA), while express at high levels p63 and the antiapoptotic protein Bcl-2; in contrast, luminal cells express high levels of AR, respond to androgens, express PSA. In addition to basal and luminal cells, immunohistochemical studies have suggested the existence of an intermediate phenotype between basal and luminal secretory epithelial cells, displaying the coexpression of CK5 and CK18 and being localized both at the level of basal and luminal cell compartments. The intermediate cells located at the level of the basal compartment have been termed transit-amplifying cells. Neuroendocrine epithelial cells express markers, such as chromogenin A and synaptolysin.

Recently, Henry and coworkers have performed a single-cell RNA sequencing and flow-cytometry study of normal human prostate cells obtained from different anatomical regions. Basal epithelial CD271^+^ cells are characterized by KRT5, KRT14, TP63, NOTCH3, LTBP2, and DKK1 expression, while luminal CD26^+^ cells are characterized by KLK3, ACPP, MSM3, GP2, NEFH, and NPY expression [427]. The single-cell characeterizatiuon allowed the identification of two now prostatic cell types chacterized by high expression of SCGBA1 and KRT13, respectively: (i) prostate KRT13^+^ cells are similar in mnorphology and trasncritpional profile to hiullock basal cells of the lung; these cells, concentrated in prostatic urethra and proximal prostatic ducts, are rare in adult prostate but abundant in fetal prostate and are characterizerd by expression of members of the androgen metabolism pathway (AKR1C1 and AKR1C2). (ii) Prostate SCGBA1^+^ cells are similar in their morphology and transcriptomic profile to Calra, or club, cells (cells of the epithelial lining of the respiratory tract, concentrated in the proximal trachea) and enriched in ummunomodulatory programs [472].

The existence of prostate stem cells was proposed when Isaacs and colleagues found that a fraction of prostate cells remain after castration-induced involution and were capable of regenerating the full gland with all different cell types [473]. In the normal prostate, luminal cells are androgen-dependent and undergo apoptosis following androgen deprivation, while basal and neuroendocrine cells are castration-resistant. On the other hand, the observations made on prostate cancer patients undergoing androgen ablation therapy allowed to postulate that certain prostate cancer cells share properties with normal adult prostate stem cells and have the capacity to survive androgen therapy and subsequently to regenerate the tumor with a more aggressive phenotype [474].

In adults, prostate epithelium is quiescent and, therefore, there is no apparent need for the activity of stem cell to maintain the normal hemostasis of this tissue. However, prostate stem cells play a key role in the context of androgen-mediated prostate regeneration. In fact, following androgen deprivation the prostate tissue regresses; however, when physiologically normal androgen levels are restored, the prostate gland regenerates back to its original size [475]. This observation is compatible with the idea that normal adult prostate stem cells exist under steady-state conditions in a completely quiescent state but are triggered to cycle and to repair when prostate tissue is damaged.

Various methods have been developed for the identification and propagation of prostate stem cells and to determine their location within the normal prostate. These methods involve the study of the expression of various luminal and basal-specific cell markers, cell lineage tracing experiments, prostate sphere (protaspheres) cultures, and organoid cultures.

To try to identify normal prostate stem cells various approaches have been attempted, providing different types of evidences. Basically, flow cytometry and tissue reconstitution studies have led to the identification of basal stem cells, while a genetic lineage-marking approach has identified luminal stem cells [476].

The characterization of prostate stem/progenitor cells was based on in vitro and ex vivo functional assays. A first approach is based on cell culture, involving prostasphere formation in suspension culture, to evaluate the self-renewal and differentiation potential of cell populations isolated by flow cytometry. A limitation of the prostasphere assay is that it allows the growth of only basal cells and not of luminal cells [432]. To bypass this limitation, 3D prostate organoid cell culture systems have been developed, allowing the growth and differentiation of both basal and luminal cell types [433]. A widely used assay consists in the dissociated prostate cell regeneration assay, which involves the mechanical and enzymatic dissociation of prostatic tissue and the combination of the dissociated prostatic cells with embryonic urogenital sinus mesenchymal cells (UGSM) and their grafting under the kidney capsule of immunodeficient male host mice. If in the prostatic tissue are present stem/progenitor cells, glandular regenerating structures resembling adult prostate tissue are observed at the level of the grafts. This assay measures at some extent the prostate stem cell activity and can be used also to evaluate the self-renewal of these cells. In this assay, the presence of UGSM cells is strictly required to obtain prostatic cell differentiation. An alternative method consists in an in vitro prostate sphere assay, a simplified surrogate assay. In this assay, a small fraction of prostate cells displays the property of forming cell spheroids when cultured in 3D Matrigel cell cultures. These spheres are of clonal origin, but usually are not able to generate in vivo prostate glandular structures when induced by UGMS cells in regenerative assays. Two main limitations of the regenerative assay are related to the disruption of the natural prostate microenvironment and the co-culture with embryonically-derived UGMS cells that could reprogram prostate stem/progenitor cells and the impossibility to identify the prostate cell types responsible for the regenerative activity. To bypass these limitations, a cell tracing strategy based on the specific genetic labeling of luminal or basal prostate cells with fluorescence proteins was developed, and then another on the tracking of the cell fates of the labeled cells in vivo in their physiological microenvironment. This cell tracking strategy has given an important contribution to the study of the development of prostate and, particularly, prostate stem cells, both normal and tumoral. This strategy uses genetically engineered mice in which the expression of Cre recombines within defined cell populations can be used to genetically mark cells with expression of a suitable reporter, such as a fluorescent protein, thus permitting the tracing of the progeny of the marked cells. Finally, a last assay consists in the so-called label-retention assay, allowing the identification of long-lived quiescent cells within the prostate: this assay was based on a pulse-chase strategy involving an initial labeling of newly synthesized DNA with bromo-deoxyuridine (BrdU) or of nucleosomes with histones fused with green fluorescent protein (GFP) [476].

In human prostate there is evidence about the existence of a stem cell located within the basal compartment. A number of markers, including α_2_β_1_ integrin, CD44, and CD133, identify basal cells exhibiting prostate stem cell properties [477,478,479,480,481,482]. These cells have a high colony-forming capacity in vitro and are able in vivo to reconstitute prostate glands in immunodeficient mice, paralleled by differentiation into luminal cells positive for Androgen Receptor and PSA expression. Particularly, Goldstein et al. using the prostate regeneration assay showed that the prostate stem cell activity is enriched at the level of CD49f^+^Trop2^+^ human prostate basal cells [483]. Interestingly, Garraway and coworkers [482] showed that the basal cells able to form prostate spheres are not the equivalent of the cells able to regenerate tubular structures in the prostate regeneration assay: EpCAM^+^CD44^-^CD49f^high^ cells are the tubular regenerating prostate stem cells, while EpCAM^+^CD44^+^CD49f^high^ cells are the sphere forming cells [482]. More particularly, a set of studies was carried out in mouse and other ones in human prostate. In a series of elegant experiments, Leong et al. have shown that single basal CD133^+^/CD44^+^/CD117^+^ cells are able to develop into a whole mouse prostate when xenografted under appropriate conditions [484]. Using combinatorial markers, Goldstein and coworkers have identified another stem cell population in the basal layer characterized by positivity for Sca-1 and CD49f^high^ [485]. Sca-1^+^/CD49f^high^ stem cells were characterized also by the expression of the polycomb group protein Bmi-1 that seems to be required to maintain the stemness of these cells inhibiting their differentiation and favoring their self-renewal [485]. The interrelationship occurring between the CD133^+^/CD44^+^/CD117^+^ and the Sca-1 and CD49f^high^ basal stem cells is unclear.

Basal progenitor cells have been functionally characterized for their capacity to generate prostate in tissue generation assays. Studies of tissutal regeneration after castration [486,487,488] or after luminal epithelial damage [489] support a role for bipotent basal progenitors in prostate regeneration supporting a role for basal progenitors to generate luminal cells.

Using a combination of lineage tracing experiments, 3D glandular reconstitutions, proliferation kinetics, and functional assays of cell differentiation, Moad and coworkers defined the lineage potential, location, and functional behavior of human adult prostate stem cells and their cell progeny [490]. This approach provided evidence about the existence of multipotent basal stem cells, located at the level of discrete niches in juxta-urethral ducts, generating a progeny of basal bipotent progenitors that migrate in coherent steams along the proximal–distal axis [490]. Basal progenitors are dispersed along all the prostate glandular tissue and when necessary divide to replace apoptotic luminal cells; luminal stem/progenitor cells are rare and are confined to proximal ducts, giving only a marginal contribution to epithelial homeostasis [490].

Hu and coworkers reported a method to isolate and to explore the functional properties of adult human prostate stem cells at single cell resolution level [491]. This methodology was based on the identification of long-term label-retaining cells in primary pentaspheres: these cells represent a scarce prostate cell type that is relatively quiescent and may represent the epithelial stem cell that initiates spheroid growth [491]. Label-retaining cells exhibit numerous stem cell properties, including symmetric and asymmetric cell divisions, low cytokeratin 14 and E-cadherin and elevated WNT10B expression, augmented autophagy activity, enriched transcriptomic profiles of stem cells, and low anabolic mitochondrial metabolism [491]. This approach identified prostate stem cells from prostate cancer specimens [491].

Genetic lineage marking studies involved genetic lineage marking of progenitor cells, followed by analysis of progeny differentiation in vivo allowed identifying a luminal prostate stem cell. Using this approach, Wang and coworkers [492] identified a rare luminal epithelial population with stem cell properties during prostate regeneration: these cells were called CARNs (Castration-Resistant NKX3.1-Expressing Cells) [492]. According to these observations it was proposed that CARNs coexist together with basal stem cells in the normal prostate and serve as luminal stem cells. Alternatively, CARNs are a “facultative” type of stem cells, corresponding to progenitors, that is, activated following androgen deprivation.

Clonality analysis of basal stem cells has shown that these cells are functionally heterogeneous. In this context, Ousset and coworkers have labeled basal cells and have followed their differentiation fate at the individual cell level: some basal cells generate only basal cells; other basal cells generate luminal cells; finally, other ones generate both basal and luminal cells [493]. Furthermore, tracing with K5 and K14 promoters generated different results at the level of cell types generated by K14^+^ and K5^+^ basal cells: particularly, these experiments showed that K14^+^ cells are more primitive, being multipotent and capable of generating basal, luminal and neuroendocrine cells [493]. These observations suggest that in normal prostate, like in normal bone marrow, there exists a progeny of prostate progenitors, multipotent or unipotent [493].

On the basis of the currently available evidences the current view indicates the existence of a stem cell population present within the basal layer of the normal prostate that possesses tubule-forming capacity and can originate through its differentiation multipotent progenitors; in turn, these progenitors are able to give rise to all three cell types present in the prostate: mature basal cells, neuroendocrine cells, and luminal progenitors that generate mature luminal cells [494]. An alternative view is that there may be multiple stem cell populations within the normal prostate epithelium, and thus, in addition to prostate basal stem cells, there is a population of luminal stem cells. In fact, two research groups have recently reported that a very small percentage (0.3–1%) of luminal cells is able to survive in vitro and to display stem cell properties [495,496]. Thus, in a first study, Chua and coworkers have generated in culture prostate organoids from either purified CARNs or from normal prostate epithelial cells (with a lower efficiency) which exhibit a tissutal architecture containing both basal and luminal cells, undergo long-term expansion in culture and exhibit functional androgen receptor signaling [495]. Lineage tracing experiments provided evidence about the preferential origin of organoids from luminal cells, while basal cells were much less efficient to generate prostate organoids [495]. In the second study Karthaus and coworkers have developed an R-spondin 1-based culture method allowing long-term growth of murine and human prostate epithelium [496]. Using this culture system, they showed that both basal and luminal prostate cells are able to induce the efficient generation of prostate organoids: the frequency of progenitors able to generate prostate organoids was higher among basal than luminal cells [124]. This study allowed to reach the important conclusion that human prostatic epithelium harbors both luminal and basal stem cells [496]. The difficulty of demonstrating the stamina capacity of luminal prostatic progenitors has not been demonstrated in prostate stem cell assays, probably due to the strong susceptibility of these cells to anoikis, i.e., to the apoptosis triggered by loss of interaction with stromal cells [491]. In this context, recently Kwon and coworkers have shown that increased NOTCH signaling suppresses anoikis of luminal epithelial cells by augmenting NF-kB activity and stimulates luminal cell proliferation by potentiating PI3K-AKT signaling and rescues the capacity of the putative prostate luminal progenitors for unipotent differentiation in vivo [487,491].

The unipotent in vivo capacities of differentiation of prostate luminal progenitors is also supported by a recent study of cell lineage tracing transgenic mice using basal and luminal cell-specific markers. These assays have shown that while basal cells display both symmetric and asymmetric divisions leading to different cell fates, luminal cells only exhibit symmetrical divisions [497].

Recent studies have investigated the properties of subsets of luminal cells isolated by flow cytometry. In the mouse prostate, Kwon and coworkers provided evidence that Sc a1^+^/CD49f^low^ cells are enriched for luminal progenitors that display bipotential properties both in organoid cell cultures and in tissue reconstitution assays [498].

Liu and coworkers reported the identification and characterization of a luminal prostate progenitor [499]. These studies were promoted by the observation that expression of the luminal cell marker CD38 is heterogeneous in the human prostate; low CD38 expression was used as a marker to isolate a subset of p63-negative, keratin 14 (K14)-negative, keratin 18 (K18)-positive luminal cells from human prostate, with peculiar properties at functional and molecular levels compared to CD38^high^ luminal cells [499]. Importantly, these CD38^low^ cells are expanded around the regions with inflammation, exhibit an inflammatory signature and are enriched in progenitor cell activity [499]. Additional observations supported a biological role for this CD38^low^ luminal cell population, showing that these cells express the therapeutic target PSCA (prostate stem cell antigen) and are able to regenerate prostate gland in transplantation experiments of organoid cultures [455]. Interestingly, it was shown the existence of a connection between CD38 levels and NAD levels in that a CD38 increase inhibits cancer metabolism through a diminution of glycolytic and mitochondrial metabolism, activation of AMP-activated protein kinase (AMPK) and inhibition of fatty acid and lipid synthesis [500,501].

Recently, Zhang and coworkers reported the isolation and characterization of a slow-cycling population of luminal cells in murine prostate [502]. These authors used a bigenic mouse model to isolate and characterize the stem cell properties and gene expression profiles of quiescent label-retaining cells from mouse prostate expressing a tunable H2B-GFP driven by the promoter of a luminal gene, probasin [502]. These slow-cycling luminal progenitors are enriched in proximal prostate, are quiescent and display bipotency in in vitro and in vivo assays [502]. These cells have low AR expression and are resistant to castration [502].

Single-cell gene profiling analysis showed a consistent heterogeneity of the luminal compartment and allowed the identification of LY6D as a marker of luminal progenitors with a bi-lineage gene expression pattern, with multipotent differentiation capacities and enriched organoid-forming capacity; these cells are resistant to androgen deprivation and could represent progenitor cells involved in the generation of CRPC [503]. Interestingly, in prostate cancer, LY6D expression correlates with early disease progression [503].

## 13. Stem Cells in Benign Prostatic Hyperplasia (BPH)

BPH is characterized by a slow and progressive enlargement of the prostatic gland due to hyperproliferation of both epithelial and stromal cells in the transition zone of the prostate gland. BPH is a very frequent condition and it was estimated that ~25% of men develop symptoms of BPH (bladder outlet obstruction) in their lifetime. Autopsy studies have shown a histological prevalence of the disease of 8%, 50%, and 80% in the fourth, sixth, and ninth decades of life, respectively. Genome-wide association studies suggest the existence of a genetic component in the development of BPH, showing that some single nucleotide polymorphisms reveal a correlation with BPH and serum levels of PSA [504,505,506]. Various pathogenic mechanisms have been proposed to explain the origin of BPH, including alterations at the level of the prostate stem cell compartment. In the BPH the ratio between stromal and epithelial cells changes from about 2:1 observed in the normal prostate gland to 5:1 observed in BPH. Therefore, the hyperproliferative activity of stromal cells is an essential component in the development of BPH. There is some evidence that stem cells present in the stromal compartment may be responsible for the development of BPH. Thus, Lin and coworkers have developed primary cultures of BPH and have observed that the cells of these cultures express stem cell markers, such as CD30, CD44, CD49, CD54, nonspecific enolase (NSE) (highly expressed), CD34, VEGF-RI, and Stem Cell Factor (Moderately expressed) [507]. The origin of these cells is unknown, but the positivity for CD49, CD54, NSE, and SCXF suggests a proximity to the mesenchymal stem cell lineage [507]. Chronic inflammation has been implicated in the initiation and progression of BPH and prostatic cancer. These inflammatory stimuli drive the recruitment of cells of the innate and adaptive immune system and mesenchymal stem cells [508]. Interestingly, in contrast to normal prostate tissue that contains only lineage-restricted mesenchymal progenitor cells, neoplastic prostate tissues contain tri-lineage differentiation potential (adipogenesis, osteogenesis, and chondrogenesis) mesenchymal stem cells [508]. The MSC is particularly high in patients with aggressive prostate cancers [508]. Aberrant transforming growth factor-beta activation represents a triggering stimulation promoting the recruitment of MSCs at the level of the hyperplastic prostate tissue [509].

Other studies have shown that CD49f^+^ cells isolated from BPH possess both monolayer and spheroid-forming capacity [510]. These spheroids undergo intensive proliferation, form branching ductal structures and expressed both basal and luminal markers [510]. The CD49f^+^ cell population comprised two components, a basal epithelial component CK5^+^ and an endothelial component CD31^+^ [510].

Some recent studies suggest a possible role of the c-kit in the physiopathology of BPH. C-kit was expressed in the human prostate at the level of the stromal component (interstitial cells) [511]. C-kit expression in BPH was increased compared to the normal counterpart [511]. The tyrosine kinase inhibitor imatinib mesylate inhibited the proliferation of prostate stromal cells by downregulating JAK2 and Stat1 [511].

A recent study suggests an important role for TROP2 as a driver of BPH. TROP2 is a cell surface protein expressed on immature stem/progenitor-like prostate cells [512]. TROP2 controls self-renewal, proliferation, and transformation of prostate epithelial cells. This function is achieved through a process of regulated intramembrane proteolysis that determines the cleavage of TROP2 with release of the extracellular and intracellular domains of TROP2. The intracellular domain stimulates stem/progenitor cells self-renewal through a signaling mechanism involving beta-catenin [512], determining the development of benign prostate hyperplasia in vivo [512]. Cleavage-defective TROP2 mutants failed to induce BPH [465]. TROP2 overexpression is a mechanism stimulating the stem-like properties of transformed prostate epithelial cells [512].

According to these observations, it was recently proposed a hypothesis to explain the development of BPH in ageing men in which various episodic occurrences of prostatic inflammation trigger the recruitment of mesenchymal stem cells to the transition periurethral zone under the effect of chemokines exerting an attractive effect on the migration of these cells from the bone marrow [513]. Recruited MSCs in the prostate tissue displace the periurethral smooth muscle sphincter and stimulate the benign prostatic neoplastic expansion of epithelial stem cells, allowing the formation of BPH nodules [513]. This hypothesis is supported by observations made at the level of BPH tissues.

## 14. Prostate Cancer Stem Cells

Studies carried out in the last years have supported an important role of prostate cancer stem cells (PCSCs) in prostate cancer initiation, progression, and maintenance [514]. Many studies have been carried out with the specific aim of characterizing and identifying cancer-initiating cells in prostate cancer. Three different sources of cells have been used to try to identify prostate cancer stem cells: either primary tumor cells or tumor cell lines or xenografts passaged in vitro. In an initial study Collins et al. reported the isolation of CD44^+^/CD133^+^/α2β1^high^ cells from primary tumors and these cells were able to form colonies of tumor cells, display a typical luminal phenotype, but were unable to induce tumor formation when injected into immunodeficient mice [515]. Given the genotypic and phenotypic heterogeneity in prostate cancer, the use of single markers for the selection, identification and characterization of tumor initiating cells was elusive, and the reliability of cell surface markers, such as CD133, as a way to isolate these cells was questioned by other studies and remains controversial to date [516,517]. This conclusion is a supported by a recent study using in vivo lineage tracing and reaching evidence that CD133 does not enrich for stemness in the normal mouse prostate [518]. Kalantari and coworkers have investigated CD133 and CD44 expression in a large set of prostate tumor tissues corresponding to various degrees of tumor progression. A higher level of CD44 expression was observed in 42% of prostate cancers, 57% of HGPIN and 42% of BPH, while in the case of CD133 expression. PCa, HGPIN and BPH samples displayed high immunoreactivity in 46%, 43%, and 42% of cases, respectively [519]. An inverse significant correlation between CD44 with Gleason score of PCa, while no significant correlation between CD133 expression and clinicopathological parameters were observed [519]. In spite these consistent doubts about the role of CD133^+^ cells in prostate cancer as tumor initiating cells, a recent study reported that CD133^+^ cells isolated from prostate cancer cell lines showed expression of markers of stem cell differentiation (CD44, OCT4, and SOX9), epithelial to mesenchymal differentiation markers, γ-radiation, and chemotherapy drug resistance [520]. These cells displayed elevated RUNX2 expression; its downmodulation restore sensitivity of these cells to chemotherapy [520]. Finally, Reyes et al. reported the presence of circulating CD133^+^ tumoral cells in the blood of patients with mCRPC [521].

Another initial study was based on the characterization of prostate cancer stem cells in immortalized human epithelial prostate cells from primary tumors: these cells do not express AR or p63, express embryonic stem cell and early progenitor cell markers and when inoculated under the renal capsule of male SCID mice, recapitulate the original tumor with multipotential differentiation [522].

A complementary strategy for the identification and isolation of normal stem cells and their malignant counterpart consisted in the measurement of aldehyde dehydrogenase (ALDH) activity: detection of ALDH activity can be carried out on viable cells and the labeled cells can be isolated according to their level of positivity. ALDH^high^ cells, isolated from primary prostate cancers, exhibit a marked proliferative activity both in vitro and in vivo and formed metastasis when inoculated into nude mice [523]. According to these findings it was suggested that ALDH-based viable cell sorting can be used to identify and characterize tumor-initiating cells in prostate cancer. Recently, the presence of ALDH^bright^ cells was investigated in freshly isolated tumor cells from 39 prostate cancer specimens, showing that among EpCAM^+^ cells, a proportion of ALDH^bright^ cells variable from 0.5 to 19% was observed [524]; higher percentages of ALDH^bright^ in high as compared to medium Gleason score were evident [522]. The large majority of these ALDH^bright^ cells displayed expression of the membrane receptor TROP2, a stem/progenitor cell marker [524]. Other studies have shown that PSA^-/low^ prostate cancer cells have stem cell properties; particularly, an ALDH^high^CD44^+^α2β1^+^ cell population is able to initiate xenograft prostate cancers in castrated mice [525]. Importantly, triple-marker^+^ cells isolated from primary prostate cancers display multiple stem cell properties at functional and phenotypic levels [525]. Expression of ALDH1A gene in prostate cancer cells is regulated by the WNT signaling pathway and co-occurs with expression of β-catenin [524]; inhibition of WNT pathway determines a decrease in ALDH^+^ tumor progenitor population and increases radio sensitization of tumor cells [526].

Other studies have suggested that c-met, the receptor of hepatocyte growth factor, could represent a marker of immature prostate cancer cells. Thus, studies of immunohistochemical staining of prostate cancers have shown that bone metastatic tumors display an increased c-met positivity [527]. In invasive prostate cancers c-met expression was found to be associated with stem-like markers CD49b and CD49f [528]. Activation of c-met in prostate cancer cells stimulates the expression of a stem cell program, characterized by upregulation of CD49b, CD49f, CD44, and SOX9, and downregulation of CD24 [528]. In line with these observations, Nishida and coworkers have isolated from prostate cancers an ALDH1^high^ cell population enriched in cancer stem cells and have shown that these cells have the dual property of expressing c-met and of producing HGF: this creates an autocrine system of HGF maintaining prostate cancer tumor-initiating cells [529]. Other recent studies have suggested an important role for EGFR in the propagation of populations of cancer stem cell like cells isolated from prostate cancer cell lines [530]. This effect of EGF promoting cancer stem cell self-renewal requires MEK/ERK activation [530].

Other studies were focused to isolate cancer stem cells from tumor cell lines. CD133^high^ or CD44^+^CD24^-^ cells or CD133^+^CD44^+^ cells have been isolated from DU145 and PC3 cell lines and have been shown to be able of inducing colony tumor formation in vitro and tumor formation in vivo after subcutaneous injection in mice [531,532].

A third approach for the identification of prostate cancer stem cells consisted in the use of prostate tumor xenografts: using these cells in a series of studies Patra Wala et al. isolated CD44^+^/α2β1^high^ cells from xenografts, and showed that these cells form tumor spheres in vitro and induce tumor formation following subcutaneous or orthoptic injection in immunodeficient mice [533,534,535]. It is important to note that the efficiency of development of tumor xenografts from primary prostate cancer specimens is low. An alternative to the xenografts consists in the development of tumor organoids from primary tumor specimens. In this context, a recent report, using a methodology used for the development of organoids from normal prostate epithelial cells, reported the development of tumor organoids from primary advanced prostate cancers, with an efficiency of ~15–20% [536]. Importantly, these tumor organoids reflect the molecular abnormalities and display the heterogeneity of the primary tumors from which they were derived [536]. Given the low frequency of obtention of tumor organoids from primary tumors, this methodology cannot be proposed to study individual tumor patients; however, these organoids represent a unique and precious tool to explore at various levels subtypes of prostate cancers bearing the major molecular abnormalities typically observed in these tumors [536].

The expression of stem cell-related markers possesses a clinical/prognostic significance. In fact, Fujimura showed that the level of expression of the stem-related markers KLF4, c-Myc, Oct 3–4, and Sox2, together with the expression of androgen and estrogen signaling components [537]. On the basis of these parameters prostate cancer patients have been stratified into three subgroups: favorable, intermediate and poor-risk groups [537].

A number of studies were focused to evaluate the cell of origin of prostate cancer, particularly in view of the luminal or basal cell origin. Since the prostate cancer cells have a luminal phenotype one can hypothesize either a cell origin from a luminal cell or from basal cells that can differentiate into luminal cell following tumorigenic transformation. A number of studies from experimental models suggest a basal cell origin: thus, experiments of transformation with lentivirus containing ERG and androgen receptor genes of sorted luminal and basal cells provided evidence that only the latter ones displayed tumor formation [538]. Furthermore, transformed basal cells can generate prostate adenocarcinomas with luminal phenotypes [538]. In contrast, other prostate cancer models, such as PSA-Cre; Pten^flox/flox^ and NKX3.1-MYC models, support a luminal cell origin of prostate cancer [277,539,540]. In favor of the luminal origin is also the recent observation showing that androgen receptor mediates formation of the *TMPRSS2-ERG* fusion in human prostate cancer cells, thus suggesting that cancer initiating events may occur at the level of androgen receptor-positive luminal cells [105,541].

Since it remained unclear whether basal or luminal cells, or both, represent the cell types originating prostate tumors in the context of *PTEN* deletion, Wang and coworkers used a novel cell-tracing strategy to assess tumor cell origin in a variety of mouse prostate cancer models, including *PTEN*^+/-^, Hi-Myc, NKX 3.1, and TRAMP mice [542]. The results of this analysis showed that luminal cells were constantly the cell of origin for these tumors. However, basal cells obtained from these tumors are in turn able to generate tumors in secondary grafts [542]. According to these findings it was proposed that luminal cells are the preferential cells of origin of prostate cancers in the majority of cases and basal cells are able to generate prostate tumors only after differentiation into luminal cells [542]. The idea that basal cells need luminal differentiation to express their tumorigenic potential is also supported by another recent study exploring the mechanisms through which inflammation favors prostate cancer formation. Thus, Kwon and coworkers have shown that acute inflammation causes tissue damage and a consequent change in prostate tissue microenvironment exerting a stimulatory effect on luminal differentiation of basal cells: this prodifferentiative effect accelerates and favors disease initiation in mouse models for prostate cancer with a basal cell origin [543].

On the other hand, other studies have shown that basal cells are efficient targets for tumorigenesis induced by *PTEN* loss in mouse models [544]; furthermore, prostate basal cells manipulated to overexpress ERG, Androgen Receptor, and/or PI3K generate PIN lesions and carcinomas [544].

The tumorigenic potential of human prostate basal and luminal cells was addressed in recent studies. When these cells were transduced with oncogenically relevant oncogenic lesions, together with a dye tracer, and transplanted into immunodeficient mice, only the basal cells were shown to be able to initiate the development of prostate cancers, similar to those that arise in humans [545]. It is important to note that genetic ablation of PTEN or expression of a dominant-negative mutant of PTEN in mouse prostate epithelial cells induces prostatic intraepithelial neoplasia.

Wang and coworkers have initiated tumors into murine basal and luminal epithelial cells and have shown that tumors developed from the transformation of these cells exhibit distinct molecular signatures [546]. Oncogenic transformation of basal cells give rise to tumors with luminal phenotypes that are less aggressive than tumors originated from the transformation of luminal epithelial cells [546]. According to these observations it was suggested that prostate cancers may derive from the transformation of different cell types and that basal cells have consistent inherent differentiation plasticity [546]. Other studies have explored the effect of removing PTEN at the level of basal or luminal cells. Thus, Choi and coworkers provided evidence that both lineages are capable of generating malignant lesions, but basal cells were more resistant to transformation [547]. On the other hand, using a keratin 5 promoter-driven Cre, Lu and coworkers have shown that tumoral processes initiated by PTEN loss in basal cells are more proliferative and aggressive than those initiated by luminal cells [548]. The selective ablation of PTEN in prostatic luminal cells at adulthood induces the slow development of PIN lesions, characterized by an initial proliferation of prostate epithelial cells, followed by a progressive growth arrest with features of cell senescence [549]. The results of a recent study provided evidence that a Lin^-^/Sca-1^+^/CD49f^mid^ could be the cell of origin of *PTEN^-/-^* prostate cancers [550]. Interestingly, these castration-resistant cells exhibit a unique gene expression profile characterized by the sharing of luminal and basal markers [550]. These cells are characterized by the expression of CK4 [550].

Recently, Cai and coworkers explored the role of various oncogenes inducing the formation of prostate cancer in mediating in vivo expansion of the tumoral progenitor/stem cell pool [551]. Particularly, they have explored the functional synergy in prostate cancers in mice resulting from the activation of the AR, KRAS, and AKT. Interestingly, any of these two events was sufficient to promote the formation of prostate cancer, but only the functional synergy of the oncogenic function of AR and KRAS signaling could promote prostate cancer progenitors in vivo and elevate EZH2 expression [551]. EZH2 is one of the key components of the Polycomb Repressive Complex 2 which controls gene expression through the methylation of H3 via its methyltransferase activity; through these effects on gene expression, EZH2 plays a fundamental role as a key regulator of stem cell pluripotency and early embryogenesis [551]. The expression of EZH2 increased in prostate cancer, and its elevation is associated with prostate cancer progression and poor prognosis.

This conclusion was confirmed through the analysis of a large set of prostate cancer biopsies showing that EZH2 expression correlates with Gleason score and lymph node metastases [552]. Furthermore, the genome-wide analysis of various datasets of primary and metastatic prostate cancers showed that the concurrent EZH2 and TOP2A expression identifies a number of patients with more aggressive disease and overlap with genes involved in mitotic regulation. In line with this observation, these tumors are sensitive to drugs targeting EZH2 and TOP2A [553]. Interestingly, EZH2 and BRCA1 are coregulated in primary prostate cancer cells and cooperate in the regulation of CSC phenotype and properties [554]. Inhibition of EZH2 and BRCA1 in experimental models of prostate cancer induces an increase of cancer stem cell properties [554]. Interestingly, other studies have shown that EZH2 oncogenic activity in CRPC cells is polycomb-independent [555]. EZH2 acts as a transcriptional coactivator for transcription factors such as AR [555]. Furthermore, EZH2 activates AR also through an additional mechanism involving direct binding at the level of the AR promoter: EZH2 overexpression increases EZH2 mRNA and protein expression, while depletion of EZH2 decreases AR expression [556]. EZH2 activates AR independently of its histone methyltransferase activity [556]. Interestingly, other studies have shown that the EZH2 inhibitor GSK126 in combination with Enzalutamide synergistically inhibits cell proliferation and induces apoptosis of enzalutamide-resistant prostate cancer cells [557]. Another mechanism through which EZH2 expression promotes castration resistance is represented by the repression of molecules such as CCN3 that normally act as repressors of AR signaling [558]. Based on the increasing evidence for the role of EZH2 in prostate cancer progression, EZH2 inhibitors are being evaluated in CRPC patients, as well as in other cancers [559]. These EZH2 inhibitors include CPI-1205, PF-06821497, Tazemetostat and DS-3201b. A randomized phase Ib/II study (ProSTAR, NCT03480646) is evaluating CPI-1205 with Enzalutamide or Abiraterone/Prednisone in patients with mCRPC after prior AR-inhibitor therapy [560].

Recently, the isolation and characterization of tumor spheres from primary human prostate cancer was reported [561]. The efficiency of tumor sphere formation was low. The characterization of these tumor spheres showed that they were (i) negative for the expression of key markers of prostate cancer, including androgen receptor, NKX3.1, PSA, and cytokeratin-18; (ii) positive for cancer stem cell and cell proliferation-associated markers, such as MET, inhibitor of differentiuation-1, Musashi-1, and Ki67; and (iii) positive for CD44, EpCAM, TRA-1-60 (a cell surface epitope of human embryonic, embryonal germline and teratocarcinoma stem cells), CD151, and CD166; triple positive CD151/CD166 and TRA-1-60 cells recapitulate the original tumor heterogeneity in serial xenotransplantation assays [561]. Interestingly, both CD151 and CD166 molecules are involved in bone metastasis of prostate cancer cells [562,563].

Recently, Tang and coworkers have utilized a PSA promoter-driven lentivirus reporter system to evaluate the relative contribution of various subpopulations of prostate cancer cells to generate hormone refractory disease. Using this approach, they demonstrate that in human prostate cancer, the PSA^-/low^ cell population contains tumor-initiating cells, resistant to castration [564]. PSA (Prostate Specific Antigen), an androgen-regulated, tissue-specific product of differentiated prostate secreted in the blood, is not expressed in prostate progenitor cells and its expression increases when these cells differentiate. Particularly, PSA^-/low^ cells undergo asymmetrical divisions, generating PSA^-^ cells that remained in their undifferentiated condition, and PSA^+^, differentiated cells [564]. Importantly, PSA^-/low^ cells are more resistant to chemotherapy than PSA^+^ cells. Two additional important properties of PSA^-/low^ cells are that they are quiescent and express several “stem cell” regulatory factors, such as Nanog, NKX3.1, ASCL1, and TBX15 [564]. The coexpression on PSA^-/low^ cells of other stem cell markers, such as CD44, ALDH1, and alpha2beta1 integrin, enabled purifying of a subpopulation of cells very enriched in tumor-initiating cells [564].

It is of interest to note that CD44 positivity was repeatedly reported as one of the properties of prostate cancer cells possessing properties of tumor-initiating cells. A recent study explored the mechanisms that could be responsible for the elevated expression of CD44 observed in these cells. Thus, miR-34a was found to be commonly underexpressed in populations enriched in prostate cancer stem cells [565]. Since CD44 was shown to be a direct target of miR-34a it becomes evident that the low expression of this miRNA in prostate cancer stem cells [565]. Interestingly, the overexpression of miR-34a in these cells induced a decrease of CD44 expression and concomitantly reduced the tumorigenic potential of these cells [565]. A recent study provided evidence that p53 plays an important role in the control of normal and tumoral prostate stem cells. In fact, it was shown that mice with prostate epithelium-specific inactivation of p53 and miR-34, a direct target of p53, exhibited a significant expansion of the prostate stem cell pool and a tendency to develop early invasive adenocarcinomas and high-grade prostatic intraepithelial neoplasia (the single inactivation of either p53 or miR-34 determines a stimulation of prostate stem cell self-renewal and an increased expression of c-met and responsiveness to c-met-mediated stimulation of cell proliferation) [566]. P53 exerts a control on prostate cancer stem cells also through modulation of CD51 expression [567]. CD51 is expressed at high levels in prostate cancer patients and correlates with poor prognosis; CD51^+^ prostate cancer cells have self-renewal capacity and its expression is required for prostate cancer stem cell-related properties and increases metastatic and drug-resistant properties [567]. Interestingly, WT-p53 downregulates CD51 expression and through this mechanism limits cancer stem properties [567].

The cancer stem cell origin of prostate cancer is supported also by the studies on embryonal stem cell markers. SOX2, NANOG, and OCT4 are key regulatory genes that maintain the pluripotency and self-renewal properties in embryonic stem cells. NANOG and OCT4 are expressed in primary prostate cancers, where their expression positively correlated with increased prostate tumor Gleason score [568]. Similar observations have been made also for SOX2 whose expression in prostate cancer correlated with histologic and Gleason score [569]. Xenotransplantation assays provided evidence that SOX2 promoted tumorigenesis and, particularly, increased the antiapoptotic properties of human prostate cancer cells [569]. NANOG expression in prostate cancer was particularly evident at the level of a progenitor/stem CD44^+^ tumor compartment [570]. In xenotransplantation assays, NANOG promoted tumorigenesis and sustained, in prostate cancer cells, the expression of stem-associated markers, such as CD133, ALDH1 and CXCR4. More recent studies have better defined the role of NANOG in the biology of prostate cancer. Jeter and coworkers have shown that NANOG is required for the growth of CRPC xenografts [571]. Studies at molecular level have shown that NANOG interacts with a distinctive region of AR/Forkhead box A1 (FOXA1) genomic loci and, inconsequence of this binding, alters AR/FOXA1 signaling, determining repression of AR-regulated prodifferentiation genes associated with stem cells, cell cycle, cell motility and castration resistance [571]. As a consequence of all these events, NANOG activates its own distinct transcriptional programme and engage other transcription regulators such as MYC, leading to acquisition of a castration-resistant stem cell-like state [571]. It is not surprising that prostate cancers highly expressing pluripotential transcription factors, such as NANOG, OCT4, and SOX2, more rapidly develop castration resistance and are associated with a poor outcome [572].

Two recent studies show the molecular mechanisms through which SPOP controls NANOG protein. Particularly, SPOP inhibits the self-renewal and stem cell properties of prostate cancer cells via the ubiquitinating-dependent degradation of NANOG regulated by AMPK activation [138] or by direct interaction with NANOG [132]. Particularly, AMPK activation promotes NANOG degradation by blocking the binding of NANOG to BRAF, which phosphorylates NANOG at Ser68; NANOG Ser68 is required for the direct interaction between SPOP and NANOG [138]. These observations suggest a novel therapeutic strategy based on AMPK activation to target prostate cancer stem cells. SPOP is able also to directly interact with NANOG, mediating its polyubiquitination and subsequent degradation [132]. Furthermore, the Pin1 oncoprotein acts as an upstream regulator of NANOG stability in a SPOP-dependent manner, protecting NANOG from NANOG polyubiquitination and degradation [132]. These findings suggest the use of Pin inhibitors to block NANOG-mediated prostate cancer stem cell traits, promoting the SPOP-mediated degradation of NANOG.

As mentioned in the section on genetic abnormalities, the fusion of the AR-regulated TMPRSS2 gene with ERG is a very frequent event during early stages of prostate cancer tumorigenesis. This fusion event causes androgen-stimulated overexpression of EGR which induces enhanced expression of SOX9, a transcription factor playing a key role in prostate ductal morphogenesis and in the maintenance of the stem cell pool. SOX9 enforced expression into murine prostate-induced tumor formation [573]. This interesting study shows how the TMPRSS2-ERG fusion protein redirects AR to a set of genes, including SOX9, that are not normally androgen-stimulated, and this results in an oncogenic effect [573]. SOX9 drives prostate cancer development through various mechanisms, particularly related to WNT/β-catenin signaling activation [304]. β-catenin forms a complex with AR and through this mechanism augments AR signaling in prostate cancer [574].

As mentioned above, populations enriched in prostate cancer-initiating cells are more chemoresistant than the bulk tumor cell population. A recent study provided evidence that these cells are resistant to radiotherapy. Thus, Cho et al. examined the relative number of cancer stem cells in irradiated prostate cancer cell lines following long-term recovery: although irradiation does not immediately favor increased survival of cancer stem cells, irradiated prostate cancer cell lines exhibit an increase in cancer stem cell properties with long-term recovery [575]. In line with these observations, also recurrent prostate cancer following radiotherapy treatment showed an increased expression of cancer stem cell markers [575]. These findings were confirmed and extended by a recent study showing that exposure of metastasis-derived prostate cancer cell line to clinically relevant doses of ionizing radiation resulted in the generation of two types of surviving cells: one corresponding to adherent senescent-like cells and the other corresponding to nonadherent anoikis-resistant stem-like cells, characterized by the expression of several stem cell markers and by active NOTCH signaling [576]. The survival of these nonadherent stem-like cells requires active ERK1/2 signaling and can be chemically inhibited by MEK inhibitors [576]. The adhesin–core subunit SMCA1 (Structural Maintenance of Chromosome 1A) promotes growth and migration of prostate cancer cells and is associated with prostate cancer radioresistance; this factor acts as a regulator of epithelial–mesenchymal transition and cancer stem-like properties [577].

In addition to be radioresistant, prostate cancer stem cells display also a consistent chemoresistance. Using a theoretical approach similar to that adopted for the characterization of the cancer stem cells involved in the insurgence of castration-resistant disease, Doming-Domenech and coworkers isolated a cancer subpopulation involved in the development of docetaxel resistance occurring in hormone-refractory prostate cancer. To this end, these authors have isolated a tumor cell subpopulation of cells that in vitro and in vivo survive to docetaxel: these cells lack standard differentiation antigens and HLA class I antigens, but overexpress the NOTCH and HEDGEHOG signaling pathways [578]. Furthermore, these cells, identified in spontaneously occurring prostate cancers, exhibit a potent tumor-initiating capacity [578]. Interestingly, targeting NOTCH and HEDGEHOG signaling induced apoptosis of these cells, decreasing the expression of the survival molecules AKT and Bcl-2 [578]. A key role of HH signaling in mediating prostate cancer stem cell chemoresistance is supported also by another recent study carried out on paclitaxel-resistant prostate cancer cell lines [579]. These cells possess distinct side-population (SP) exhibiting stem cell properties, which increased following paclitaxel monotherapy; however, cyclopamine, a potent HH inhibitor, restored the sensitivity of SP cells to paclitaxel [579]. Both NOTCH and HEDGEHOG signaling, highly active in prostate cancer stem cells, contribute to tumor chemoresistance through induction of the expression of ATP-binding cassette (ABC) transporters on the surface of these cells [580,581]. In line with these findings, a recent study explored a subpopulation of Docetaxel-resistant cells in the tumor populations of CRPC patients and showed that these cells are enriched in cells with stem cell markers (ABCB1, encoding the multidrug resistance membrane glycoprotein); NOTCH signaling was found to be activated in these cells and its inhibition increased the sensitivity to Docetaxel [582]. On the other hand, another study provided evidence that BMI1, a factor playing a key role in the control of self-renewal of prostate cancer stem cells, plays an important role in chemoresistance of prostate cancer cells through the transcriptional control of BCL2, mediated by the transcription factor TCF4 [583]. Activation of NOTCH pathway in prostate cancer cells is linked with both an epithelial–mesenchymal transition and enhanced cancer stem phenotype [584]. A recent study identified a prostate cancer subpopulation with low expression of Numb, a factor playing a key role as cell fate determinant, characterized by a pronounced resistance e to androgen deprivation [585]. Numb expression is downregulated in prostate cancer and is negatively associated with prostate cancer progression [585]. Studies in various experimental models support the view that Numb exerts a suppressive effect on prostate cancer development and castration resistance by inducing an inhibitory effect on NOTCH and Hedgehog signaling [585]. NOTCH and Hedgehog inhibitors induce a depletion of Numb^-/low^ prostate cancer cells [585]. The analysis of Enzalutamide resistant cells showed that the NOTCH pathway is linked to Enzalutamide resistance. Abrogation of NOTCH activity both in vitro and in vivo restores sensitivity of prostate cancer cells to Enzolutamide [586].

There is evidence that inflammatory mechanisms participate to tumor development and this effect seems to be mediated through a stimulation of prostate progenitor cell proliferation. Using mouse models, evidence was provided that androgen deprivation determines the migration of B lymphocytes at the level of regressing tumors: these lymphocytes release lymphotoxin and TNF-alpha that determine IKKα activation at the level of surviving prostate cancer cells and, through this mechanism, accelerating the emergence of castration resistant prostate cancer [587]. A more recent study by the same authors showed that IKKα is required for expansion of prostate progenitors. In fact, in prostate cancer cells, IKKα phosphorylates the transcription factor E2F1 on a site that promotes its nuclear translocation and the subsequent association with CBP and recruitment at the level of some important gene promoters, including Bmi1, a transcription factor playing a key role as regulator of prostate cancer self-renewal [588].

As mentioned above, BMI1 is a key regulator of prostate cancer stem cells. BMI1 is overexpressed in prostate cancers associated with negative pathologic and; the presence of BMI1 expression is predictive of disease recurrence. Importantly, BMI1 was found to be particularly expressed at the level of a subpopulation of prostate cancer cells, displaying properties of tumor-initiating cells. Using in vitro sphere forming assays and mouse tissue regeneration assays it was provided evidence that BMI1 plays an essential role in regulating normal mouse prostate stem cell self-renewal and cancer initiation [486]. Particularly, inhibition of BMI1 protects prostate cells from FGF10-induced hyperplasia and reduces the growth of aggressive PTEN-deletion-induced prostate cancer [486]. Various mechanisms are responsible for the increased BMI1 expression at the level of prostate cancer stem cells. Among these various mechanisms a peculiar role is played by a miR, miR-128, whose expression is downmodulated in prostate adenocarcinoma, particularly in metastatic prostatic cancer. miR-128 targets BMI1 mRNA, and therefore it is not surprising that BMI1 protein levels are increased in prostate cancer cells, where miR-128 levels are decreased [589]. Exogenous miR-128 expression into prostate cancer cells suppresses tumor regeneration in various tumor xenograft models though inhibition of cancer stem cell-associated properties, such as holoclone and tumor sphere formation, as well as clonogenic activity and survival [589]. Inhibition of BMI1 expression, as well as of other stem cell-associated markers, such as Nanog and TGFBR1, is responsible at large extent of these effects of miR-128. Several recent studies have contributed to better define the role of BMI1 in the biology of prostate cancer stem cells. Bansal and coworkers showed that BMI1 is overexpressed in populations of CD49^high^CD29^high^CD44^high^ enriched in prostate cancer stem-like cells [531]. BMI1 pharmacologic inhibition in patient-derived tumor cells decreased tumor colony formation in vitro and tumor initiation in vivo [590]. BMI1 exerts the majority of its biologic effects through the polycomb repressive complex 1 (PRC1), enhancing the activities of RING1B to ubiquitinate histone H2A at lysine 119 and to repress gene transcription. However, BMI1 exerts also some important oncogenic effects in prostate cancer cells through PRC1-independent mechanisms, involving direct binding to AR and consequent inhibition of MDM2-mediated AR degradation and sustained AR signaling in prostate cancer cells [591].

Importantly, BMI1 marks a population of castration-resistant luminal epithelial cells enriched in the mouse proximal prostate [592]. Lineage tracing experiments provided evidence that these castration-resistant BMI1-expressing cells, were called CARBs and were shown to be capable of self-renewal and regeneration [592]. CARBs may initiate prostate cancer development upon PTEN deletion, generating luminal prostate cancers [592]. In a second study, the same authors have investigated the potential of CARB cells that survival castration to initiate recurrence in vivo. To this end, it was analyzed the response to androgen deprivation in tumor initiated in CARB cells by PTEN loss: the treatment with androgen deprivation determines the regression of luminal tumors, remaining in a dormant condition for about three months and displaying a luminal-to-basal lineage switching [593]. In the residual cells a subpopulation of BMI1^+^ subpopulation was observed to be expressing the stem cell reprogramming factor SOX2; lineage retracing experiments have shown that the recurrence occurs from these cells present in the regressed tumors [593]. Interestingly, inhibition of BMI1 using the mall molecule inhibitor PTC-209 resulted in decreased expression of SOX2, increased cellular senescence and a delay of tumor recurrence after castration [593].

Additional mechanisms play a relevant role in prostate cancer development acting at the level of the cancer stem cell compartment. One of these mechanisms is triggered by the cytokine IL-6, whose levels are considerably increased in prostate cancer. A recent study by Kroon and coworkers provided evidence that most primitive populations of prostate cancer cells (i.e., CD44^+^CD133^+^) express 6-fold to 7-fold higher IL-6 levels than the more mature tumor cells (i.e., CD44^+^CD133^-^) and IL-6 receptor is expressed on progenitor cancer cells and is activated by IL-6, with consequent STAT3 activation. Importantly, blockade of activated STAT3, either by a neutralizing anti-IL-6 mAb or by a STAT3 inhibitor, greatly inhibited the clonogenicity of stem-like cells in patients with high-grade disease [594]. In a murine model of xenograft prostate cancer, an anti-IL-6 monoclonal antibody was able to induce the inhibition of outgrowth of a patient-derived castrate-resistant tumor [594]. In line with these findings, Qu and coworkers have shown the essential role of IL-6/Stat3 on the generation of tumor-initiating cells from the EPT2 premalignant cells [595]; these cells when grown in serum-free medium, without exogenous growth factors, develop tumor spheres exhibiting properties of tumor-initiating cells [595]. When ETP2 cells are grown under serum-free conditions produce reactive oxygen species (ROS), triggering the autocrine production of IL-6, which in turn activates Stat3; experiments using neutralizing anti-IL6 mAb have shown the essential role of IL-6 in the process of generation of tumor-initiating cells [595]. Interestingly, a recent study showed a link between AR and the IL-6/Stat3 system. In fact, it was shown that AR loss in prostate cancer cells favors stem-like phenotype and triggers IL-6 production, which in turn activates Stat3 [595]. These findings suggest that IL-6 secretion and downstream Stat3-mediated signaling is a critical pathway after AR blockade [596]. These observations suggest that STAT3 could represent a potentially important target for prostate cancer. Recent studies with Stat3 small molecule inhibitors supported a high antitumor activity exerted both at the level of differentiated prostate cancer cells and of prostate cancer-initiating cells [597]. These experiments showed that a significant part of the antitumor activity of Stat3 inhibitors is mediated through an inhibitory effect of these compounds on tumoral microvascular niche [597]. Prostate cancer stem cells displayed a high constitutive Stat3 activity and were highly sensitive to the inhibitory effect of Stat3 inhibitors, even at low doses [597]. Importantly, Stat3 inhibitors were highly efficient in eradicating prostate cancer in xenotransplantation models from primary prostate cancers [597]. The important role of the tumor microenvironment in promoting prostate cancer progression and castration resistance, through the interaction between tumor associate macrophages and cancer stem cells is supported by recent study [598]. The combined targeting of cancer stem cells and their interaction with macrophages by inhibiting the IL6 receptor improved the efficacy of androgen deprivation therapy in an orthoptic prostate cancer model [598]. Interestingly, a recent study provided evidence about a key role of the upregulation of miR-424 in impaired ubiquitination and degradation of Stat3, through the targeting of the E3 ubiquitin ligase COP1, leading to accumulation and activation of Stat3 in prostate cancer cells [599]. miR-424 may represent a potential therapeutic target to block its oncogenic effects mediated via noncanonical activation of Stat3 [599]. Recent studies explored the levels of total and activated Sta3 (phosphorylated Stat3, pStat3) during prostate cancer progression. The nuclear expression levels of total Stat3 and pStat3 in the epithelial cells of benign glands were significantly higher than in cancerous glands [600]. Low pStat3 expression in the epithelial cells of cancer prostatic glands in hormone-naïve prostate cancers was associated with faster tumor progression [600]. pStat3 and IL6R levels were explored in metastatic tissues, showing the bone metastatic lesions of prostate cancer express high levels of pStat3 and IL6R [601]. The anti-IL6 antibody siltuximab (CNTO 328) was shown to inhibit the proliferation in vitro and in vivo of prostate tumors and delayed progression to castration resistance in experimental models; however, at clinical level, this antibody was not successful in monotherapy in phase I/II studies in prostate cancer patients [602].

The signals that regulate the expansion of prostate cancer/progenitor cells are not very well known. In this context, recent studies suggest, in addition to the above discussed role of MNI-1 and TROP2, a relevant role of the β4 integrin, a laminin-5 ligand, in the mechanism of prostate self-renewal. These studies were stimulated by the observation that β4 integrin is highly expressed in a significant fraction of advanced prostate cancers and in castration-resistant tumors [603]. The deletion of β4 integrin resulted in a pronounced inhibition of prostate tumor growth and progression in a mouse model initiated by *RB* and *p53* loss [603]; these inhibitory effects on tumor growth were related to a defective self-renewal of cancer stem cells and to a reduced proliferation of transit-amplifying cells in vivo [603]. Biochemical studies have indicated that the mutant β4 integrin fails to promote cancer stem cell self-renewal due to a defective transactivation of ERBB2 and c-Met in prostate tumor progenitors: combined pharmacological inhibition of ERBB2 and c-Met greatly reduced the number of prostate cancer stem cells [603].

Glycogen synthase kinase-3beta (GSK-3beta) is a serine/threonine kinase acting on various biochemical substrates and is involved in the regulation of numerous key cellular pathways involved in regulation of proliferation, apoptosis, autophagy, and glycogen metabolism. GSKβ expression is increased in CRPC and stimulates the transactivation of AR [604]. GSKβ was identified as a crucial kinase for the maintenance of prostate cancer stem/progenitor cells and pharmacological inhibition of this kinase dramatically reduced the size of these cells [605]. As a consequence of these effects, GSK inhibitors greatly reduced the clonogenicity of tumors and decreased their migratory potential. Interestingly, GSK inhibition elicited also a consistent inhibition of various integrins, including the cancer stem cell-associated alpha2beta1 integrin [605].

Although prostate cancers are among the tumors with a low score of hypoxic gene signature, there is a strong association between hypoxia with prostate cancer progression: in fact, tumor hypoxia was associated with biochemical relapse [368,606], and predicts a negative response to radiotherapy [607] in localized prostate cancer. A more recent study confirmed an association between higher hypoxia scores with advanced prostate cancer; in contrast, PSA pretreatment levels do not correlate with hypoxia, as well with the most recurrent somatic mutational alterations [238]. The study of localized prostate cancers provides strong evidence that hypoxia correlates with miR-133a-3p dysregulation, elevated rates of chromotripsis, allelic loss of PTEN, and shorter telomeres [238]. Furthermore, hypoxia-mediated signals are essential for the survival of prostate CSCs. Thus, the Hypoxia-inducible factor 1 alpha (HIF-1 alpha) was found to be highly expressed in purified putative murine (Sca-1^+^/CD49f^+^) and human (CD44^+^/CD49f^+^) prostate cancer stem cell populations. The activation of HIF pathway in these cells was shown to be essential for induction of PI3K, AKT, mTOR pathway which plays a critical role for cancer stem cell quiescence and maintenance by attenuating cancer stem cell metabolism and growth via mTOR and promoting cell survival by AKT signaling [608]. Prostate CSCs appear to be resistant to mTOR inhibitors due to HIF-1 alpha upregulation [608].

WNT/β-catenin signaling activation plays a major role in prostate cancer stem cells. In normal prostate, WNT/β-catenin signaling plays an essential role for stem cell specification, proliferation and homeostasis during development. WNT/β-catenin signaling is required for basal progenitors to generate luminal cells in adult mice. A recent study provided clear evidence about a key functional role of WNT signaling at the level of prostate tissutal niches. Particularly, it was shown that stromal cells near the proximal prostatic duct near the urethra synthesize and release various WNT ligands and display strong WNT activity [609]. The noncanonical WNT ligand WNT5a, released by stromal cells, exerts an inhibitory effect on epithelial stem cells, while stromal WNT/β-catenin signaling indirectly suppresses prostate stem cells through the TGF-β pathway [609]. These observations support the view that these WNT-mediated pathways exert a restrictive effect on prostate stem cells in a restricted tissutal area. Recent studies have highlighted the key role of WNT signaling in prostate cancer development and progression and in the maintenance of cancer stem cells [610]. Several abnormalities of the WNT signaling pathway have been described in prostate cancer: (i) genetic changes of APC and CTNNB1 are observed in 20–25% of CRPC patients and are responsible for the activation of the β-catenin dependent canonical WNT signaling; (ii) mutations involving the ubiquitin ligases RNF43 and ZNRF3 and gene fusions increasing RSPO2 expression are observed in ~6% of metastatic CRPC; (iii) activation of β-catenin-independent, noncanonical, WNT signaling is also observed in CRPC; and (iv) activation of WNT signaling is promoted by prostate cancer stroma, through the secretion of WNT proteins that activate WNT signaling prostate tumor cells [610]. Several recent studies indicate a role of WNT signaling in prostate cancer stem cells. Yun and coworkers showed that genetic silencing of the tumor suppressor gene DAB21P in human prostate epithelial cells induced the generation of CSCs with activated WNT-β signaling [552]. In these cells, WNT induced CD44 expression, directly interacting with the promoter of this gene [611]. Bioinformatic analysis of transcriptomic data predicts that WNT/β-catenin activation and its interaction with AR may play a role in the development of Enzalutamide resistance. To analyze the real impact of this prediction it was provided evidence that WNT/β-catenin was upregulated in enzalutamide-resistant prostate cancer cells, through a mechanism in part due to inhibition of a proteasome pathway (E3 Ligase β-TrCP) involved in β-catenin degradation [612]. Importantly, inhibition of the WNT/β-catenin pathway in enzalutamide-sensitive prostate cancer cells resulted in Enzalutamide resistance and in the acquisition of stemness properties (including OCT4, SOX2, CD44, ALDH1A, and ABCB1) [613]. Analysis of clinical datasets supported a molecular shift at various stages of prostate cancer progression, indicating the existence of a consistent correlation between β-catenin and AR expression [613].

It is important to note that loss of CHD1, observed in ~15% of primary prostate cancer cases, is rarely deleted in other cancer types, suggesting a prostate-specific role as a tumor suppressor of prostate tumorigenesis [614]. Initial studies have provided evidence that, in contrast to PTEN prostate conditional models, which generate highly invasive adenocarcinomas, homozygous deletion of CHD1 in prostate progenitors causes only low-grade PIN lesions [148]. A recent study provided evidence, through the analysis of prostate organoids, that CHD1 loss drives prostate tumorigenesis by modifying AR binding at the level of lineage-specific enhancers and, particularly, shifts AR chromatin occupancy to trigger HOXB13-directed oncogenic transcription, in a context also dependent on the function of other tumor suppressors [614].

A major tool in the study of naturally occurring tumors derives from the possibility to grow primary tumor specimens into immunodeficient mice, maintaining its original characteristics. However, this approach is considerably limited in prostate cancer, given the difficulty to grow primary prostate tumors into immunodeficient mice. In order to bypass this important limitation Toivanen and coworkers have developed a peculiar methodology enabling to grow primary prostatic tumors. This methodology was based on the observation that some studies have shown that adult epithelial stem cells isolated from murine or human prostate require the presence of embryonic mesenchyme to differentiate into mature prostate epithelium [434,615,616,617]. The instructive potential of stroma was further supported by the observation that this tissue was able to instruct embryonic stem cells to differentiate into prostate epithelium [434,615,616,617]. Therefore, these studies carried out on the normal prostate stem cell counterpart have shown that the supportive effect of stromal cells is strictly required to promote the growth and prostate differentiation of epithelial cells. Given these observations, it seemed logical to evaluate the effect of stroma to promote the growth of prostate cancer cells. Therefore, Toivanen and coworkers have attempted to define a prostate cancer bioassay by mixing prostate cancer cells isolated from primary prostate tumors with murine seminal vesicle mesenchyme (SVM) and to inoculate this cell combination encapsulated into collagen gel under the renal capsule of mice, which is highly vascularized. The survival and proliferation of localized prostate cancer tissue grafted alone or recombined with SVM was compared [559,560]. Tumor cell survived in a greater proportion of grafts and of patients and proliferated more when tumoral tissues were recombined with SVM compared to tumoral tissues grafted alone [618,619]. Furthermore, although technically more complex, the subrenal grafting ensures a significantly greater grafting efficiency than the subcutaneous implantation of prostate tumoral tissues.

During these last years, various preclinical models have been developed to study the pathophysiology of prostate cancer [620]. Historically, the first models were based on prostate cancer cell lines obtained from clinical metastatic lesions: however, this model had intrinsic limitations for preclinical studies, mainly related to the growth of tumor cells in a two-dimensional monolayer culture [620]. The development of new tissue engineering techniques allowed to obtain the generation of patient-derived xenografts from primary or metastatic prostate cancer lesions. The rate of generation of xenografts from primary prostate cancer specimens is limited, due to the low tumor proliferation, lack of stroma, and site of xenograft [621]. The technique of prostate cancer xenograft generation was improved by the coinjection in highly immunodeficient mice of prostate tumor specimens and extracellular matrix, compared to the simple injection of fresh tumor cells in immunodeficient mice [561,562,563]. Xenografts offer the unique advantage to provide a reproduction of the original tumor in terms of molecular abnormalities, cellular complexity and histological features [620,621,622]. Alternatively, to xenografts, organoid and spheroid prostate cancer three-dimensional cultures provide more feasible preclinical models, retaining the mean features of the original tumors [620,621,622].

The study of a model based on primary tumor cells was used to explore the hypothesis that prostate cancer recurrence was due to the presence in localized tumors of tumor stem cells that survive to castration. This study was carried out on 12 primary prostate cancer specimens: all these tumors reproduced the characteristics of the originary tumors [623]. These tumors were used to explore the effect of castration: short-term castration of these mice resulted in reduced proliferation and increased apoptosis of tumor cells [623]. After 4 weeks of castration, residual populations of stem-like cells survived to treatment [623]. In the absence of a specific treatment, these castration-resistant stem-like cells resulted in regeneration of the tumor [623]. These results indicate that residual populations of quiescent stem-like cells persist to treatment and may act as precursors of castrate-resistant prostate cancer and represent important targets for future specific therapies.

The finding that castration-resistant stem-like cancer cells are pre-existent to androgen deprivation therapy has important implications and suggests also that this therapy should lead in the time to a selection and to an increase of the number of these cells. To explore this important issue, Lee and coworkers investigated the effect of ADT in vitro at the level of some prostate cancer cell lines, in vivo in immunodeficient mice implanted with human prostate cancer cells and in vivo in human patients with prostate cancer undergoing standard treatment, providing evidence in all these conditions about an increase of the number of stem-like cancer cells. Particularly, the study of prostate cancer tissue in patients undergoing ADT showed that after therapy in the tumoral tissue an increase of CD133^+^/CD44^+^/CK5^+^ cells was observed, concomitantly with a decrease of CK8^+^ cells [624]. The analysis of prostate cancer tissues showed that they contain 1–1.5% of castration-resistant stem/progenitor cells and 98.5–99% of nonstem/progenitor cells [624]. The analysis of AR expression at the level of these two cell populations showed that the stem/progenitor cells have very low AR expression compared to the nonstem/progenitor cell population [624]. In addition, it provided evidence that AR could differentially function in these two tumoral cell populations: in fact, the overexpression of AR at the level of stem/progenitor cells elicited an inhibitory effect on cell growth, while an opposite effect was induced in nonstem/progenitor cells [624]. According to these findings it was suggested the combined targeting of the stem and nonstem cell populations is required to obtain an efficacious therapy of prostate cancer and particularly to bypass the development of castration-resistant disease.

The development of a large number of patient-derived xenografts from prostate cancer patients resistant to second-generation antiandrogens, such as abiraterone and enzalutamide, offers a unique platform for the identification of more effective therapies for CRPC [625]. These xenografts were representative of the heterogeneous population of CRPC, including AR mutations, genomic structural rearrangements of AR gene and a neuroendocrine-like AR-null disease [625]. Despite this consistent heterogeneity, all the xenografts were sensitive to the combination of ribosome-targeting agents CX-5461 and CX-6258 [625]. This methodology can be used to define the presence of castration-resistant cells in prostate cancer specimens. Thus, Porter et al. reported the existence of castration-resistant cells in patient-derived xenografts obtained from patients with intraductal carcinoma of the prostate relapsing after androgen deprivation therapy [626].

Additional in vitro studies on prostate cancer cell lines supported also the concept that androgen deprivation may lead to functional enrichment of putative prostate cancer stem cells. Thus, Seiler and coworkers using three different prostate cancer cell lines showed that androgen deprivation (first obtained in vitro by androgen deprivation and then in vivo by passage into castrated mice) determined the enrichment of putative cancer stem cells displaying enhanced expression of pluripotency transactivators, potentiated in vitro and in vivo tumorigenicity [627]. The CD44^+^CD24^-^ cells isolated through this procedure displayed high tumorigenicity when transplanted into immunodeficient mice and show limited differentiation capacities [627]. It is also important that SOX2 overexpression in prostate cancer cell lines resulted in androgen-independent growth [627].

Therefore, it appears evident from these studies that ADT leads to the suppression of prostate cancer nonstem/progenitor cells but determines an unwanted expansion of stem/progenitor cells, thus explaining the initial efficacy of this therapeutic strategy, but its incapacity to cure this cancer.

In addition to the development of cultures from primary prostate cancer samples it is also of fundamental importance the development of suitable animal models of castration-resistant prostate cancer, particularly in view of the identification of new suitable therapeutic targets. The development of this model was greatly stimulated by the investigation of new genetic pathways underlying the development of resistance to androgen deprivation therapy. Thus, Wang and coworkers have recently reported the identification of ZBTB7A, a transcription factor of the Pokemon family, as a tumor suppressor of prostate cancer, whose inactivation (occurring in a subset of advanced human prostate cancers) leads to a marked acceleration of PTEN loss-driven prostate tumorigenesis [302,628]. ZBTB7A physically interacts with SOX9 and functionally antagonizes its transcriptional activity on key target genes; *ZBTB7A* inactivation determines Rib downregulation, bypass of *PTEN* loss-induced cellular senescence and invasive prostate cancer [628]. The study of the various genetic mouse models of prostate cancer along with clinical data directly derived from patients allowed to identify new mechanisms of castration resistance. In fact, Lunardi and coworkers reported that androgen deprivation was sufficient to counteract tumor progression in prostate cancer model driven by *PTEN* loss alone. However, this response to androgen deprivation is lost in prostate cancer mouse models driven by *PTEN* loss together with *p53* or *ZBTB7A* loss, determining the development of castration-resistant prostate cancer [629]. The integrated acquisition of molecular data from these mouse models and from patients allowed to identify three genes, *XAF1*, *XIAP*, and *SRD5A1*, playing a major role in castration-resistance [629]. The combined inhibition of XIAP, SRD5A1 and AR pathways overcomes castration-resistance [629].

In conclusion, in spite the numerous studies devoted to defining the nature of prostate cancer stem cells and their possible role in androgen resistance, the cellular origin of prostate cancer still remains unclear. Studies on cancer stem cells have suggested that either a basal or a luminal stem cell as the cell of origin of prostate cancer. Future studies will improve our understanding of normal prostate stem cells and will help to define the prostate epithelial hierarchy. This step will enable to improve the definition of the malignant counterpart of these cells. Anyway, any future development on prostate cancer stem cells has to take into account both the heterogeneity existing within the whole tumor cell population and the plasticity existing at the level of individual cells.

Another point of crucial importance is related to the definition of the cellular and molecular mechanisms responsible for the development of resistance to androgen deprivation therapy. The studies carried out during the last years have involved two different models to explain the development of castration resistance: adaption and selection. While the adaption model implies the acquisition by tumor cells of new alterations, enabling them to survive in a condition of androgen deprivation, the selection model directly involves cancer stem cells. In fact, this last model involves the existence of rare cells, tumor stem cells, pre-existing to castration therapy and outgrowing because capable of surviving hormonal therapy. Therefore, it is evident that targeting of these cells is essential to prevent the development of castration-resistant prostate cancer.

## 15. Novel Therapies for Prostate Cancer

The progress in the elucidation of the genomic, biochemical, and metabolic alterations of prostate cancer at various stages of disease development has promoted the identification of new potential therapeutic targets. This has led to the investigation of novel prostate cancer therapies: in these new therapeutic approaches the element of novelty is represented by introduction of new drugs or of biomarkers to select patients to be treated (biomarker selection trials).

Since AR is the key driver pathway of prostate cancer development, it is not surprising that the most consistent efforts in the development of new therapies consisted in identification and clinical study of new agents blocking AR, suitable for the treatment of patients during the stages of androgen dependency and independency of the disease. This has led to the first generation, second generation, and third generation androgen inhibitors and the registration of Abiraterone, enzalutamide, apulatamide, and more recently Darolutamide. The clinical benefit deriving from the administration of these drugs was discussed above.

In addition to the traditional methods of inhibiting the AR pathway at various levels using chemical compounds, novel methods have been developed using antisense oligonucleotides (ASO). An important advantage of these agents is their capacity to target both the full-length and the splice variants transcript of a gene. Thus, Yamamoto and coworkers designed ASOs able to target exon-1, intron-1 and exon-8 in AR pre-mRNA to knockdown both AR_FL_ and AR-Vs: these ASOs suppress the growth of Enzalutamide-resistant prostate cancer cell lines and xenografts [630]. Xiao and coworkers have recently performed a study based on the combined targeting of EZH2 and AR ASOs and in suitable preclinical models provided evidence that this combined treatment with ASO is a promising strategy for the treatment of CRPC [631].

A first clinical trial was based on EZN-4176, a third generation LNA-ASO targeting the ligand binding domain of AR mRNA and resulting in full-length AR mRNA degradation and decreased AR expression: the administration of this ASO to 22 progressing CRPC patients has shown very limited antitumor activity [632].

Another alternative therapy based on AR targeting is represented by the Bipolar Androgen Therapy (BAT). This therapy is based on the observation that reintroduction by supraphysiologic doses of androgen is tumoricidal for prostate cancer cells, promoting DNA double-strand breaks and cell death [633]. The development of this therapy strategy is based on the observation that administration of testosterone to patients with CRPC could result in antitumor responses [633]. Preclinical studies have shown that biphasic responses to testosterone, with proliferation at low androgen concentrations (≅150 ng/dL) and growth inhibition at supraphysiological (≅1500 ng/dL) testosterone concentrations; using the exposure to these different androgen concentrations adaptive changes in AR expression are blunted, thus delaying the emergence of resistance [633]. The mechanisms exactly responsible for growth inhibition of high testosterone doses remain largely unknown [633]. The first clinical trial with this approach was carried out by Schweitzer et al. in 16 patients with asymptomatic mCRPC treated with 400 mg of testosterone intramuscularly monthly, reporting a PSA decline ≥50% in one third of patients and a radiographic response in 50% of patients; four patients remained on treatment for >1 year; at progression, ADT or AR inhibitory therapy induced a response in 100% of patients [634]. A recent phase III classical study evaluated the response to BAT treatment in men with CRPC after progression on Enzalutamide: 30% of patients had a ≥50% PSA decrease and 15/21 patients moving to an Enzalutamide rechallenge achieved a PSA response [635]. Additional studies will be required to assess the potential clinical role of BAT in the management of mCRPC and to define the optimal strategy for the alteration of androgen and antiandrogen therapies in mCRPC [635]. Additional studies will be required to assess the potential clinical role of BAT in the management of mCRPC and to define the optimal strategy for the alternation of androgen and antiandrogen therapies in mCRPC [635].

Other studies were based on the use of some new drugs that could restore the sensitivity to AR-signaling inhibitors in CRPC. As above reported, EZH2 inhibitors are able to epigenetically reprogram CRPC by restoration of AR expression and resensitization to AR-signaling inhibitors. Ongoing phase I/II trials, such as Pro STAR trial, are being conducted using the EZH2 inhibitor CPI-1205, inhibiting both WT and mutant EZH2. The ProSTAR trial is evaluating patients with mCRPC treated with the combination of CPI-1205 and either Abiraterone or Enzalutamide: preliminary results on the phase I of this trial presented at the 2019 AACR Meeting showed an encouraging clinical activity and warrant a phase II study at optimal CPI-1205 dosages.

Prostate-Specific Membrane Antigen (PSMA) is a type II membrane protein expressed in all forms of prostate tissue, including carcinoma. Radiolabeled PSMA conjugates represent an important diagnostic and therapeutic tool for prostate cancer. Thus, ^68^Ga-PSMA in radio imaging has shown a 42% detection rate of occult metastatic disease and detection has been very high (>95%) when PSA levels are higher than 2 ng/dL [636]. Lutetium-177 (^177^Lu)-PSMA is a radiolabeled small molecule that binds to PSMA enabling beta particle therapy to metastatic prostate cancer lesions [637]. A recent study provided a first evaluation of the safety, efficacy and effects on the quality of life in a group of mCRPC patients, previously treated with chemotherapy and or ADT or AR inhibitors [637]. 57% of the 30 treated patients achieved ≥50% PSA reduction; the treatment was well tolerated and 13% of patients developed grade 3 or 4 thrombocytopenia; most treated patients had a reduction of pain [637]. These encouraging results support randomized controlled studies to better assess the efficacy of (^177^Lu)-PSMA treatment compared to current standards of care.

Prostate cancer patients beneficiated in these last years by radiological techniques based on alpha particle-emitting alpha isotopes: alpha particles have a higher level of radiological effectiveness than beta emitters, since require fewer particle tracks to induce cell death [638]. Radium 223 dichloride is first-in-class, and is a commercially available targeted alpha therapy approved for the treatment of patients with mCRPC with bone metastases providing an overall survival benefit [638].

The optimal duration of androgen suppression therapy, associated with radiotherapy, in the attempt to obtain a curative effect of locally advanced prostate cancer remains still unclear. However, the long-term results of two important studies have in part contributed to clarify this complex issue. Thus, the RADAR randomized phase 3 study reported the long-term results, showing that 18 months of androgen suppression plus radiotherapy is a more effective therapeutic option for locally advanced prostate cancer than 6 months of androgen suppression plus radiotherapy, but the addition of zolendronic acid (a bisphosphonate compound used to prevent skeletal fractures in patients with cancer) to this regimen is not beneficial [639]. The long-term report of the study NRG/RTOG 9413 provided evidence that whole pelvic radiotherapy (WPRT) plus neoadjuvant hormonal therapy (NHT) improved progression-free survival in patients with intermediate-risk or high-risk localized prostate cancer compared with treatments based on prostate only radiotherapy (PORT) and NHT or WPRT plus adjuvant hormonal therapy (AHT) or PORT plus AHT [640].

It is important to point out that the radiotherapy treatments of Gleason score 9–10 prostate cancer patients treated with radical prostatectomy and ADT involve either adjuvant external beam radiotherapy (EBRT) alone or ERBT and brachytherapy. The analysis of a large set of patients carried out at the Chicago Prostate Cancer Center provided evidence that both these treatment procedures, either including or not brachytherapy, lead to an equivalent risk of prostate cancer-specific mortality and all-cause mortality [641].

Recent studies have explored the sensitivity of prostate cancer to immunotherapy based on immunechek inhibitors. As above discussed, these studies have shown only a limited rate of responses to anti-PD-1 agent (Pembrolizumab), with an overall response rate in not more than 5% of prostate cancer patients [407]. Anti-PD1 blocking agents displayed a high response rate in tumors with mismatch repair deficiency, regardless of primary site, thus leading to a tissue agnostic FDA approval. Analysis of the prevalence of MSI-High/dMMR in prostate cancer patients showed a global prevalence of 3.1% (2.2% with high MSI sensor scores) [642]. Eleven patients with MSI-H/dMMR CRPC were treated with anti-PD-1/PD-L1 therapy; 60% of these patients responded to treatment with ≥50% PSA decline and five of these patients remained on therapy for as long as 89 weeks [642].

Although the treatment of castration-resistant prostate cancer remains an unresolved issue, recent studies have identified peculiar pathways activated in these tumor cells that could be targeted for a therapeutic effect. A diverse array of molecular determinants has been reported accounting for resistance of prostate cancer cells to ADT and, among them, a key role is played by alterations in cofactor activity [162,445,643]. FOXA1 targeting may represent a potential tool for these new therapeutic interventions. FOXA1 is a transcription factor essential for epithelial lineage differentiation. The *FOXA1* gene in one of the genes most frequently mutated in prostate cancer, its mutation being more frequent in metastatic than in primary tumors [55]. In addition to mutations in the coding region, mutations of the gene promoter were reported in ~5% of advanced prostate cancers [65]. Interestingly, a study by Annala and coworkers identified *FOXA1* 3′-UTR mutations in 12% of mCRPC patients: the mutations were predominantly insertions or deletions, covering the entire UTR without motif enrichment [644]. These mutations were not detected in other cancers [644]. In addition to these genetic alterations, *FOXA1* is downregulated in CRPC as compared with primary prostate cancer, suggesting a tumor suppressor role [645]. FOXA1 is a key modulator of AR-regulated transcriptional signaling [646]. *FOXA1* downregulation observed in CRPC triggers a complete reprogramming of the hormonal response by causing a shift in AR binding to a distinct cohort of enhancers [646]. A delicate balance between nuclear FOXA1 and AR protein levels for their cooperation in the activation of prostate-specific AR transcriptional program [647]. FOXA1 downregulation triggers epithelial–mesenchymal transition and cell differentiation to neuroendocrine phenotype [648,649]. *FOXA1* loss determines TGFβ3 up modulation, EMT transition, and cell motility, all are events blocked by the TGFβ inhibitor Galunisertib [650]. Studies in prostate cancer specimens confirmed reduced FOXA1 levels and increased TGFβ signaling in CRPC specimens compared to primary tumors [650]. Combinatorial treatment with Galusirtenib sensitized prostate cancer cells to Enzalutamide [650]. These observations support a role for FOXA1 as a regulator of prostate cancer plasticity, in part mediated by TGFβ signaling and supports a strategy to control these events and, potentially, to potentiate the clinical response to antiandrogen therapies [650]. The effect of FOXA1 on AR signaling is dichotomy in that increasing FOXA1 activity induces non-selective opening of closed chromatin, inducing the binding of AR to ARE half-sites at expense of gens with canonical ARE that promote cancer progression; in contrast, FOXA1 inhibition reprogrammed AR binding to AREs, leading to overexpression of some androgen-responsive genes, thus promoting CRPC cell growth [648]. The dichotomous function of FOXA1 in AR signaling may be explained taking into account the key role of this transcription factor as a pioneer factor in the reprogramming of the *AR* and *GATA2* cistromes: FOXA1 represses AR binding to DNA, while GATA2 cooperates with the AR in androgen-mediated gene expression in prostate cancer [651].

GATA2 plays a key role in driving prostate cancer aggressiveness: GATA2 acts as a pioneer transcription factor that increases AR binding and activity and regulates a core subset of clinically relevant genes in an AR-independent manner, thus suggesting that GATA2 is a potential target in prostate cancer therapy [652]. GATA2 specific inhibition using the small compound K-7174 lowered AR expression and the proliferation of CRPC cells [653,654]. The study of AR-variant transcriptome is in part dependent upon FOXA1 [655]. A recent study showed that GATA2 is a critical regulator of AR-Vs [292]. Interestingly, the *GATA2* cistrome shares a consistent overlap with bromodomain and extra terminal (BET) proteins and is codependent for DNA binding; in line with this observation, BET inhibitors compromise GATA2 activity in CRPC cells [292]. These observations support the use of BET inhibitors in CRPC patients overexpressing *GATA2* [292].

Several recent studies addressed a considerable interest for BET inhibitors as potential drugs for treatment of CRPC patients. BET proteins mediate acetyl addition and, as a result, transcription. Three types of proteins regulate lysine acetylation: histone acetyltransferases that are the writers; BET proteins that are the readers and deacetylases and sirtuins that are the erasers. The BET family of proteins consists of four members: BRDT primarily expressed in germ cells and BRD2, 3, and 4 ubiquitously expressed: these proteins interact with acetylated histones and regulate transcription. A fundamental study by Asangani and coworkers showed that BRD4 physically interacts with the N-terminal domain of AR and this interaction can be blocked by the JQ1 BET inhibitor, with consequent inhibition of AR-mediated transcription [288]. In vivo, BET inhibition was more efficacious than direct AR antagonism [288]. The essential role of histone acetylation in CRPC phenotype is further supported by other fundamental observations showing that inhibitors of p300/CBP acetyltransferases inhibited the AR-mediated transcriptional progress in both AR-sensitive and AR-resistant prostate cancer cells and tumor growth in a CFRPC xenograft model [656,657].

Other studies have explored the expression and function of BET domain factors in prostate cancer cells. Thus, Welti and coworkers explored BRD2, BRD3, and BRD4 expression at various stages of prostate cancer development. BRD4 protein expression increases significantly with castration resistance development, as shown by longitudinal studies in individual patients; BRD2, BRD3, and BRD4 mRNA expression correlates with AR-driven transcription [658]. Importantly, this study showed also that BET inhibitors were able to reduce expression and signaling of AR-V7 [658]. Urbanucci et al. have explored chromatin accessibility at various stages of prostate cancer development and provided evidence that increases with tumor progression and is fully developed in CRPC due to increased expression of AR and BET domain proteins [659]. *BRD2*, *BRD4*, and *ATAD2* (a chromatin regulator harboring an ATPase domain and a bromodomain) are overexpressed in CRPC [659]. Interestingly, this study defined also a gene stratification signature (BROMO-10) for bromodomain response [659]. Coleman and coworkers have screened a set of prostate cancer cell lines to identify transcriptional pathways modified by BET inhibitors in prostate cancer cells [289]. Bet domain inhibition with the JQ1 inhibitor suppresses the growth of a large panel of cell lines, including those that are Enzalutamide-resistant or AR-independent [289]. Interestingly, these authors identified a number of transcription factors, CBX3 (Chromobox protein homolog 3), MCM2 and MCM5 (Minichromosome Maintenance Complex Component 2 and 5) whose modifications of expression/activity are essential for mediating the antitumor effects of BET inhibitors in CRPC cells [289]. In view of possible clinical applications, it is of fundamental importance to identify highly active BET inhibitors that can be used at low concentrations. In this context, two different reports described BET protein degraders able in vitro and in vivo to suppress AR and MYC signaling at lower concentrations than classical BET inhibitors [290,291].

Asangani and coworkers have explored in preclinical models the important issue of a possible synergism between BET inhibitors and AR antagonists [660]. Using a panel of Enzalutamide-resistant prostate cancer cell lines, these authors provided evidence that BET bromodomain inhibitors enhance efficacy and disrupt resistance to AR antagonists in the treatment of prostate cancer [660]. These observations support clinical studies based on the combined treatment with BET inhibitors and AR antagonists [660].

Some BET bromodomain inhibitors are under clinical investigation in CRPC and in other solid tumors. The BET protein inhibitor Birabresib was evaluated in monotherapy for safety and preliminary antitumor activity in 24 CRPC patients, showing no objective responses and a stabilization of disease in 63% of treated patients; the discontinuous treatment was better tolerated than the continuous treatment [661]. Other studies are exploring ENZ-3694, an orally bioavailable BET bromodomain inhibitor with preclinical activity in Enzalutamide-resistant CRPC models: this inhibitor downmodulates the expression of putative drivers of Enzalutamide resistance, including AR splice variants, glucocorticoid receptor, and MYC, and shows synergy with Enzalutamide. The results of a phase Ib/IIa clinical study evaluating the combination ZEN-3694+Enzalutamide in Abiraterone/Enzalutamide-resistant mCRPC. The preliminary results of this study presented at the AACR Meeting, April 2019, showed a good tolerability of this treatment and prolonged radiographic progression-free survival and longer duration of treatment were observed, comparing favorably with historical control of sequential AR targeting [662]. These initial observations warrant further clinical development of this drug combination [662]. Very interestingly, the study of a case report of a single patient treated with ZEN-3694 and Enzalutamide combination was very instructive about the clinical potentialities of this treatment [663,664]. This CRPC patient was treated with ZEN-3694 and Enzalutamide, following a peculiar program of drug dosage adjustments, carried out according to the technology platform CURATE [663]; this technology enabled individualized ZEN-3694 dosages, using markedly reduced ZEN-3694 doses compared to the starting dose [664]. This strategy contributed to obtain a durable treatment response in this patient [664].

The development of initial clinical studies based on the use of BET inhibitors in CRPC raised the problem of the adaptation of tumor cells to these drugs, related to the induction of resistance mechanisms. Pawar and coworkers have explored this issue showing that BET inhibitor-resistant CRPC cells display cross-resistance to a variety of BET bromodomain inhibitors in the absence of gatekeeper mutations; these resistant cells showed reactivation of AR signaling due to CDK9-mediated phosphorylation of AR, determining a condition of sensitivity to CDK9 inhibitors and Enzalutamide [665]. Furthermore, these cells showed also increased DNA damage associated with *PRC2*-mediated transcriptional silencing of DDR genes, leading to PARP inhibitors sensitivity [665].

Other recent studies support a possible role for heat shock protein 90 inhibitors in the therapy of CRPC. There is a link between AR and HSP90 in the normal physiology of this receptor. In fact, AR initially localizes in the cytoplasm in a molecular complex with heat shock proteins, cytoskeletal proteins and co-chaperone proteins. Following binding to androgens, AR translocates to the nucleus, where it binds at the level of ARERs in the promoters or enhancers of target genes with other coactivators. HSPs, expressed during stress conditions, facilitate the stabilization, folding, and translocation of their client proteins. HSP90 is one of the most important members of the HSP family, has more than 200 client proteins, including oncogenic proteins such as c-Met, v-Src, BCR-Abl, and Plk1. HSP90 regulates the stability and activity of AR forming a complex in the cytoplasm, inducing the stabilization of AR prior to binding with its ligand; HSP90 inhibitors lead to AR degradation and its accumulation in the cytoplasm [666]. HSP90 is overexpressed in prostate cancer; co-treatment of prostate cancer cells with AR antagonist Enzalutamide and a HSP90 inhibitor induces an increased rate of cell death due to a synergistic reduction of AR protein induced by the two drugs [667]. The downregulation of Polo-like kinase 1 (PLK1), a key regulator of many cell cycle events, induced by HSP90 inhibitors is essential for their antitumor effects [667]. HSP90 is involved also in the control of AR nuclear translocation through a mechanism involving its phosphorylation by the cAMP-dependent protein kinase A (PKA); in fact, activated PKA induces HSP90 phosphorylation that mediates the release of AR from HSP90, thus enabling AR binding to HSP27 and its migration in the nucleus [668]. Thus, activated PKA, a phenomenon frequently observed in advanced prostate cancer, potentiates and allows the AR activation event in the presence of low androgen concentration [668]. A small molecule inhibitor against HSP90 phosphorylation at the level of Thr-89 residue, in association with an AR antagonist, was more potent than androgen deprivation alone in inhibiting the proliferation of prostate cancer cells [668].

Jansson and coworkers have used prostate cancers developing in *PTEN/TP53* null mice to perform a high-throughput analysis of a large panel of drugs and reached the conclusion that HSP90 inhibitors are among the most active compounds and have clinical potential for use in drug combinations to enhance the efficacy of current drugs and to delay the occurrence of resistance [669]. These findings were confirmed in 15 CRPC-derived organoid models [669]. Ganetespib, an HSP90 inhibitor, targets several oncogenic proteins in prostate cancer cells, including AR and PI3K/AKT pathway; this drug, combined with castration, induced more pronounced tumor regressions and delayed castration resistance relative to either monotherapy [669]. However, a phase II clinical study of Ganetespib used as a monotherapy in CRPC patients, previously treated with Docetaxel, showed only minimal clinical activity [670]. The reasons of this low clinical efficacy of Ganetespib could be related to the selection of a heavily pretreated patient population and lack of pharmacologic potency in mCRPC [670]. A second-generation of SSP90 inhibitors was developed and one of these compounds, Onmalespib, was found to be very active in preclinical models and was beneficial in prostate cancer models expressing AR-V7 [671]. However, in spite this promising preclinical background, Onalespib in combination with Abiraterone failed to show any significant clinical activity in CRPC patients in progression [672].

The conclusion deriving from this study was that, while being a promising area in preclinical studies, the clinical applications of HSP90 inhibitors in prostate cancer have shown very limited clinical activity.

Other studies are exploring a possible role of the targeting of the protein MDM2 (an E3 ubiquitin-protein ligase), a negative regulator of p53, as a tool to improve the response of prostate cancer cells to AR antagonists. *MDM2* was overexpressed in 44% of prostate cancers [673]. Global MDM2 overexpression does not correlate with disease outcome [674]; however, high-MDM2 score was associated with distant metastasis and mortality in patients treated with radiotherapy and androgen deprivation for prostate cancer [674]. A recent study provided evidence that MDM2 is an important regulator of prostate cancer stem cells. In fact, Giridhar and coworkers have shown that these cancer stem cells exhibit an AR^-^ signature, related to the effect in these cells of MDM2 promoting the constant ubiquitination and degradation of AR, resulting in a consistent loss of AR protein [675]. Furthermore, MDM2 promoted CSC self-renewal, expression of CSC markers and proliferation of CSCs [675]. Importantly, loss of MDM2 in these cells reversed MDM2-mediated processes, induced expression of FL-AR and induced differentiation to luminal cells [675]. These observations support the view that MDM2 inhibition, which is associated with AR targeting, may represent a strategy for killing cancer stem cells [675].

Other studies confirmed treatment of prostate cancer cells with MI-219, an inhibitor of MDM2 results in tumor sensitization to radiation or androgen deprivation therapy in vitro and in vivo [676]. A clinical phase Ib trial is evaluating the MDM2 inhibitor Idasanutlin in association with Enzalutamide or Abiraterone/Prednisone in CRPC patients [677].

As above-discussed, ~2% of prostate cancers at diagnosis exhibit a neuroendocrine phenotype; however, following treatment with AR pathway inhibitors, ~20–25% of patients relapse with tumors that have lost their AR dependence and have neuroendocrine features [678]. NEPCs have a limited number of genetic alterations, with the exception of *MYCN* amplification, *TP53* mutation or deletion and loss of *RB1*, thus supporting the view that the acquisition of a neuroendocrine phenotype is mainly driven by epigenetic dysregulation [678].

Importantly, the study of prostate cancers obtained through rapid autopsies from 2012 to 2016, compared to a cohort of tumors prior to the introduction of androgen signaling inhibitors (from 1998 to 2011) and observed a doubling (from 6.3% to 13.3%) in the percentage of patients with neuroendocrine features in the 2012–2016 cohort [215].

Neuroendocrine prostate cancers are intrinsically resistant to standard treatments and have a negative prognosis. However, several recent studies have shown that there are some exploitable vulnerabilities conferred by neuroendocrine differentiation.

A recent study showed that the NOTCH ligand delta-like 3 (DLL3) is expressed in >76% of neuroendocrine prostate cancers, while it is expressed in only a small subset of castration-resistant prostate adenocarcinomas and virtually not expressed in localized prostate cancers and in benign prostate tumors [679]. DLL3 can be targeted by the antibody-drug conjugate SC16LD6.5 (Rovalpituzumab Tesirine); treatment with this antibody resulted in complete and durable response in DLL3^+^ prostate cancer xenograft models [679]. Interestingly, a patient with neuroendocrine prostate cancer displayed a meaningful clinical and radiological response to SC16LD6.5 when treated in the context of a phase I basket trial [679]. The longitudinal study of some patients during tumor evolution suggests the appearance of DLL3 positivity concomitantly with induction of the neuroendocrine phenotype [679].

As above-discussed, a recent study by Reina-Campos et al. [92] led to the discovery of a potential metabolic vulnerability of NEPC. In fact, bioinformatic studies showed that NEPCs express low levels of PKCλ/ι-coding gene *PKRCI*; *PRKCI* low gene expression was shown to be essential for the development of NEPC phenotype [92]. The study of the mechanism through which PKCλ/ι deficiency causes NEPC development showed a role for ATF4-mediated transcriptional reprogramming with mTORC1 activation and expression of several genes involved in serine, glycine, and one-carbon (SGOT) metabolic network, such as *PHGDH, MTHFD2* and *PSAT1* [92]. The alterations in SGOT metabolism in turn caused a deregulation of the methionine cycle, with consequent changes of the levels of S-adenosylmethionine and changes in chromatin methylation [92]. These observations support the potential use of drugs lowering DNA methylation as a tool to reduce the NEPC phenotype and increase the responsiveness of these tumors to AR antagonists [92].

*MYCN* overexpression represents one of the molecular events frequently associated with NEPC. In 2011, Beltran and coworkers reported the frequent gene amplifications of *MYCN* and *AURKA* occurring in 40% of NEPC and 5% of prostate cancer adenocarcinomas, conferring a pronounced sensitivity of tumor cells to Aurora kinase inhibitors [680]. The study of a large set of treatment-related NEPCs showed *AURKA* amplification in 65% of primary prostate cancers and 86% of metastases and concurrent MYCN amplification was detected in 69% of treated prostate cancers and 83% of metastases [681]. Molecular and functional studies have defined the essential role of N-MYC and activated AKT as oncogenic components to induce the transformation of human epithelial prostate cells to NEPCs, with phenotypic and molecular features of aggressive, late-stage human disease [682]. N-MYC was shown to be required for tumor maintenance and its destabilization with Aurora A kinase inhibition reduces tumor growth [682].

The mechanisms through which N-MYC drives NEPC are largely unknown. A recent study by Zhang et al. showed that using comparative bioinformatic analyses of various types of prostate cancer, an enrichment of a MYC-N-PARP-DDR (DNA damage response) pathway in NPEC was observed [683]. The activity of BRCA1, PARP1 and PARP2 was shown to be important for NEPC cells [683]. Targeting this pathway with a PARP inhibitor (Olaparib) and an Aurora kinase inhibitor resulted in synergistic antitumor effects [683]. *N-MYC* overexpression contributes to develop a condition of drug resistance in NEPC cells in part through suppression of the serine/threonine kinase ATM (Ataxia Telangiectasia Mutated), mediated via miR-421 upregulation [684]. This finding suggested a possible therapeutic option for NEPC based on an ATM kinase inhibitor and Enzalutamide [684].

The Aurora kinase inhibitor Alisertib was explored in monotherapy in a population of mCRPC patients, a part with NEPC features [96]. Although the study failed its primary endpoint on the progression-free survival, it has, however, reported some exceptional responders among patients bearing MYC-N-deregulated tumors [96].

A consistent problem in the treatment of neuroendocrine prostate cancers is represented by the presence by the presence in these tumors of *RB1* loss, a genomic alteration associated with a poor response to available therapies [387,388]. Studies of monitoring of genomic alterations in cell-free DNA have clearly shown that in mCRPC patients’ resistance to Enzalutamide treatment was associated with RB1 loss as well as AE amplification and heavily mutated AR [685]. Interestingly, a recent study identified a pan-cancer transcriptomic signature for predicting *RB1* loss, using validated data sets from The Cancer Genome Atlas (TCGA) and Pan-Cancer [686]. High RB score (RBS) was associated in the whole cancer population with short progression-free survival, overall survival and disease-specific survival [686]. The study of RBS in mCRPC patients showed that a high RBS was strongly associated with RB biallelic loss; however, not all tumors with high RBS displayed biallelic *RB1* alterations and vice versa [686]. A high RBS was associated with short overall survival in mCRPC patients (median overall survival of 15.0 vs. 42.0 months, comparing RBS^high^ versus RBS^low^ patients) [686].

A recent study showed that either a biallelic *RB1* inactivation or a monoallelic RB1 inactivation was present in virtually all NEPCs, suggesting that this genetic abnormality is a hallmark of NEPCs; furthermore, a near mutual exclusivity with AR enhancer amplification, present in ~80% of mCRPC not bearing NEPC features, was observed in NEPC [243]. Another recent study provided clear evidence that RB1 alterations observed in CRPC are heterogenous, involving copy number deletions, structural variants and intragenic tandem duplication, involving multiple exons and associating with protein loss [76].

The main biologic function of RB1 consists in acting as a tumor suppressor, preventing excessive proliferation by blocking the G1 to S progression in the cell cycle, mediated through binding with transcription factors of the E2F family and consequent inhibition of E2F-regulated cell proliferation genes, such as cyclin A and cyclin E. The biologic activity of RB1 is regulated by its phosphorylation, induced by cyclin-dependent kinases (CDKs). However, in addition to its function as a main regulator of cell cycle progression, RB1 also has additional noncanonical functions, mediated by its capacity to interact with other transcription factors or chromatin regulators [687]. In this context, as above-reported [293], RB1 loss occurring at late stages of CRPC development is induced through E2F1 upregulation of the AR and increased recruitment of AR to its target gene promoters. Furthermore, high AR levels transcriptionally suppress DNA replication in prostate cancer cells; this activity is increased in CRPC cells expressing high AR levels and is mediated by recruitment of hypophosphorylated RB1; however, AR stimulates also DNA replication through AR hyperphosphorylation [688]. Consequently, blocking CDK4/6 activity prevents RB1 hyperphosphorylation and, through this mechanism, stimulates AR-mediated suppression of proliferation [688].

The treatment of RB-deficient tumors, including NEPC, remains a largely unmet clinical need. These patients are less likely to respond to CDK4/6 inhibitors since these drugs act by blocking RB1 phosphorylation in G1-phase of the cell cycle, triggering proliferation arrest and are thus suitable in tumors in which *RB1* is intact. Patients with RB-deficient tumors are most likely to respond to DNA damaging agents (such as platinum compounds) or alternative therapeutic strategies based on the targeting of noncanonical functions of RB1 or synthetic lethality [689]. Thus, it was shown that therapies aiming to reactivate the RB pathway may have tumor suppressive effects, not through a suppression of cell proliferation, but inducing the reversion of cell state changes associated with advanced tumor progression and related to RB1 deficiency [690].

Interestingly, two recent studies carried out in RB-deficient small cell lung cancer cells [691] and triple-negative breast cancer cells [692] have shown that Aurora A or B kinase inhibition is synthetic lethal with loss of the RB1 tumor suppressor gene. These observations support additional studies with Aurora kinase inhibitors in NEPC patients designed on the basis of molecularly defined inclusion criteria and distinguishing patients with *de novo* NEPC from those with late-occurring secondary NEPC. A recent study suggests that the type of Aurora kinase inhibitor used could play a relevant role in the definition of the pharmacologic effects on prostate cancer cells. In fact, Lake and coworkers, using a high-throughput method for tracking structural changes in Aurora kinase in solution, provided evidence that many clinically important inhibitors of this kinase trigger structural changes to their target kinase, promoting either the active DFG-in conformation or the inactive DFG-out conformation [693]. These conformational preferences may explain differential patterns of inhibitor selectivity in various tumor types [693]. Aurora kinase binds to an N-terminal segment of N-MYC, thus preventing its proteolytic degradation; N-MYC binds at the level of the activation loop of Aurora A kinase, trapping the kinase in an active DFG-in conformation; N-MYC binding can be disrupted by DFG-out kinase inhibitors [693]. The Aurora A kinase inhibitor Alisertib showed a limited efficacy in initial studies in prostate cancer [96] and this inhibitor promotes the DFG-out state, but its efficacy is limited by the level of N-MYC overexpression in tumor cells; the Aurora A kinase inhibitor AMG-900 is a strong DGF-out inducer and might be more effective in displacing N-MYC and in inducing an antitumor effect [693].

## 16. Prostate Cancer Models

Preclinical models are an essential tool for the study of tumoral processes and for the identification and evaluatiuon of new anticancer drugs. Prostate cancer is a particularly difficult cancer type to model for preclinical studies due to the complex pathogenesis and the presence of multiple stages of disease, characterized by different molecular features, drug resistance, resistance mechanisms, and effects of previous treatments. Numerous cellular-based preclinical models of prostate cancer have been developed that were of fundamental importance for the understanding of the pathophysiology of this tumor. It is of fundamental importance that preclinical models are able to capture and reproduce the tumor heterogeneity typically observed in spontaneously occurring cancers and this explains the evolution in the time of prostate cancer models, moving from cell lines to patient-derived xenografts (PDXs) and three-dimensional organoid cultures [620,622,694].

Historically, most of experimental studies on prostate cancer have been carried out with three cancer cell lines: PC-3, DU-145, and LNCaP. More recently, additional prostate cancer cell lines have been obtained using various methods. The properties of these cell lines were analyzed in detail by Namekawa et al. [620]. The cell lines have the davantege of unlimited growth in vitro and are suitable for high-throughput screening and tumor xenograft in vivo testing; however, the great limitation of these cancer cell lines is that they do not represent the heterogeneity of primary human tumors and acquire peculiar properties during the process of adaptation to grow in vitro [620].

To bypass the consistent limitations of cancer cell lines, patient-derived xenografts (PDXs) have been developed: PDXs allow the obtaining of preclinical results that better reflecting the therapy responses in patients. PDXs retain the molecular features of the prostate cancers from which they were generated, including mutations, genetic structural rearrangements and gene expression profiles. However, the successful generation of PDXs from human prostate cancer specimens is low and this is due to the low tumor proliferation, lack of stroma and site of xenograft [621,622]. The introduction of some technical procedures, such as the coinjection of prostate cancer tissue together with proteins of the extracellular matrix, transplantation into renal capsules, and the use of highly immunodeficient mice improved the success rate of generation of PDXs [620,621,622]. In spite all these technological difficulties, PDXs represent a unique model reflecting the heterogeneity of advanced prostate cancer and being suitable for accelerating prostate cancer discovery and drug development [695,696]. The progresses made at the level of development of PDXs from prostate cancer specimens allowed using these cancer models to explore some peculiar aspects of the biology of these tumors. Thus, the study of PDXs contributed to better understand the role of REST, a transcription factor inducing EMT in prostate cancer cells and stemness acquisition via direct transcriptional repression of Twist1 and CD44 [697]. Proteomics of a unique set of 17 prostate cancer-derived PDXs indicated a role of REST in regulating neuronal gene expression in prostate cancer cells, thus favoring neuroendocrine differentiation of these cells [698]. Other studies have shown that the analysis of some PDX models in serial transplantation assays allowed to develop models of prostate cancer metastatization and to analyze genomic and transcriptomic alterations during metastatic progression achieved by serial transplantation [699].

In 2018, the Movember Foundation presented the results of the GAP1 PDX project, reporting through an international cooperation the development of a total of 98 authenticated human prostate cancer PDXs, serially transplantable and characterizerd for their cellular and molecular properties [700].

A major limitation of the PDX model is the suppressed immune component in host mice used to grow these tumors.

PDX are limited by the long take rate, long duration of establishment (engraftment and drug validation in mice normally requires >6 months) and the elevated associated costs. Thus, it was developed by some rersearchers an alternative method, termed patient-derived explants (PDEs) and consisting in the ex vivo culture of freshly resected prostate tumor specimens obtained from surgery [701]. PDE provides a high-throughput model in which the tumor retains its native tissue architecture, microenvironment, cell viability and genetic alterations [701]. Thus, the PDE model provides a relatively simple and economic method for direct drug evaluation on an individual patient’s tumor [701].

During the last decade techniques to grow tissues in vitro in three dimensions (3D) as organotypic structures have been developed. These structures are known as organoids and have been established from a variety of human tissues, including prostate [494,495]. Organoid culture protocols have been established for patient-derived tumor tissue as well, including prostate cancer [478,536]. Thus, Gao et al. initially reported the growth of organoid cultures from prostate cancer specimens with an efficiency of 15–20%; importantly, these tumor organoids reflect the tumor genetic abnormalities and manifest the heterogeneity observed in the primary tumor from which they were derived [478,536].

Prostate cancer-derived organoids represent a precious tool to explore some biological properties of these tumors. Thus, Park and coworkers have used prostate organoids to demonstrate that c-Myc/AKT1-transduced luminal organoids exhibit histological features of well-differentiated acinar adenocarcinoma, with AR and PSA expression, while c-Myc/AKT1-trasduced basal organoids are biologically more aggressive, with a loss of acinar morphology, low/absent AR and PSA expression [702]. Puca and coworkers have used prostate cancer organoids to better define the biology of prostate cancers with neuroendocrine differentiation [703]. Thus, four organoids developed from needle biopsies of metastatic lesions showed tumors displaying neuroendocrine features in concordance with the tumors of origin [703]. These organoids were studied to assess the functional impact of genes involved in CRPC-NE pathogenesis, highlighting the key role of EZH2: the inhibition of this gene resulted in a downregulation of neuroendocrine pathway genes and of those associated with stem cell and neuronal pathways [703]. In these NEPC organoids N-Myc is a key driver of neuroendocrine differentiation with induction of epigenetic reprogramming, acting in cooperation with AR-cofactors FOXA1 and HOXB13, at the level of genomic loci implicated in neural lineage [704]. Interestingly, EZH2 inhibition reversed the N-Myc-induced suppression of epithelial lineage genes [704].

The study of prostate cancer organoids allowed to show that PTEN ablation is associated with higher expression of the Brahma-related gene 1 (BRG1), an ATPase subunit of the SWI/SNF chromatin remodeling complex. BRG1 depletion inhibited the progression of PTEN-deficient prostate tumors [705].

Prostate cancer organoids were used to define the role of lipid metabolism in prostate cancer cells. Studies on PDXs and tumor organoids showed that fatty acid uptake is increased in prostate cancer in part mediated through overexpression of the fatty acid transporter CD36 [706]. Combined dual targeting of fatty acid uptake and de novo lipogenesis potently inhibited proliferation of prostate cancer-derived organoids [706].

Prostate cancer organoids can be obtained also from genetically engineered mouse models of prostate cancer, thus improving the potential impact of these models in the understanding of cancer biology [707]. Furthermore, organoids can be generated from PDX models introducing some changes in the culture conditions for the growth of organoids and 50% of these PDX-derived organoids can be cultured long-term [708]. Finally, the introduction of primary prostate stromal cells increased organoid formation from normal prostatic tissue and directed organoid morphology into a branched acini structure similar what observed in vivo; the co-culture with stromal cells improved the vitality also of cancer-derived organoids [709].

## 17. Emerging Topics and Conclusions

In conclusion, during the last years considerable progresses have been made in our understanding of the molecular and cellular basis of prostate cancer. However, in spite these progresses, our understanding of prostate tumorigenesis, as well as of the factors that trigger and initiate prostate tumor development still remains very limited. Particularly, it remains unclear whether prostate cancers originate from the malignant transformation of luminal or basal cells and in various experimental models both luminal and basal cells are vulnerable to oncogenic transformation and can originate prostate cancers.

Prostate regeneration experiments following castration have supported the existence of prostate stem cells, displaying either basal or luminal cell characteristics.

Treatment of advanced prostate cancer and, particularly, of CRPC remains an unresolved medical problem. Progresses have been made in the understanding of the cellular and molecular mechanisms underlying the development of castration resistance. The profiling of primary tumors allowed for the defining of genetic alterations that drive prostate cancer. Thus, recent studies have supported the routine use of tumor and germline DNA proling for patients with advanced prostate cancer, with the specific aim of guiding enrollment in targeted clinical trials. In this context, particularly relevant and instructive were two recent studies.

A first study was carried out at the Memorial Sloan Kettering Center of New York [710]. The study, based on targeted deep sequencing of tumor and normal DNA fron 451 patients with prostate cancer (locoregional prostate cancer, metastatic noncastrate, and CRPC) identified single nucleotide variation, small insertions and deletions, copy number alterations, and structural rearrangements in over 300 cancer-related genes [710]. Importantly, potentially actionable alterations were identified in DNA damage repair (27% in either germline or somatic), PI3K (34% of patients), and MAPK pathway and MMR genes (3%) [710]. The comparative analysis of various tumor stages showed an increase in tumor mutational burden from localized prostate cancer (1.74 mutations/megabase) to noncastrate resistant prostate cancer (2.08 mutations/megabase) and to mCRPC (4.02 mutation/megabase) [710]. The genes more altered in mCRPC compared to localized prostate cancer included AR amplification/mutation and alterations of *TP53*, *RB1*, *PTEN*, *APC*, *ATM*, *FANCA*, and *CDK12* [710]. In contrast, mestatic noncastrate resistant prostate cancers displayed a frequency of the major genetic alterations similar to that observed in localized prostate cancers, with the exception of a moderate increase of alterations of *PTEN* (18% vs. 12%), *RB1* (7% vs. 2%), *APC* (14% vs. 4%), *KMT2C* (9% vs. 4%), and *CDK2* (6% vs. 4%); the frequency of *AR* alterations remained low in these tumors (4%) and much lower than in mCRPC (54%).

The second study was a multi-institutional study providing the largest available comprehensive genomic profiling on advancer prostate cancer (1160 primary site and 1816 metastatic site tumors) performed at the Foundation Medicine, Cambridge, MA, USA [711]. Importantly, in this study the tumor samples were analyzed by genomic alterations and for genomic signatures (genome wide loss of heterozygosity (gLOH), microsatellite instability (MSI) status, and tumor mutational burden (TMB)). An average of 3.5 genomic aberrations per primary tumor and per metastatic tumor was reported. The most frequently altered genes were those reported in other studies. The PI3K/AKT/mTOR pathway was frequently altered (40.8%) (with frequent *PTEN* abnormalities), the cell cycle pathway was altered in 23.4% of cases, the WNT pathway was altered in 16.2% of cases, homologous recombination repair-related pathway alterations were observed in 23.4% of cases, other DNA repair pathway in 4.8% of cases, *CDK2* in 5.6% of cases, and MMR genes in 4.3% of cases [711]. Importantly, overall 57% of cases harbored genomic aberrations that are investigational biomarkers for targeted therapies [711]. Some genomic aberrations are enriched in metastatic site tumors compared to primary tumors: *AR* (10.6-fold), *LYN* (3.6-fold), 11q13(*CCND1* (2.5-fold), *FGF19* (3.0-fold), *FGF3* (2.8-fold), *FGF4* (2.9-fold)), *MYC* (2.7-fold), *NCOR1* (2.1-fold), *PIK3CB* (2.7-fold), and *RB1* (2.0-fold); furthermore, G_1_/S cell cycle genes were altered in 30.7% of metastatic site versus 15.4% of primary site tumors [711].

The identification of some genetic abnormalities had potential implication for guiding targeting therapies. HRR and Fanconi Anemia genomic abnormalities have been associated with responses to PARP inhibitors [150,151,423] and *CDK12* mutations were associated in preclinical studies with immunotherapy sensitivity [305] and in HRR, affecting DNA damage response genes [712]. Importantly, the HRR abnormalities are observed in 31% of advanced prostate cancers and include: *BRCA2* (9.8%), *CDK12* (5.6%), *ATM* (5.2%), *CHEK2* (1.8%), *BRCA1* (1.4%) *FANCA* (1.3%), and *ATR* (1.1%) [711].

The definition of a defective homologous recombination repair deficiency is supported also through the study of gLOH signature: gLOH-high was associated with *BRCA1/2* alterations and *ATR* or *FANCA* genomic alterations [711]. The definition of MSI-H and TMB-H helps in the identification of biomarkers of sensitivity to immunotherapies; in many cases TMB-H overlaps with MSI-H [711].

These findings support the screening of prostate cancer patients for microsatellite instability and the investigation of the mechanisms inducing resistance to immunotherapy in a part of patients with MSI-H/dMMR phenotype [642]. As above discussed, ~5% of mCRPC patients harbor genetic alterations of *CDK12* and these tumors are associated with a high tumor neoantigen burden, which might increase the probability of response to immunechekpoint inhibition [305]. As above discussed, it was shown that ~11% of men with advanced prostate cancer display germline defects in DNA repair genes and a comparable proportion of mCRPC harbor somatic alterations in these genes as well. These patients respond better to PARP inhibitors than to standard treatments and some current studies are exploring PARP inhibitors in association with AR inhibitors. Thus, the current studies suggest that germline mutations in DNA damage repair genes (*BRCA1*, *BRCA2*, *ATM*, and *PALB2*) and in DNA mismatch repair genes (*MHL1*, *MSH2*, *MSH6*, and *PMS2*) can drive the development of prostate cancer and this finding supports germline screening of pathogenic mutations. Interestingly, among common cancers, prostate cancer was found the most heritable [713], and genome wide-association studies have identified 150 variants associated with prostate cancer [714]. Familial clustering of prostate cancers was reported and ~5% of cases of prostate cancer could be directly related to highly penetrant mutations at the level of *BRCA1*, *BRCA2*, and *HOXB13* [715]. A large screening in a Japanese population provided evidence that among eight genes (*ATM*, *BRCA1*, *BRCA2*, *BRIP1*, *CHEK2*, *HOXB13*, *NBN*, and *PALB2*), whose rare germline variants show high penetrance for prostate cancer, germline mutations of *BRCA2*, *HOXB13*, and *ATM* globally observed in 2.9% of patients, were significantly associated with prostate cancer [716].

The emerging evidences deriving from these studies support the implementation of germline genetic counseling and testing as an important component in the current and more advanced procedures of prostate cancer management [717,718].

The early diagnosis of BRCA2 mutations is important because these mutations have been associated with earlier onset and highly aggressive prostate cancers with poorer outcomes and with a higher sensitivity to PARP inhibitors and platinum-based chemotherapies. This global screening and treatment strategy are strongly supported by the results of the PROREPAIR-B trial showing that germline BRCA2 mutations have a negative impact on mCRPC outcomes that may be affected by the first line of treatment used, as above discussed [432].

As above mentioned, clinical trials have supported the use of PARP inhibitors for advanced cancers carrying BRCA1/2 or ATM mutations, thus receiving breakthrough therapy designation by the FDA [719,720]. Several clinical trials are exploring in prostate cancer patients the clinical utility of various PARP inhibitors [719,720]. Since almost 30% of patients with CRPC carry germline or somatic alterations in DDR genes, it is therefore evident that the identification of these mutations, the definition of the DDR defects that induce sensitivity to PARP inhibitors represents an important objective of future studies [719,720].

Other recent studies support a relevant role of PARP-1/PARP-2 in prostate cancer biology, not only related to the well-described functions in DNA damage repair, but also as regularors of transcription factors. Thus, the study of genetic models of prostate cancer showed an essential role for PARP-1 in sustaining AR transcriptional function; furthermore, PARP-1 activity increases during disease progression [721]. Furthermore, PARP-2 was shown to be a critical component in AR transcriptional machinery, through interaction with the pioneer factor FOXA1, facilitating the recruitment of AR to some enhancer regions [722]. PARP-2 expression increases during disease progression. Selective targeting of PARP-2 by genetic or pharmacological approaches blocks the interaction between PARP-2 and FOXA1, attenuating AR-mediated gene expression and inhibiting AR-positive prostate cancer growth [722]. Interestingly, the ongoing clinical studies involving the administration of supraphysiological androgen levels, inducing AR-mediated induction of DNA double-strand breaks, cell cyle arrest, and cellular senescence showed preferential responsiveness of prostate cancer patients with mutations in genes mediating homology-directed DNA repair [723].

Another major contribution of genomic profiling study consists in providing potential biomarkers to predict the sensitivity to standard treatments using new generation AR antagonists. The improvement of the sensitivity anf the precision of the cfDNA techniques allowed the unique opportunity of monitoring in the time during treatment the genomic profiling of prostate cancer patients undergoing treatment with AR antagonists. In thisn context, recent studies strongly support the clinical utility of the introduction of these techniques in clinical practice. The study by Torquato and coworkersexplored genetic alterations detectable in cfDNA in a group of mCRPC patients treated with Enzalutamide or Abiraterone, before treatment and at disease progression [724]. In these patients, elevated cfDNA was associated with a worse PSA response, PFS and OS; furthermore, AR ligand binding domain mutations were associated with shorter PFS in multivariable models and TP53 and PI3K pathway defects were associated with a shorter OS in multivariable models [725].

In addition to the detection of genomic mutations in plasma DNA, the evaluation of circulating tumor cells is also a tool to predict the therapy response of prostate cancer patients undergoing various types of treatments. Thus, a recent study by Salami and coworkers showed that in a group og highg-risk prostate cancer patients with localized disease undergoing treatment with radical prostectomy or radiotherapy plus androgen deprivation therapy, biochemical recurrence post-therapy was associated with a higher number of CTCs [725].

These observations, as well as many of the basic studies analyzed and discussed in this review strongly support the introduction of molecular biomarkers as a selection guide for inclusion criteria of selected patients in clinical targeted therapy. The development of these studies should help to define subsets of prostate cancer patients responding to new therapies and possibly to prolongate their survival.

As above discussed, recent studies in whole genome and transcriptome sequencing of advanced prostate cancer have shown that some mutations observed in these tumors are located at the level of nonprotein-coding regions of the genome and lead to dysregulated gene expression [726]. A remarkable example is given by a genomic rearrangement determining the tandem duplication of an intergenic tandem enhancer element located 600–700 kilobase-pairs upstream the AR gene identified in CRPC patients [68,163,164]. Some factors involved in enhancer and super-enhnacer proteins such as bromodomain proteins, represent potential drug targets for prostate cancer therapy [726]. Among the various bromodomain regulators, particularly interesting id BRD4, an epigenetic reader protein in the BET family, which binds to enhancers and super-enhnacers of several genes involved in the control of cell proliferation and involved in tumor cell transcriptional addiction. Several observations support BRD4 as a potential therapeutic target in prostate cancer: (a) SPOP mutations impair ubiquitin-dependent proteasomal degradation of BRD4, upregulating BRD4 levels; (b) DUB3 deubiquitinates BRD4, increasing its levels and promoting prostate cancer progression [727]; (c) BRD4 impedes mitochondrial fission at the level of prostate cancer stem cells through induction of mitochondrial fission factor (MFF) [728]; and (d) BRD4 regulates metastatic potential of castration-resistant prostate cancer through AHNAK, a large scaffolding protein linked to promotion of metastasis [729]; BRD4 and ZFX modulate noncanonical oncogenic functions of the AR splice varian 7 in CRPC cells [730].

As above discussed, targeting de novo-fatty acid synthesis represents a vulnerability of CRPC exploitable at clinical level [242,454]. High-fat diets promote PC development and lipid metabolism, rewire the PC metabolome to support tumor growth, and increase resistance to endocrine therapies [731]. Pharmacological suppression of FASN may represent a new therapeutic tool to both rewire the PC metabolome and to target AR signaling.

As above observed, some recent studies indicate that FOXA1 is a pioneer transcription factor essential for prostate gland development and frequently mutated in prostate cancer. However, it is unclear the precise contribution of FOXA1 alterations in prostate cancer development since this transcription factor may exert both tumor-suppressive and oncogenic roles. Two very recent studies have greatly contributed to clarify this issue. Thus, parolia and coworkers, through the analysis of 1,546 prostatye cancerts reached the conclusion that FOXA1 alterations fall into three structural classes, diverging in clinical incidence and in the frequency of genetic co-alterations profiles: class-1 activating mutations orioginate in early prostate cancer development, structurally involve the WING-2 section of nthe DNA-binding forkhead domain, and occur without alterations in ETS or SPOP and strongly induce a luminal AR program of prostate oncogenesis; class-2 mutations occur in metastatic prostate cancers, structurally correspond to truncation of the C-terminal domain of FOXA1, induce dominant chromatin binding by enhancing DNA affinity and promote metastasis though WNT pathway activation; class-3 mutations are more frequent in metastatic prostate cancer, structurally involve duplications or translocations within the FOXA1 locus and determine overexpression of FOXA1 and of other oncogenes [732]. These observations support a central role of FOXA1 in mediating oncogenesis driven by AR [732]. In the second study, Adams and coworkers have analyzed 3,086 prostate cancers and have defined two hot spots in the forkhead domain involved in FOXA1 mutatrioins: WING2 (corresponding to about 50% of all mutations) and the DNA-contact residue R219 (about 5% of all mutations); WING2 mutations are observed in adenocarcinomas at all stages, while R219 mutations are enriched in neuroendocrine prostate cancers [733]. The large majority of FOXA1 mutants observed in prostate cancers promote a pronounced luminal differentiation program, whereas R219 mutants block luminal differentiation and promote a mesenchymal and neuroendocrine transcriptional program [734].

## Figures and Tables

**Figure 1 medicines-06-00082-f001:**
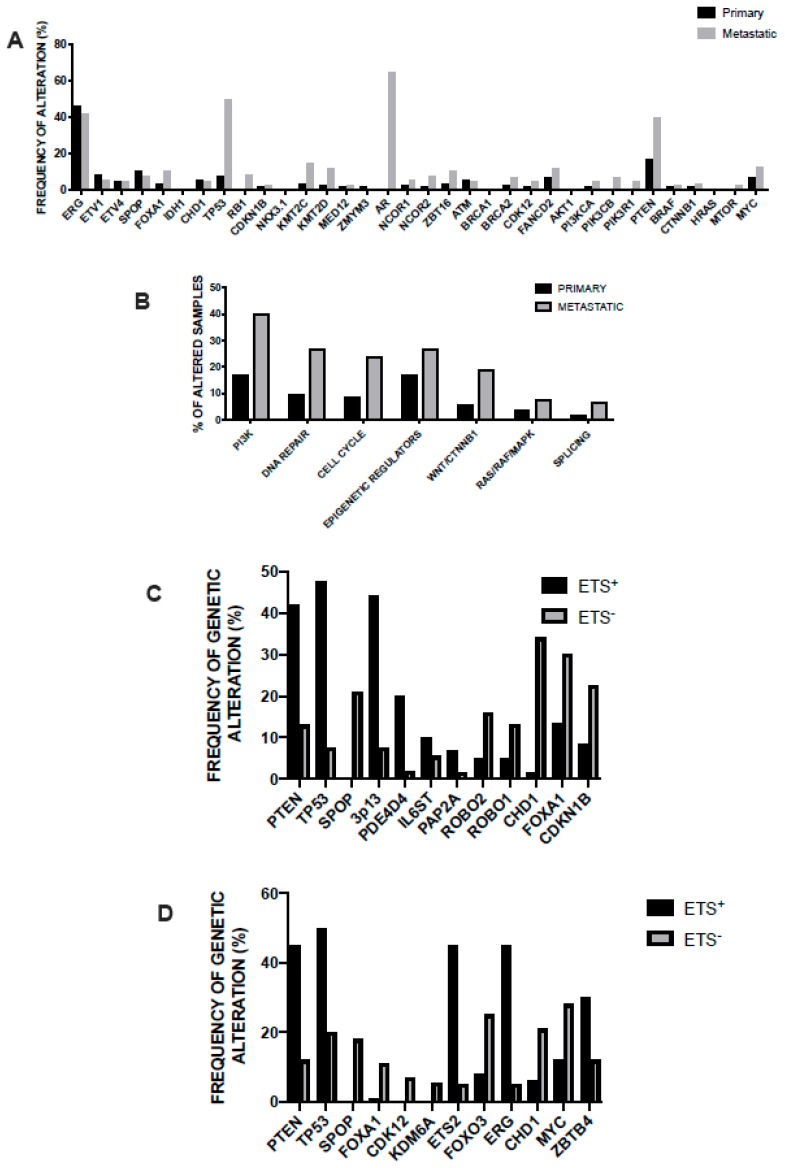
(**A**) Comparison of the main genetic alterations (copy number alterations and mutations) observed in primary and metastatic prostate cancer. The data are reported in the study of the Cancer Genome Atlas Project (TCGA) [38]. (**B**) Comparison of the alterations in signaling and biochemical pathways observed in primary and metastatic prostate cancer. The data are reported in Armenia et al. [61]. (**C**) Common genomic alterations observed in prostate cancer (mostly primary cancers) patients subdivided according to the presence of ERG gene fusions into ETS^+^ and ETS^-^. The data are reported in Wedge et al. 2018 [65]. (**D**) Common genomic alterations observed in primary prostate cancer patients subdivided into ETS^+^ and ETS^-^ groups according to the presence of ETS gene fusions. Data are reported in Xiao et al., 2018 [66].

**Figure 2 medicines-06-00082-f002:**
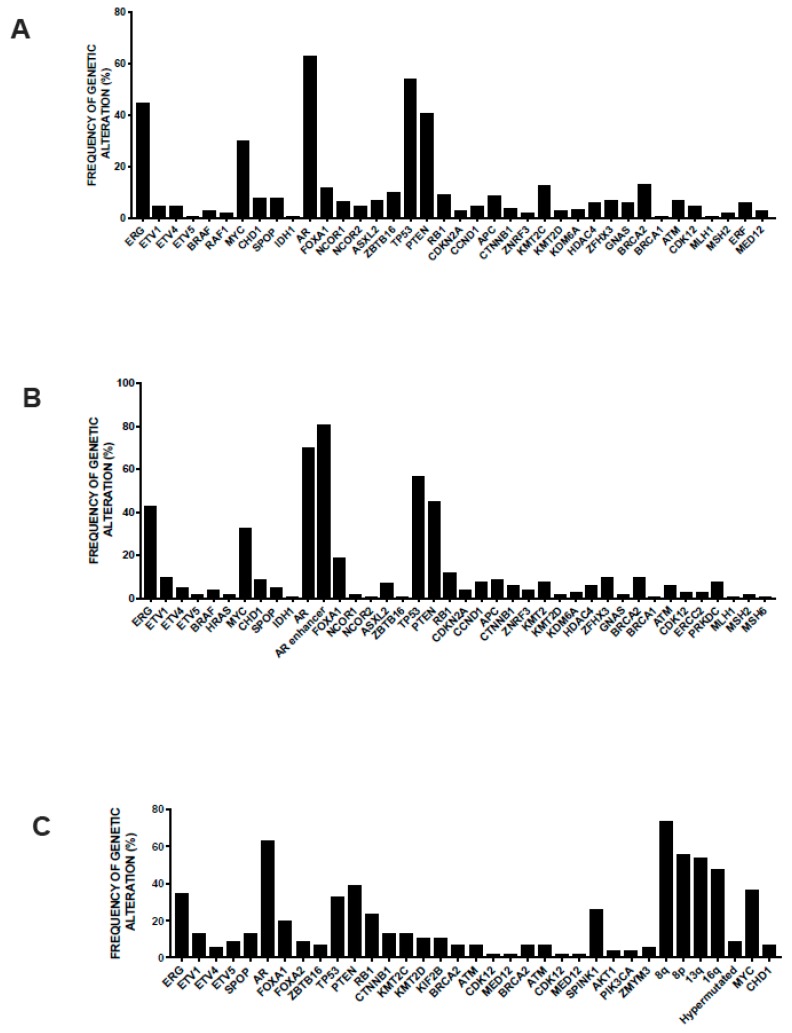
(**A**) Most recurrent somatic and germline genetic alterations observed in metastatic CRPC through DNA and RNA sequencing of clinical biopsies. The data are reported by Robinson et al., 2015 [67]. (**B**) Recurrent somatic genetic alterations in metastatic lesions of CRPC patients through whole genome sequencing. The data are reported in Quigley et al., 2018 [68]. (**C**) Recurrent somatic molecular aberrations observed in metastatic lesions of CRPC patients analyzed by wide exome sequencing. The data are reported in Kumar et al. 2016 [73].

**Figure 3 medicines-06-00082-f003:**
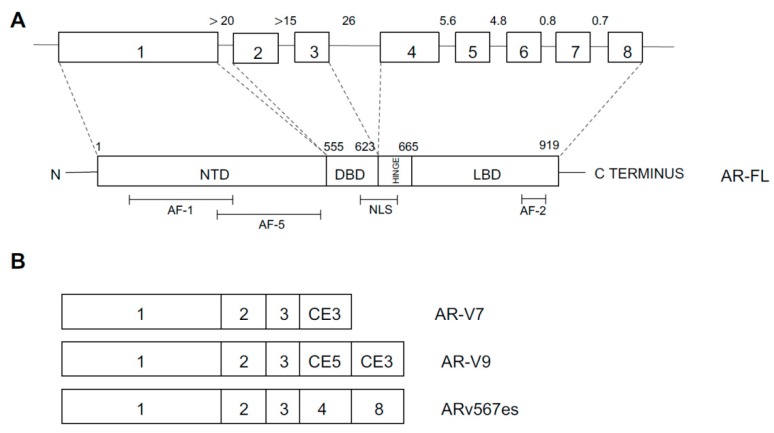
Gene and protein organization of AR and AR variants. (**A**) Human AR is encoded by a single gene located at Xq 11–12 and is normally organized into 8 exons, encoding a protein of 919 amino acids. The full-length AR protein (AR-FL) is divided into structural and functional domains: (i) a large amino terminal transactivation domain (NTD) containing activation function-1 (AF-1) and activation function-5 (AF-%); (ii) a DNA-binding domain (DBD); and (iii) a small hinge region, containing a nuclear localization signal (NLS), and a ligand-binding domain (LBD), containing activation function-2 (AF-2). It is important to note that cryptic exons (CE) are located either between exon2 and 3 (CE 2b/CE4) or between exon 3 and 4 (CE1, CE2, CE3, CE5, and 3′): alternative splicing of CEs can give rise to carboxy-terminally truncated AR isoforms. (**B**) Structure of three clinically relevant AR-Vs: AR-V7, AR-V9 and Arv567es.

**Figure 4 medicines-06-00082-f004:**
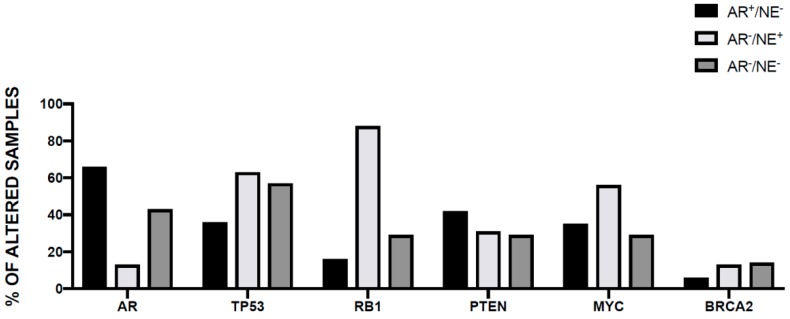
Molecular features (recurrent genomic alterations) or prostate cancer (CRPC) subdivided into three subgroups according to the expression and signaling activity of AR into AR^+^ and AR^-^ and to the presence or not of neuroendocrine features into NE^+^ and NE^-^. The data are reported in Bluemn et al. [215].

**Figure 5 medicines-06-00082-f005:**
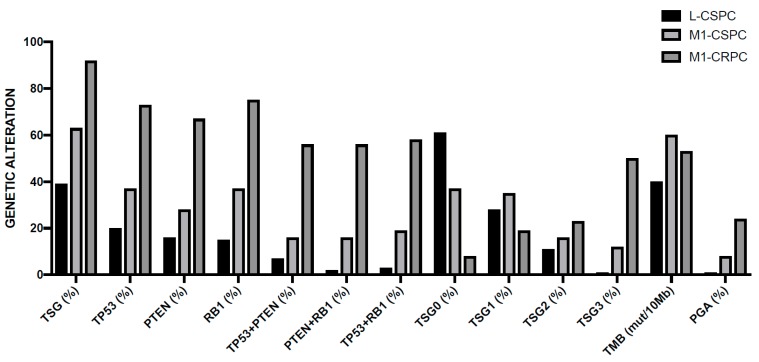
Frequency and distribution of genetic alterations of PTEN, RB1 and TP53 genes and measures of genomic instability in three groups of prostate cancer patients: L-CSPC (Localized Castration-Sensitive Prostate Cancer), M1-CSPC (Metastatic Castration-Sensitive Prostate Cancer), and M1-CRPC (Metastatic Castration-Resistant Prostate Cancer). TSG: Tumor Suppressor Gene. TMB: Tumor Mutation Burden. PGA: Percent of Genome Copy number Altered. Data are reported from Hamid et al., 2019 [233].

**Table 1 medicines-06-00082-t001:** Recurrent copy number alterations observed in prostate cancers (data reported in Refs. [78,79]).

Chromosome Region	Genetic Event	Genes Involved	Frequency in Primary Tumors (%)	Frequency in Advanced Tumors (%)
2q	Deletion	CXCR4	23	61
3p13	Deletion	FOXP1, RYBP, SQQ1	20	32
5q	Deletion	CHD1, APC	36	76
6q	Deletion	MAP3K7, ZNF292	41	74
8p	Deletion	NKX3-1, PPP2B2A	56	90
10q	Deletion	PTEN	26	83
12p	Deletion	CDKN1B	24	53
13q	Deletion	BRCA2, RB1	45	90
16q	Deletion	CDH1	44	90
17p	Deletion	TP53	28	78
17q	Deletion	BRCA1, ETV4	17	41
18q	Gain	SMAD4, BCL2	25	67
3q	Gain	PI3KCA, ETV5	10	61
7	Gain	ETV1, EGFR, MCM7, BRAF	14	75
8q	Gain	MYC	21	84
16p	Gain	-	18	64
21q	Fusion	ERG, TMPRSS2	25	48

**Table 2 medicines-06-00082-t002:** Different drugs used as androgen inhibitors for prostate cancer treatment.

Drug Category	Compound	Chemical Structure	Mechanism of Action
First-generation antiandrogen	Bicalutamide	Nonsteroidal antiandrogen	It blocks the effects of androgens at AR level
First-generation antiandrogen	Flutamide	Nonsteroidal antiandrogen	It blocks the effects of androgens at AR level
First-generation antiandrogen	Nilutamide	Nonsteroidal antiandrogen	It blocks the effects of androgens at AR level
First-generation antiandrogen	Cyproterone	Derivative of progesterone	It binds to AR and blocks the effects of testosterone and DHT
GnRH Agonist	Leuprorelin	Synthetic analog of gonadotropin-releasing hormone (GnRH)	It binds with high affinity to GnRH receptor on anterior pituitary cells, where it acts as an agonist
GnRH Agonist	Triptorelin	Synthetic analog of GnRH: more potent than native hormone and more resistant to proteolysis	It binds with high affinity to GnRH receptor on anterior pituitary cells, where it acts as an agonist
GnRH Agonist	Goserelin	Synthetic analog of GnRH: more potent than native hormone and more resistant to proteolysis	It binds with high affinity to GnRH receptor on anterior pituitary cells, where it acts as an agonist
GnRH Agonist	Degorelix	Synthetic peptide derivate of GnRH	It binds with high affinity to GnRH receptor on anterior pituitary cells, where it acts as an agonist
GnRH Agonist	Relugolix	Synthetic nonpeptide analog of GnRH	It binds with high affinity to GnRH receptor on anterior pituitary cells, inhibits the secretion of FSH and LH, preventing the release of testosterone by Leidig cells
Second-generation androgen inhibitors	Abiraterone	Steroidal compound inhibitor of androgen synthesis	It blocks the enzyme cytochrome P450 1 alpha-hydroxylase (CYP17), an enzyme required for testosterone synthesis
Second-generation androgen inhibitors	Enzalutamide	Synthetic AR signaling inhibitor	It blocks AR signaling at three key stages: it blocks the binding of androgens to AR, it inhibits nuclear translocation of activated AR and impairs binding of activated AR with DNA
Second-generation androgen inhibitors	Apalutamide	Small molecule synthetic AR antagonist	It selectively binds to the ligand-binding domain of AR and blocks nuclear translocation and binding to androgen response elements
Second-generation androgen inhibitors	Darolutamide	Nonsteroidal AR antagonist	It selectively binds to the ligand-binding domain of AR and blocks nuclear translocation and binding to androgen response elements
Second-generation androgen inhibitors	Galeterone	Steroidal compound inhibitor of androgen synthesis	It blocks the enzyme CY17, acts as an AR antagonist, promotes AR degradation
Second-generation androgen inhibitors	EPI-506	Nonsteroidal small molecule AR antagonist	It selectively binds to the NTD of AR and blocks AR signaling

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
