# Peer review of "Cellular and Molecular Mechanisms Underlying Prostate Cancer Development: Therapeutic Implications"

_medicines, 2019, doi:10.3390/medicines6030082_

Round 1
Reviewer 1 Report
In this article, Testa, Castelli and Pelosi make an extensive review about the molecular and cellular mechanisms involved in prostate cancer, discussing how the new genomic profiling techniques can help to introduce new molecular biomarkers directed to targeted therapy. As said above, it is an interesting review, compiling a lot of information and I think this great effort should be vaIued. Hoever, I think several points must be improved:
Major concerns
In my opinion, though, the review is too long, the reader gets exhausted throughout so many pages, and the division of the subsections is not the most appropriated. After a short introduction (1), a section dedicated to tumor evolution of prostate cancer from precursor lesions (2) and (3) Genetic abnormalities of prostate cancer, the authors develop most of the points in the last section: (4) More (most?) recurrent genetic abnormalities observed in breast cancer, is too long, should start by the androgen receptor abnormalities, (and afterwards, PTEN, MYC, P53, Rb, etc alterations can follow). I do not understand why racial influences, response to immunotherapy, Stem cells or hormonal regulation are included in the section 4. I would also suggest to create an independent section for androgen-dependent and androgen-independent prostate cancer treatments. The conclusion is far too long and includes explanations based in even more articles, that should be discussed in previous sections, not in the conclusion.
Minor concerns:
Many misspellings throughout the article: examples: "vulnerabiolity" in page 324 or "thisn" in page 4369. Please check the text carefully.
The reference section should be profoundly revised; there are many mistakes (the most obvious one is ref 72), many journal titles are in italic, others are not, etc…
A short paragraph dedicated to prostate cancer and light pollution might be interesting.
There are many acronyms in the text. Explain the first time they appear in the text what they stand for.
There are many interesting papers published in IJMS suitable for this review and none of them has been included (examples: Criscuolo et al., 2019 25;20(12); Dougan et al., 2019 20(12); Baumgart et al., 2019, 20(11); Stoykova et al., 2019 20(11); Frame et al., 2019 20(10); Eventually, I found one of them (Khurana et al, ref 575) by the name of the journal is wrong ( Mol. Sci, instead of IJMS)
Author Response
First, we would like to acknowledge the reviewer for his/her helpful comments.
The manuscript was now reorganized to make clear that sections concerning racial influences, response to immunotherapy, stem cells or hormonal regulation are separate sections are separate sections and do dot make part of section 4. The last section of the manuscript was now modified to make clear that this section concerns current topics and conclusions.
Text misspellings have been corrected.
Reference was made now in the last section to the suggested IJMS papers.
Reviewer 2 Report
The review article is very well written and covers the relevant literature. It also includes a lot of manuscripts which are at present in press, providing an up-to-date overview of current findings in the field of prostate cancer.
Considering the length of this manuscript, there are some typos which should be corrected.e.g.
West Coast Cancer Dream Team
2522: Their
2581: targeting
2593: montherapy
Immunocheckpoint inhibitors instead of immunocheck inhibitors
2675: CNA instead of CAN
consistency STAT/Stat and IL6/IL-6
In light of the fact that this is already a very comprehensive review, it feels not appropriate to suggest more aspects to be included. However, I personally would have liked a paragraph on current models in prostate cancer and how they react to current treatment modalities. This would be something for the authors to consider.
Author Response
First, we want to acknowledge the referee for his/her comments.
The numerous mistakes were corrected.
A paragraph analyzing and discussing the various current models in prostate cancer was now included and discussed.
Reviewer 3 Report
This is a very extensive effort to merge and explain histology and cancer outcomes with genetic evidence for prostate cancer. The review is very good. However, it is full of grammatical and spelling errors; a few are listed-
to suggest the existence of two main molecular groups of ovarian cancers: one characterized by the; line 20
Gleason patter 4 (score 4+4) corresponds to the presence of cribriform, poorly-formed, fused glands; line 82
that HGPIN is only distant precursors of adjacent invasive prostatic adenocarcinoma 16. line 163
Other two prostate tumor lesions have a "large gland" morphology, at variance with the large line 182 recurrence and mortality 21. Few information’s are available about the molecular features line 194 It goes on to the end getting worse as the reviewer progresses into the document. Also consider giving sub-titles to sections 1-3 as you did in section 4 to help the reader
Author Response
Mistakes present in the manuscript have been corrected.
Subtitles for sections 1-3 were now provided.
The subdivision of the manuscript in sections and subsections was now improved, to simplify the reading of the manuscript.
Round 2
Reviewer 1 Report
The authors have reorganized the manuscript, corrected text misspellings and included the references suggested, thus, I think the manuscript is now suitable for publication in Medicines